# Provably Efficient Multi-Task Reinforcement Learning with Model Transfer

**Chicheng Zhang**
University of Arizona
chichengz@cs.arizona.edu

**Zhi Wang**
University of California San Diego
zhiwang@eng.ucsd.edu

## Abstract

We study multi-task reinforcement learning (RL) in tabular episodic Markov decision processes (MDPs). We formulate a heterogeneous multi-player RL problem, in which a group of players concurrently face similar but not necessarily identical MDPs, with a goal of improving their collective performance through inter-player information sharing. We design and analyze an algorithm based on the idea of model transfer, and provide gap-dependent and gap-independent upper and lower bounds that characterize the intrinsic complexity of the problem.

## 1 Introduction

In many real-world applications, reinforcement learning (RL) agents can be deployed as a group to complete similar tasks at the same time. For example, in healthcare robotics, robots are paired with people with dementia to perform personalized cognitive training activities by learning their preferences [42, 21]; in autonomous driving, a set of autonomous vehicles learn how to navigate and avoid obstacles in various environments [27]. In these settings, each learning agent alone may only be able to acquire a limited amount of data, while the agents as a group have the potential to collectively learn faster through sharing knowledge among themselves. Multi-task learning [7] is a practical framework that can be used to model such settings, where a set of learning agents share/transfer knowledge to improve their collective performance.

Despite many empirical successes of multi-task RL (see, e.g., [51, 28, 27]) and transfer learning for RL (see, e.g., [26, 39]), a theoretical understanding of when and how information sharing or knowledge transfer can provide benefits remains limited. Exceptions include [16, 6, 11, 17, 32, 25], which study multi-task learning from parameter or representation transfer perspectives. However, these works still do not provide a completely satisfying answer: for example, in many application scenarios, the reward structures and the environment dynamics are only slightly different for each task—this is, however, not captured by representation transfer [11, 17] or existing works on clustering-based parameter transfer [16, 6]. In such settings, is it possible to design provably efficient multi-task RL algorithms that have guarantees never worse than agents learning individually, while outperforming the individual agents in favorable situations?

In this work, we formulate an online multi-task RL problem that is applicable to the aforementioned settings. Specifically, inspired by a recent study on multi-task multi-armed bandits [43], we formulate the $\epsilon$-Multi-Player Episodic Reinforcement Learning (abbreviated as $\epsilon$-MPERL) problem, in which all tasks share the same state and action spaces, and the tasks are assumed to be similar—i.e., the dissimilarities between the environments of different tasks (specifically, the reward distributions and transition dynamics associated with the players/tasks) are bounded in terms of a dissimilarity parameter $\epsilon \geq 0$. This problem not only models concurrent RL [34, 16] as a special case by taking $\epsilon = 0$, but also captures richer multi-task RL settings when $\epsilon$ is nonzero. We study regret minimization for the $\epsilon$-MPERL problem, specifically:

35th Conference on Neural Information Processing Systems (NeurIPS 2021).

1. We identify a problem complexity notion named *subpar* state-action pairs, which captures the amenability to information sharing among tasks in $\epsilon$-MPERL problem instances. As shown in the multi-task bandits literature (e.g., [43]), inter-task information sharing is *not* always helpful to reduce the players' collective regret. Subpar state-action pairs, intuitively speaking, are clearly suboptimal for all tasks, for which we can robustly take advantage of (possibly biased) data collected for other tasks to achieve a lower regret in a certain task.

2. In the setting where the dissimilarity parameter $\epsilon$ is known, we design a model-based algorithm MULTI-TASK-EULER (Algorithm 1), which is built upon state-of-the-art algorithms for learning single-task Markov decision processes (MDPs) [3, 46, 36], as well as algorithmic ideas of model transfer in RL [39]. MULTI-TASK-EULER crucially utilizes the dissimilarity assumption to robustly take advantage of information sharing among tasks, and achieves regret upper bounds in terms of subpar state-action pairs, in both (value function suboptimality) gap-dependent and gap-independent fashions. Specifically, compared with a baseline algorithm that does not utilize information sharing, MULTI-TASK-EULER has a regret guarantee that: (1) is never worse, i.e., it avoids negative transfer [33]; (2) can be much superior when there are a large number of subpar state-action pairs.

3. We also present gap-dependent and gap-independent regret lower bounds for the $\epsilon$-MPERL problem in terms of subpar state-action pairs. These lower bounds nearly match the upper bounds when the episode length of the MDP is a constant. Together, the upper and lower bounds characterize the intrinsic complexity of the $\epsilon$-MPERL problem.

## 2 Preliminaries

Throughout this paper, we denote by $[n] := \{1, \ldots, n\}$. For a set $A$ in a universe $U$, we use $A^C = U \setminus A$ to denote its complement. Denote by $\Delta(\mathcal{X})$ the set of probability distributions over $\mathcal{X}$. For functions $f, g$, we use $f \lesssim g$ or $f = O(g)$ (resp. $f \gtrsim g$ or $f = \Omega(g)$) to denote that there exists some constant $c > 0$, such that $f \leq cg$ (resp. $f \geq cg$), and use $f \approx g$ to denote $f \lesssim g$ and $f \gtrsim g$ simultaneously. Define $a \vee b := \max(a, b)$, and $a \wedge b := \min(a, b)$. We use $\mathbb{E}$ to denote the expectation operator, and use $\mathrm{var}$ to denote the variance operator. Throughout, we use $\tilde{O}(\cdot)$ and $\tilde{\Omega}(\cdot)$ notation to hide polylogarithmic factors.

**Multi-task RL in episodic MDPs.** We have a set of $M$ MDPs $\left\{ \mathcal{M}_p = (H, \mathcal{S}, \mathcal{A}, p_0, \mathbb{P}_p, r_p) \right\}_{p=1}^{M}$, each associated with a player $p \in [M]$. Each MDP $\mathcal{M}_p$ is regarded as a task. The MDPs share the same episode length $H \in \mathbb{N}_+$, finite state space $\mathcal{S}$, finite action space $\mathcal{A}$, and initial state distribution $p_0 \in \Delta(\mathcal{S})$. Let $\perp$ be a default terminal state that is not contained in $\mathcal{S}$. The transition probabilities $\mathbb{P}_p : \mathcal{S} \times \mathcal{A} \rightarrow \Delta(\mathcal{S} \cup \{\perp\})$ and reward distributions $r_p : \mathcal{S} \times \mathcal{A} \rightarrow \Delta([0,1])$ of the players are not necessarily identical. We assume that the MDPs are layered[1], in that the state space $\mathcal{S}$ can be partitioned into disjoint subsets $(\mathcal{S}_h)_{h=1}^{H}$, where $p_0$ is supported on $\mathcal{S}_1$, and for every $p \in [M]$, $h \in [H]$, and every $s \in \mathcal{S}_h, a \in \mathcal{A}$, $\mathbb{P}_p(\cdot \mid s, a)$ is supported on $\mathcal{S}_{h+1}$; here, we define $\mathcal{S}_{H+1} = \{\perp\}$. We denote by $S := |\mathcal{S}|$ the size of the state space, and $A := |\mathcal{A}|$ the size of the action space.

**Interaction process.** The interaction process between the players and the environment is as follows: at the beginning, both $(r_p)_{p=1}^{M}$ and $(\mathbb{P}_p)_{p=1}^{M}$ are unknown to the players. For each episode $k \in [K]$, conditioned on the interaction history up to episode $k-1$, each player $p \in [M]$ independently interacts with its respective MDP $\mathcal{M}_p$; specifically, player $p$ starts with state $s_{1,p}^k \sim p_0$, and at every step (layer) $h \in [H]$, it chooses action $a_{h,p}^k$, transitions to next state $s_{h+1,p}^k \sim \mathbb{P}_p(\cdot \mid s_{h,p}^k, a_{h,p}^k)$ and receives a stochastic immediate reward $r_{h,p}^k \sim r_p(\cdot \mid s_{h,p}^k, a_{h,p}^k)$; after all players have finished their $k$-th episode, they can communicate and share information. The goal of the players is to maximize their expected collective reward $\mathbb{E}\left[ \sum_{k=1}^{K} \sum_{p=1}^{M} \sum_{h=1}^{H} r_{h,p}^k \right]$.

**Policy and value functions.** A deterministic, history-independent policy $\pi$ is a mapping from $\mathcal{S}$ to $\mathcal{A}$, which can be used by a player to make decisions in its respective MDP. For player $p$ and step $h$, we

---

[1]This is a standard assumption (see, e.g., [44]). It is worth noting that any episodic MDP (with possibly nonstationary transition and reward) can be converted to a layered MDP with stationary transition and reward, with the state space size being $H$ times the size of the original state space.

use $V_{h,p}^\pi : \mathcal{S}_h \to [0, H]$ and $Q_{h,p}^\pi : \mathcal{S}_h \times \mathcal{A} \to [0, H]$ to denote its respective value and action-value functions, respectively. They satisfy the following recurrence known as the Bellman equation:

$$\forall h \in [H]: \quad V_{h,p}^\pi(s) = Q_{h,p}^\pi(s, \pi(s)), \quad Q_{h,p}^\pi(s, a) = R_p(s, a) + (\mathbb{P}_p V_{h+1,p}^\pi)(s, a),$$

where we use the convention that $V_{H+1,p}^\pi(\bot) = 0$, and for $f : \mathcal{S}_{h+1} \to \mathbb{R}$, $(\mathbb{P}_p f)(s, a) := \sum_{s' \in \mathcal{S}_{h+1}} \mathbb{P}_p(s' \mid s, a) f(s')$, and $R_p(s, a) := \mathbb{E}_{\hat{r} \sim r_p(\cdot | s, a)}[\hat{r}]$ is the expected immediate reward of player $p$. For player $p$ and policy $\pi$, denote by $V_{0,p}^\pi = \mathbb{E}_{s_1 \sim p_0}\left[V_{1,p}^\pi(s_1)\right]$ its expected reward.

For player $p$, we also define its optimal value function $V_{h,p}^\star : \mathcal{S}_h \to [0, H]$ and the optimal action-value function $Q_{h,p}^\star : \mathcal{S}_h \times \mathcal{A} \to [0, H]$ using the Bellman optimality equation:

$$\forall h \in [H]: \quad V_{h,p}^\star(s) = \max_{a \in \mathcal{A}} Q_{h,p}^\star(s, a), \quad Q_{h,p}^\star(s, a) = R_p(s, a) + (\mathbb{P}_p V_{h+1,p}^\star)(s, a), \quad (1)$$

where we again use the convention that $V_{H+1,p}^\star(\bot) = 0$. For player $p$, denote by $V_{0,p}^\star = \mathbb{E}_{s_1 \sim p_0}\left[V_{1,p}^\star(s_1)\right]$ its optimal expected reward.

Given a policy $\pi$, as $V_{h,p}^\pi$ for different $h$'s are only defined in the respective layer $\mathcal{S}_h$, we "collate" the value functions $(V_{h,p}^\pi)_{h=1}^H$ and obtain a single value function $V_p^\pi : \mathcal{S} \cup \{\bot\} \to \mathbb{R}$. Formally, for every $h \in [H+1]$ and $s \in \mathcal{S}_h$,

$$V_p^\pi(s) := V_{h,p}^\pi(s).$$

We define $Q_p^\pi, V_p^\star, Q_p^\star$ similarly. For player $p$, given its optimal action value function $Q_p^\star$, any of its greedy policies $\pi_p^\star(s) \in \operatorname{argmax}_{a \in \mathcal{A}} Q_p^\star(s, a)$ is optimal with respect to $\mathcal{M}_p$.

**Suboptimality gap.** For player $p$, we define the suboptimality gap of state-action pair $(s, a)$ as $\operatorname{gap}_p(s, a) = V_p^\star(s) - Q_p^\star(s, a)$. We define the mininum suboptimality gap of player $p$ as $\operatorname{gap}_{p,\min} = \min_{(s,a):\operatorname{gap}_p(s,a)>0} \operatorname{gap}_p(s, a)$, and the minimum suboptimality gap over all players as $\operatorname{gap}_{\min} = \min_{p \in [M]} \operatorname{gap}_{p,\min}$. For player $p \in [M]$, define $Z_{p,\mathrm{opt}} := \left\{(s, a) : \operatorname{gap}_p(s, a) = 0\right\}$ as the set of optimal state-action pairs with respect to $p$.

**Performance metric.** We measure the performance of the players using their collective regret, i.e., over a total of $K$ episodes, how much extra reward they would have collected in expectation if they were executing their respective optimal policies from the beginning. Formally, suppose for each episode $k$, player $p$ executes policy $\pi^k(p)$, then the collective regret of the players is defined as:

$$\operatorname{Reg}(K) = \sum_{p=1}^M \sum_{k=1}^K \left(V_{0,p}^\star - V_{0,p}^{\pi^k(p)}\right).$$

**Baseline: individual STRONG-EULER.** A naive baseline for multi-task RL is to let each player run a separate RL algorithm without communication. For concreteness, we choose to let each player run the state of the art STRONG-EULER algorithm [36] (see also its precursor EULER [46]), which enjoys minimax gap-independent [3, 8] and gap-dependent regret guarantees, and refer to this strategy as individual STRONG-EULER. Specifically, as it is known that STRONG-EULER has a regret of $\tilde{O}(\sqrt{H^2 SAK} + H^4 S^2 A)$, individual STRONG-EULER has a collective regret of $\tilde{O}(M\sqrt{H^2 SAK} + MH^4 S^2 A)$. In addition, by a union bound and summing up the gap-dependent regret guarantees of STRONG-EULER for the $M$ MDPs altogether, it can be checked that with probability $1 - \delta$, individual STRONG-EULER has a collective regret of order[2]

$$\ln\left(\frac{MSAK}{\delta}\right)\left(\sum_{p \in [M]}\left(\sum_{(s,a) \in Z_{p,\mathrm{opt}}} \frac{H^3}{\operatorname{gap}_{p,\min}} + \sum_{(s,a) \in Z_{p,\mathrm{opt}}^C} \frac{H^3}{\operatorname{gap}_p(s,a)}\right) + MH^4 S^2 A \ln \frac{SA}{\operatorname{gap}_{\min}}\right). \quad (2)$$

---

[2]The originally-stated gap-dependent regret bound of STRONG-EULER ([36], Corollary 2.1) uses a slightly different notion of suboptimality gap, which takes an extra minimum over all steps. A close examination of their proof shows that STRONG-EULER has regret bound (2) in layered MDPs. See also Remark 21 in Appendix C.4.

Our goal is to design multi-task RL algorithms that can achieve collective regret strictly lower than this baseline in both gap-dependent and gap-independent fashions when the tasks are similar.

**Notion of similarity.** Throughout this paper, we will consider the following notion of similarity between MDPs in the multi-task episodic RL setting.

**Definition 1.** *A collection of MDPs* $(\mathcal{M}_p)_{p=1}^M$ *is said to be $\epsilon$-dissimilar, if for all $p, q \in [M]$, and $(s, a) \in \mathcal{S} \times \mathcal{A}$,*

$$\left| R_p(s, a) - R_q(s, a) \right| \leq \epsilon, \ \|\mathbb{P}_p(\cdot \mid s, a) - \mathbb{P}_q(\cdot \mid s, a)\|_1 \leq \frac{\epsilon}{H}.$$

*If this happens, we call $(\mathcal{M}_p)_{p=1}^M$ an $\epsilon$-Multi-Player Episodic Reinforcement Learning (abbrev. $\epsilon$-MPERL) problem instance.*

If the MDPs in $(\mathcal{M}_p)_{p=1}^M$ are 0-dissimilar, then they are identical by definition, and our interaction protocol degenerates to the concurrent RL protocol [34]. Our dissimilarity notion is complementary to those of [6, 16]: they require the MDPs to be either identical, or have well-separated parameters for at least one state-action pair; in contrast, our dissimilarity notion allows the MDPs to be nonidentical and arbitrarily close.

We have the following intuitive lemma that shows the closeness of optimal value functions of different MDPs, in terms of the dissimilarity parameter $\epsilon$:

**Lemma 2.** *If $(\mathcal{M}_p)_{p=1}^M$ are $\epsilon$-dissimilar, then for every $p, q \in [M]$, and $(s, a) \in \mathcal{S} \times \mathcal{A}$, $\left| Q_p^\star(s, a) - Q_q^\star(s, a) \right| \leq 2H\epsilon$; consequently, $\left| \mathrm{gap}_p(s, a) - \mathrm{gap}_q(s, a) \right| \leq 4H\epsilon$.*

## 3 Algorithm

We now describe our main algorithm, MULTI-TASK-EULER (Algorithm 1). Our model-based algorithm is built upon recent works on episodic RL that provide algorithms with sharp instance-dependent guarantees in the single task setting [46, 36]. In a nutshell, for each episode $k$ and each player $p$, the algorithm performs optimistic value iteration to construct high-probability upper and lower bounds for the optimal value and action value functions $V_p^\star$ and $Q_p^\star$, and uses them to guide its exploration and decision making process.

**Empirical estimates of model parameters.** For each player $p$, the construction of its value function bound estimates relies on empirical estimates on its transition probability and expected reward function. For both estimands, we use two estimators with complementary roles, which are at two different points of the bias-variance tradeoff spectrum: one estimator uses only the player's own data (termed *individual estimate*), which has large variance; the other estimator uses the data collected by all players (termed *aggregate estimate*), which has lower variance but can easily be biased, as transition probabilities and reward distributions are heterogeneous. Such algorithmic idea of "model transfer", where one estimates model in one task using data collected from other tasks has appeared in prior works (e.g., [39]). Specifically, at the end of episode $k$, for every $h \in [H]$ and $(s, a) \in \mathcal{S}_h \times \mathcal{A}$, the algorithm maintains its empirical count of encountering $(s, a)$ for each player $p$, along with its total empirical count across all players, respectively:

$$n_p(s, a) := \sum_{l=1}^k \mathbf{1}\left( (s_{h,p}^l, a_{h,p}^l) = (s, a) \right), \ n(s, a) := \sum_{l=1}^k \sum_{p=1}^M \mathbf{1}\left( (s_{h,p}^l, a_{h,p}^l) = (s, a) \right). \quad (3)$$

The individual and aggregate estimates of immediate reward $R(s, a)$ are defined as:

$$\hat{R}_p(s, a) := \frac{\sum_{l=1}^k \mathbf{1}\left( (s_{h,p}^l, a_{h,p}^l) = (s, a) \right) r_{h,p}^l}{n_p(s, a)}, \ \hat{R}(s, a) := \frac{\sum_{l=1}^k \sum_{p=1}^M \mathbf{1}\left( (s_{h,p}^l, a_{h,p}^l) = (s, a) \right) r_{h,p}^l}{n(s, a)}.$$
$$(4)$$

**Algorithm 1:** MULTI-TASK-EULER

**Input :** Failure probability $\delta \in (0,1)$, dissimilarity parameter $\epsilon \geq 0$.

**Initialize:** Set $V_p(\bot) = 0$ for all $p$ in $[M]$, where $\bot$ is the only state in $\mathcal{S}_{H+1}$ ;

1    **for** $k = 1, 2, \ldots, K$ **do**

2      **for** $p = 1, 2, \ldots, M$ **do**

       // Construct optimal value estimates for player $p$

3        **for** $h = H, H-1, \ldots, 1$ **do**

4          **for** $(s,a) \in \mathcal{S}_h \times \mathcal{A}$ **do**

5            Compute:

6            $\overline{\text{ind-}Q}_p(s,a) = \hat{R}_p(s,a) + (\hat{\mathbb{P}}_p \overline{V}_p)(s,a) + \text{ind-}b_p(s,a);$

7            $\underline{\text{ind-}Q}_p(s,a) = \hat{R}_p(s,a) + (\hat{\mathbb{P}}_p \underline{V}_p)(s,a) - \text{ind-}b_p(s,a);$

8            $\overline{\text{agg-}Q}_p(s,a) = \hat{R}(s,a) + (\hat{\mathbb{P}} \overline{V}_p)(s,a) + \text{agg-}b_p(s,a);$

9            $\underline{\text{agg-}Q}_p(s,a) = \hat{R}(s,a) + (\hat{\mathbb{P}} \underline{V}_p)(s,a) - \text{agg-}b_p(s,a);$

10            Update optimal action value function upper and lower bound estimates:

11            $\overline{Q}_p(s,a) = \min\left\{ H-h+1, \overline{\text{ind-}Q}_p(s,a), \overline{\text{agg-}Q}_p(s,a) \right\};$

12            $\underline{Q}_p(s,a) = \max\left\{ 0, \underline{\text{ind-}Q}_p(s,a), \underline{\text{agg-}Q}_p(s,a) \right\};$

13          **for** $s \in \mathcal{S}_h$ **do**

14            Define $\pi^k(p)(s) = \text{argmax}_{a \in \mathcal{A}} \overline{Q}_p(s,a)$ ;

15            Update $\overline{V}_p(s) = \overline{Q}_p\left(s, \pi^k(p)(s)\right), \underline{V}_p(s) = \underline{Q}_p\left(s, \pi^k(p)(s)\right).$

     // All players $p$ interact with their respective environments, and update reward and transition estimates

16      **for** $p = 1, 2, \ldots, M$ **do**

17        Player $p$ executes policy $\pi^k(p)$ on $\mathcal{M}_p$ and obtains trajectory $(s_{h,p}^k, a_{h,p}^k, r_{h,p}^k)_{h=1}^H$.

18        Update individual estimates of transition probability $\hat{\mathbb{P}}_p$, reward $\hat{R}_p$ and count $n_p(\cdot, \cdot)$ using the first parts of Equations (3), (4) and (5).

19      Update aggregate estimates of transition probability $\hat{\mathbb{P}}$, reward $\hat{R}$ and count $n(\cdot, \cdot)$ using the second parts of Equations (3), (4) and (5).

---

Similarly, for every $h \in [H]$ and $(s, a, s') \in \mathcal{S}_h \times \mathcal{A} \times \mathcal{S}_{h+1}$, we also define the individual and aggregate estimates of transition probability as:

$$
\hat{\mathbb{P}}_p(s' \mid s, a) := \frac{\sum_{l=1}^k \mathbf{1}\left( (s_{h,p}^l, a_{h,p}^l, s_{h+1,p}^l) = (s, a, s') \right)}{n_p(s,a)},
$$
$$
\hat{\mathbb{P}}(s' \mid s, a) := \frac{\sum_{l=1}^k \sum_{p=1}^M \mathbf{1}\left( (s_{h,p}^l, a_{h,p}^l, s_{h+1,p}^l) = (s, a, s') \right)}{n(s,a)}. \tag{5}
$$

If $n(s,a) = 0$, we define $\hat{R}(s,a) := 0$ and $\hat{\mathbb{P}}(s' \mid s, a) := \frac{1}{|\mathcal{S}_{h+1}|}$; and if $n_p(s,a) = 0$, we define $\hat{R}_p(s,a) := 0$ and $\hat{\mathbb{P}}_p(s' \mid s, a) := \frac{1}{|\mathcal{S}_{h+1}|}$. The counts and reward estimates can be maintained by MULTI-TASK-EULER efficiently in an incremental manner.

**Constructing value function estimates via optimistic value iteration.** For each player $p$, based on these model parameter estimates, MULTI-TASK-EULER performs optimistic value iteration to compute the value function estimates for states at all layers (lines 3 to 15). For the terminal layer $H + 1$, $V_{H+1}^\star(\bot) = 0$ trivially, so nothing needs to be done. For earlier layers $h \in [H]$, MULTI-TASK-EULER iteratively builds its value function estimates in a backward fashion. At the time of estimating values for layer $h$, the algorithm has already obtained optimal value estimates for layer $h + 1$. Based on the Bellman optimality equation (1), MULTI-TASK-EULER

estimates $(Q_p^\star(s,a))_{s\in\mathcal{S}_h, a\in\mathcal{A}}$ using model parameter estimates and its estimates of $(V_p^\star(s))_{s\in\mathcal{S}_{h+1}}$, i.e., $(\overline{V}_p(s))_{s\in\mathcal{S}_{h+1}}$ and $(\underline{V}_p(s))_{s\in\mathcal{S}_{h+1}}$ (lines 5 to 12).

Specifically, MULTI-TASK-EULER constructs estimates of $Q_p^\star(s,a)$ for all $s\in\mathcal{S}_h, a\in\mathcal{A}$ in two different ways. First, it uses the individual estimates of model of player $p$ to construct $\underline{\text{ind-}Q}_p$ and $\overline{\text{ind-}Q}_p$, upper and lower bound estimates of $Q_p^\star$ (lines 8 and 9); this construction is reminiscent of EULER and STRONG-EULER [46, 36], in that if we were only to use $\underline{\text{ind-}Q}_p$ and $\overline{\text{ind-}Q}_p$ as our optimal action value function estimate $\overline{Q}_p$ and $\underline{Q}_p$, our algorithm becomes individual STRONG-EULER. The individual value function estimates are key to establishing MULTI-TASK-EULER's fall-back guarantees, ensuring that it never performs worse than the individual STRONG-EULER baseline. Second, it uses the aggregate estimate of model to construct $\underline{\text{agg-}Q}_p$ and $\overline{\text{agg-}Q}_p$, also upper and lower bound estimates of $Q_p^\star$ (lines 6 and 7); this construction is unique to the multitask learning setting, and is our new algorithmic contribution.

To ensure that $\overline{\text{agg-}Q}_p$ and $\overline{\text{ind-}Q}_p$ (resp. $\underline{\text{agg-}Q}_p$ and $\underline{\text{ind-}Q}_p$) are valid upper bounds (resp. lower bounds) of $Q_p^\star$, MULTI-TASK-EULER adds bonus terms ind-$b_p(s,a)$ and agg-$b_p(s,a)$, respectively, in the optimistic value iteration process, to account for estimation error of the model estimates against the true models. Specifically, both bonus terms comprise three parts:

$$\text{ind-}b_p(s,a) := b_{\text{rw}}\left(n_p(s,a),0\right) + b_{\text{prob}}\left(\hat{\mathbb{P}}_p(\cdot\mid s,a), n_p(s,a), \overline{V}_p, \underline{V}_p, 0\right) +$$
$$b_{\text{str}}\left(\hat{\mathbb{P}}_p(\cdot\mid s,a), n_p(s,a), \overline{V}_p, \underline{V}_p, 0\right),$$

$$\text{agg-}b_p(s,a) := b_{\text{rw}}\left(n(s,a),\epsilon\right) + b_{\text{prob}}\left(\hat{\mathbb{P}}(\cdot\mid s,a), n(s,a), \overline{V}_p, \underline{V}_p, \epsilon\right) +$$
$$b_{\text{str}}\left(\hat{\mathbb{P}}(\cdot\mid s,a), n(s,a), \overline{V}_p, \underline{V}_p, \epsilon\right),$$

where

$$b_{\text{rw}}(n,\kappa) := 1 \ \wedge \ \kappa + \Theta\left(\sqrt{\frac{L(n)}{n}}\right),$$

$$b_{\text{prob}}\left(q,n,\overline{V},\underline{V},\kappa\right) := H \ \wedge \ 2\kappa + \Theta\left(\sqrt{\frac{\text{var}_{s'\sim q}\left[\overline{V}(s')\right]L(n)}{n}} + \right.$$
$$\left. \sqrt{\frac{\mathbb{E}_{s'\sim q}\left[(\overline{V}(s') - \underline{V}(s'))^2\right]L(n)}{n}} + \frac{HL(n)}{n}\right),$$

$$b_{\text{str}}\left(q,n,\overline{V},\underline{V},\kappa\right) := \kappa + \Theta\left(\sqrt{\frac{S\,\mathbb{E}_{s'\sim q}\left[(\overline{V}(s') - \underline{V}(s'))^2\right]L(n)}{n}} + \frac{HSL(n)}{n}\right),$$

and $L(n) \approx \ln(\frac{MSAn}{\delta})$.

The three components in the bonus terms serve for different purposes:

1. The first component accounts for the uncertainty in the reward estimation: with probability $1 - O(\delta)$, $\left|\hat{R}_p(s,a) - R_p(s,a)\right| \leq b_{\text{rw}}\left(n_p(s,a),0\right)$, and $\left|\hat{R}(s,a) - R_p(s,a)\right| \leq b_{\text{rw}}\left(n(s,a),\epsilon\right)$.

2. The second component accounts for the uncertainty in estimating $(\mathbb{P}_p V_p^\star)(s,a)$: with probability $1 - O(\delta)$, $\left|(\hat{\mathbb{P}}_p V_p^\star)(s,a) - (\mathbb{P}_p V_p^\star)(s,a)\right| \leq b_{\text{prob}}\left(\hat{\mathbb{P}}_p(\cdot\mid s,a), n_p(s,a), \overline{V}_p, \underline{V}_p, 0\right)$ and $\left|(\hat{\mathbb{P}}V_p^\star)(s,a) - (\mathbb{P}_p V_p^\star)(s,a)\right| \leq b_{\text{prob}}\left(\hat{\mathbb{P}}(\cdot\mid s,a), n(s,a), \overline{V}_p, \underline{V}_p, \epsilon\right)$.

3. The third component accounts for the lower order terms to ensure strong optimism [36]: with probability $1 - O(\delta)$, $\left|(\hat{\mathbb{P}}_p - \mathbb{P}_p)(\overline{V}_p - V_p^\star)(s, a)\right| \leq$ $b_{\text{str}}\left(\hat{\mathbb{P}}_p(\cdot \mid s, a), n_p(s, a), \overline{V}_p, \underline{V}_p, 0\right)$, and $\left|(\hat{\mathbb{P}} - \mathbb{P}_p)(\overline{V}_p - V_p^\star)(s, a)\right| \leq$ $b_{\text{prob}}\left(\hat{\mathbb{P}}(\cdot \mid s, a), n(s, a), \overline{V}_p, \underline{V}_p, \epsilon\right)$.

Based on the above concentration inequalities and the definitions of bonus terms, it can be shown inductively that, with probability $1 - O(\delta)$, both $\overline{\text{agg-}Q}_p$ and $\overline{\text{ind-}Q}_p$ (resp. $\underline{\text{agg-}Q}_p$ and $\underline{\text{ind-}Q}_p$) are valid upper bounds (resp. lower bounds) of $Q_p^\star$.

Finally, observe that for any $(s, a) \in \mathcal{S}_h \times \mathcal{A}$, $Q_p^\star(s, a)$ has range $[0, H-h+1]$. By taking intersections of all confidence bounds of $Q_p^\star$ it has obtained, MULTI-TASK-EULER constructs its final upper and lower bound estimates for $Q_p^\star(s, a)$, $\overline{Q}_p(s, a)$ and $\underline{Q}_p(s, a)$ respectively, for $(s, a) \in \mathcal{S}_h \times \mathcal{A}$ (line 11 to 12). Similar ideas on using data from multiple sources to construct confidence intervals and guide explorations have been used by [37, 43] for multi-task noncontextual and contextual bandits. Using the relationship between the optimal value $V_p^\star(s)$ and and optimal action values $\left\{Q_p^\star(s, a) : a \in \mathcal{A}\right\}$, MULTI-TASK-EULER also constructs upper and lower bound estimates for $V_p^\star(s)$, $\overline{V}_p(s)$ and $\underline{V}_p(s)$, respectively for $s \in \mathcal{S}_h$ (line 15).

**Executing optimistic policies.** At each episode $k$, for each player $p$, its optimal action-value function upper bound estimate $\overline{Q}_p$ induces a greedy policy $\pi^k(p) : s \mapsto \text{argmax}_{a \in \mathcal{A}} \overline{Q}_p(s, a)$ (line 14); the player then executes this policy at this episode to collect a new trajectory and use this to update its individual model parameter estimates. After all players finish their episode $k$, the algorithm also updates its aggregate model parameter estimates (lines 16 to 19) using Equations (3), (4) and (5), and continues to the next episode.

## 4 Performance guarantees

Before stating the guarantees of Algorithm 1, we define an instance-dependent complexity measure that characterizes the amenability to information sharing.

**Definition 3.** *The set of subpar state-action pairs is defined as:*

$$\mathcal{I}_\epsilon := \left\{(s, a) \in \mathcal{S} \times \mathcal{A} : \exists p \in [M], \text{gap}_p(s, a) \geq 96H\epsilon\right\},$$

*where we recall that* $\text{gap}_p(s, a) = V_p^\star(s) - Q_p^\star(s, a)$.

Definition 3 generalizes the notion of subpar arms defined for multi-task multi-armed bandit learning [43] in two ways: first, it is with regards to state-action pairs as opposed to actions only; second, in RL, suboptimality gaps depend on optimal value function, which in turn depends on both immediate reward and subsequent long-term return.

To ease our later presentation, we also present the following lemma.

**Lemma 4.** *For any* $(s, a) \in \mathcal{I}_\epsilon$, *we have that: (1) for all* $p \in [M]$, $(s, a) \notin Z_{p,\text{opt}}$, *where we recall that* $Z_{p,\text{opt}} = \left\{(s, a) : \text{gap}_p(s, a) = 0\right\}$ *is the set of optimal state-action pairs with respect to* $p$; *(2) for all* $p, q \in [M]$, $\text{gap}_p(s, a) \geq \frac{1}{2}\text{gap}_q(s, a)$.

The lemma follows directly from Lemma 2; its proof can be found in the Appendix along with proofs of the following theorems. Item 1 implies that any subpar state-action pair is suboptimal for all players. In other words, for every player $p$, the state-action space $\mathcal{S} \times \mathcal{A}$ can be partitioned to three disjoint sets: $\mathcal{I}_\epsilon, Z_{p,\text{opt}}, (\mathcal{I}_\epsilon \cup Z_{p,\text{opt}})^C$. Item 2 implies that for any subpar $(s, a)$, its suboptimal gaps with respect to all players are within a constant of each other.

### 4.1 Upper bounds

With the above definitions, we are now ready to present the performance guarantees of Algorithm 1. We first present a gap-independent collective regret bound of MULTI-TASK-EULER.

**Theorem 5** (Gap-independent bound). *If $\left\{\mathcal{M}_p\right\}_{p=1}^M$ are $\epsilon$-dissimilar, then* MULTI-TASK-EULER *satisfies that with probability $1 - \delta$,*

$$\text{Reg}(K) \leq \tilde{O}\left(M\sqrt{H^2|\mathcal{I}_\epsilon^C|K} + \sqrt{MH^2|\mathcal{I}_\epsilon|K} + MH^4S^2A\right).$$

We again compare this regret upper bound with individual STRONG-EULER's gap independent regret bound. Recall that individual STRONG-EULER guarantees that with probability $1 - \delta$,

$$\text{Reg}(K) \leq \tilde{O}\left(M\sqrt{H^2SAK} + MH^4S^2A\right).$$

We focus on the comparison on the leading terms, i.e., the $\sqrt{K}$ terms. As $M\sqrt{H^2SAK} \approx M\sqrt{H^2|\mathcal{I}_\epsilon|K} + M\sqrt{H^2|\mathcal{I}_\epsilon^C|K}$, we see that an improvement in the collective regret bound comes from the contributions from subpar state-action pairs: the $M\sqrt{H^2|\mathcal{I}_\epsilon|K}$ term is reduced to $\sqrt{MH^2|\mathcal{I}_\epsilon|K}$, a factor of $\tilde{O}(\sqrt{\frac{1}{M}})$ improvement. Moreover, if $|\mathcal{I}_\epsilon^C| \ll SA$ and $M \gg 1$, MULTI-TASK-EULER provides a regret bound of lower order than individual STRONG-EULER.

We next present a gap-dependent upper bound on its collective regret.

**Theorem 6** (Gap-dependent upper bound). *If $\left\{\mathcal{M}_p\right\}_{p=1}^M$ are $\epsilon$-dissimilar, then* MULTI-TASK-EULER *satisfies with probability $1 - \delta$,*

$$\text{Reg}(K) \lesssim \ln(\frac{MSAK}{\delta})\left(\sum_{p\in[M]}\left(\sum_{(s,a)\in Z_{p,\text{opt}}}\frac{H^3}{\text{gap}_{p,\min}} + \sum_{(s,a)\in(\mathcal{I}_\epsilon\cup Z_{p,\text{opt}})^C}\frac{H^3}{\text{gap}_p(s,a)}\right) + \right.$$

$$\left.\sum_{(s,a)\in\mathcal{I}_\epsilon}\frac{H^3}{\min_p\text{gap}_p(s,a)}\right) + \ln(\frac{MSAK}{\delta})\cdot MH^4S^2A\ln\frac{MSA}{\text{gap}_{\min}},$$

*where we recall that $\text{gap}_{p,\min} = \min_{(s,a):\text{gap}_p(s,a)>0}\text{gap}_p(s,a)$, and $\text{gap}_{\min} = \min_p\text{gap}_{p,\min}$.*

Comparing this regret bound with the regret bound obtained by the individual STRONG-EULER baseline, recall that by summing over the regret guarantees of STRONG-EULER for all players $p \in [M]$, and taking a union bound over all $p$, individual STRONG-EULER guarantees a collective regret bound of

$$\text{Reg}(K) \lesssim \ln(\frac{MSAK}{\delta})\left(\sum_{p\in[M]}\left(\sum_{(s,a)\in Z_{p,\text{opt}}}\frac{H^3}{\text{gap}_{p,\min}} + \sum_{(s,a)\in(\mathcal{I}_\epsilon\cup Z_{p,\text{opt}})^C}\frac{H^3}{\text{gap}_p(s,a)}\right) + \right.$$

$$\left.\sum_{(s,a)\in\mathcal{I}_\epsilon}\sum_{p\in[M]}\frac{H^3}{\text{gap}_p(s,a)}\right) + \ln(\frac{MSAK}{\delta})\cdot MH^4S^2A\ln\frac{SA}{\text{gap}_{\min}},$$

that holds with probability $1 - \delta$. We again focus on comparing the leading terms, i.e., the terms that have polynomial dependences on the suboptimality gaps in the above two bounds. It can be seen that an improvement in the regret bound by MULTI-TASK-EULER comes from the contributions from the subpar state-action pairs: for each $(s,a) \in \mathcal{I}_\epsilon$, the regret bound is reduced from $\sum_{p\in[M]}\frac{H^3}{\text{gap}_p(s,a)}$ to $\frac{H^3}{\min_p\text{gap}_p(s,a)}$, a factor of $O(\frac{1}{M})$ improvement. Recent work of [44] has shown that in the single-task setting, it is possible to replace $\sum_{(s,a)\in Z_{p,\text{opt}}}\frac{H^3}{\text{gap}_{p,\min}}$ with a sharper problem-dependent complexity term that depends on the multiplicity of optimal state-action pairs. We leave improving the guarantee of Theorem 6 in a similar manner as an interesting open problem.

Key to the proofs of Theorems 5 and 6 is a new bound on the *surplus* [36] of the value function estimates. Our new surplus bound is a minimum of two terms: one depends on the usual state-action visitation counts of player $p$, the other depends on the task dissimilarity parameter $\epsilon$ and the state-action visitation counts of all players. Detailed proofs can be found at Appendix C.

## 4.2 Lower bounds

To complement the above upper bounds, we now present gap-dependent and gap-independent regret lower bounds that also depends on our subpar state-action pair notion. Our lower bounds are inspired by regret lower bounds for episodic RL [36, 8] and multi-task bandits [43].

**Theorem 7** (Gap-independent lower bound). *For any $A \geq 2$, $H \geq 2$, $S \geq 4H$, $K \geq SA$, $M \in \mathbb{N}$, and $l, l^C \in \mathbb{N}$ with $l + l^C = SA$ and $l \leq SA - 4(S + HA)$, there exists some $\epsilon$ that satisfies: for any algorithm Alg, there exists an $\epsilon$-MPERL problem instance with $S$ states, $A$ actions, $M$ players and an episode length of $H$ such that $\left|\mathcal{I}_{\frac{\epsilon}{192H}}\right| \geq l$, and*

$$\mathbb{E}\left[\text{Reg}_{\text{Alg}}(K)\right] \geq \Omega\left(M\sqrt{H^2 l^C K} + \sqrt{MH^2 lK}\right).$$

We also present a gap-dependent lower bound. Before that, we first formally define the notion of sublinear regret algorithms: for any fixed $\epsilon$, we say that an algorithm Alg is a sublinear regret algorithm for the $\epsilon$-MPERL problem if there exists some $C > 0$ (that possibly depends on the state-action space, the number of players, and $\epsilon$) and $\alpha < 1$ such that for all $K$ and all $\epsilon$-MPERL environments, $\mathbb{E}\left[\text{Reg}_{\text{Alg}}(K)\right] \leq CK^\alpha$.

**Theorem 8** (Gap-dependent lower bound). *Fix $\epsilon \geq 0$. For any $S \in \mathbb{N}$, $A \geq 2$, $H \geq 2$, $M \in \mathbb{N}$, with $S \geq 2(H-1)$, let $S_1 = S - 2(H-1)$; and let $\left\{\Delta_{s,a,p}\right\}_{(s,a,p)\in[S_1]\times[A]\times[M]}$ be any set of values that satisfies: (1) each $\Delta_{s,a,p} \in [0, H/48]$, (2) for every $(s,p) \in [S_1] \times [M]$, there exists at least one action $a \in [A]$ such that $\Delta_{s,a,p} = 0$, and (3) for every $(s,a) \in [S_1] \times [A]$ and $p, q \in [M]$, $\left|\Delta_{s,a,p} - \Delta_{s,a,q}\right| \leq \epsilon/4$. There exists an $\epsilon$-MPERL problem instance with $S$ states, $A$ actions, $M$ players and an episode length of $H$, such that $\mathcal{S}_1 = [S_1]$, $|\mathcal{S}_h| = 2$ for all $h \geq 2$, and*

$$\text{gap}_p(s,a) = \Delta_{s,a,p}, \quad \forall(s,a,p) \in [S_1] \times [A] \times [M];$$

*for this problem instance, any sublinear regret algorithm Alg for the $\epsilon$-MPERL problem must satisfy:*

$$\mathbb{E}\left[\text{Reg}_{\text{Alg}}(K)\right] \geq \Omega\left(\ln K\left(\sum_{p\in[M]} \sum_{\substack{(s,a)\in\mathcal{I}^C_{(\epsilon/192H)}:\\ \text{gap}_p(s,a)>0}} \frac{H^2}{\text{gap}_p(s,a)} + \sum_{(s,a)\in\mathcal{I}_{(\epsilon/192H)}} \frac{H^2}{\min_p \text{gap}_p(s,a)}\right)\right).$$

Comparing the lower bounds with MULTI-TASK-EULER's regret upper bounds in Theorems 5 and 6, we see that the upper and lower bounds nearly match for any constant $H$. When $H$ is large, a key difference between the upper and lower bounds is that the former are in terms of $\mathcal{I}_\epsilon$, while the latter are in terms of $\mathcal{I}_{\Theta(\frac{\epsilon}{H})}$. We conjecture that our upper bounds can be improved by replacing $\mathcal{I}_\epsilon$ with $\mathcal{I}_{\Theta(\frac{\epsilon}{H})}$—our analysis uses a clipping trick similar to [36], which may be the reason for a suboptimal dependence on $H$. We leave closing this gap as an open question.

## 5 Related Work

**Regret minimization for MDPs.** Our work belongs to the literature of regret minimization for MDPs, e.g., [5, 18, 8, 3, 9, 19, 10, 46, 36, 49, 45, 44]. In the episodic setting, [3, 10, 46, 36, 49] achieve minimax $\sqrt{H^2 SAK}$ regret bounds for general stationary MDPs. Furthermore, the EULER algorithm [46] achieves adaptive problem-dependent regret guarantees when the total reward within an episode is small or when the environmental norm of the MDP is small. [36] refine EULER, proposing STRONG-EULER that provides more fine-grained gap-dependent $O(\log K)$ regret guarantees. [45, 44] show that the optimistic Q-learning algorithm [19] and its variants can also achieve gap-dependent logarithmic regret guarantees. Remarkably, [44] achieve a regret bound that improves over that of [36], in that it replaces the dependence on the number of optimal state-action pairs with the number of non-unique state-action pairs.

**Transfer and lifelong learning for RL.** A considerable portion of related works concerns transfer learning for RL tasks (see [40, 24, 50] for surveys from different angles), and many studies investigate a batch setting: given some source tasks and target tasks, transfer learning agents have access to batch data collected for the source tasks (and sometimes for the target tasks as well). In this setting, model-based approaches have been explored in e.g., [39]; theoretical guarantees for transfer of samples across tasks have been established in e.g., [25, 41]. Similarly, sequential transfer has been studied under the framework of lifelong RL in e.g., [38, 1, 15, 22]—in this setting, an agent faces a sequence of RL tasks and aims to take advantage of knowledge gained from previous tasks for better performance in future tasks; in particular, analyses on the sample complexity of transfer learning algorithms are presented in [6, 29] under the assumption that an upper bound on the total number of unique (and well-separated) RL tasks is known. We note that, in contrast, we study an online setting in which no prior data are available and multiple RL tasks are learned concurrently by RL agents.

**Concurrent RL.** Data sharing between multiple RL agents that learn concurrently has also been investigated in the literature. For example, in [20, 35, 16, 12], a group of agents interact in parallel with *identical* environments. Another setting is studied in [16], in which agents solve different RL tasks (MDPs); however, similar to [6, 29], it is assumed that there is a finite number of unique tasks, and different tasks are well-separated, i.e., there is a minimum gap. In this work, we assume that players face similar but not necessarily identical MDPs, and we do not assume a minimum gap. [17] study multi-task RL with linear function approximation with representation transfer, where it is assumed that the optimal value functions of all tasks are from a low dimensional linear subspace. Our setting and results are most similar to [32] and [13]. [32] study concurrent exploration in similar MDPs with continuous states in the PAC setting; however, their PAC guarantee does not hold for target error rate arbitrarily close to zero; in contrast, our algorithm has a fall-back guarantee, in that it always has a sublinear regret. Concurrent RL from similar *linear* MDPs has also been recently studied in [13]: under the assumption of small heterogeneity between different MDPs (a setting very similar to ours), the provided regret guarantee involves a term that is linear in the number of episodes, whereas our algorithm in this paper always has a sublinear regret; concurrent RL under the assumption of large heterogeneity is also studied in that work, but additional contextual information is assumed to be available for the players to ensure a sublinear regret.

**Other related topics and models.** In many multi-agent RL models [47, 31], a set of learning agents interact with a common environment and have shared global states; in particular, [48] study the setting with heterogeneous reward distributions, and provide convergence guarantees for two policy gradient-based algorithms. In contrast, in our setting, our learning agents interact with separate environments. Multi-agent bandits with similar, heterogeneous reward distributions are investigated in [37, 43]; herein, we generalize their multi-task bandit problem setting to the episodic MDP setting.

## 6 Conclusion and Future Directions

In this paper, we generalize the multi-task bandit learning framework in [43] and formulate a multi-task concurrent RL problem, in which tasks are similar but not necessarily identical. We provide a provably efficient model-based algorithm that takes advantage of knowledge transfer between different tasks. Our instance-dependent regret upper and lower bounds formalize the intuition that subpar state-action pairs are amenable to information sharing among tasks.

There still remain gaps between our upper and lower bounds which can be closed by either a finer analysis or a better algorithm: first, the dependence on $\mathcal{I}_\epsilon$ in the upper bound does not match the dependence of $\mathcal{I}_{\Theta(\epsilon/H)}$ in the lower bound when $H$ is large; second, the gap-dependent upper bound has $O(H^3)$ dependence, whereas the gap-dependent lower bound only has $\Omega(H^2)$ dependence; third, the additive dependence on the number of optimal state-action pairs can potentially be removed by new algorithmic ideas [44].

Furthermore, one major obstacle in deploying our algorithm in practice is its requirement for knowledge of $\epsilon$; an interesting avenue is to apply model selection strategies in bandits and RL to achieve adaptivity to unknown $\epsilon$. Another interesting future direction is to consider more general parameter transfer for online RL, for example, in the context of function approximation.

## 7 Acknowledgements

We thank Kamalika Chaudhuri for helpful initial discussions, and thank Akshay Krishnamurthy and Tongyi Cao for discussing the applicability of adaptive RL in metric spaces to the multitask RL problem studied in this paper. CZ acknowledges startup funding support from the University of Arizona. ZW thanks the National Science Foundation under IIS 1915734 and CCF 1719133 for research support.

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
