# A  Proofs of Lemmas 2 and 4

## A.1  Proof of Lemma 2

**Lemma 2.** *If $(\mathcal{M}_p)_{p=1}^M$ is $\epsilon$-dissimilar, then for every $p, q \in [M]$, and $(s, a) \in \mathcal{S} \times \mathcal{A}$,*

$$\left| Q_p^\star(s, a) - Q_q^\star(s, a) \right| \le 2H\epsilon,$$

*consequently,* $\left| \mathrm{gap}_p(s, a) - \mathrm{gap}_q(s, a) \right| \le 4H\epsilon.$

*Proof.* For the first claim, we prove a stronger statement by backward induction on $h$, namely, for every $p, q \in [M]$, every $h \in [1, H + 1]$, and $(s, a) \in \mathcal{S}_h \times \mathcal{A}$,

$$\left| Q_p^\star(s, a) - Q_q^\star(s, a) \right| \le 2(H - h + 1)\epsilon.$$

**Base case:**  For $h = H + 1$, we have $Q_p^\star(s, a) = 0$ for every $(s, a) \in \mathcal{S}_h \times \mathcal{A}$, and $p \in [M]$. It follows trivially that $\left| Q_p^\star(s, a) - Q_q^\star(s, a) \right| = 0 \le 2(H - h + 1)\epsilon.$

**Inductive case:**  Suppose by inductive hypothesis that for some $h \in [1, H]$ and, for every $(s, a) \in \mathcal{S}_{h+1} \times \mathcal{A}$ and $p, q \in [M]$, $\left| Q_p^\star(s, a) - Q_q^\star(s, a) \right| \le 2(H - h)\epsilon.$

We first prove the following auxiliary statement: for every $s \in \mathcal{S}_{h+1}$ and $p, q \in [M]$,

$$\left| V_p^\star(s) - V_q^\star(s) \right| \le 2(H - h)\epsilon. \tag{6}$$

Let $a_p = \mathrm{argmax}_{a \in \mathcal{A}} Q_p^\star(s, a)$ and $a_q = \mathrm{argmax}_{a \in \mathcal{A}} Q_q^\star(s, a)$. The above auxiliary statement can be easily proven by contradiction: without loss of generality, suppose that $V_p^\star(s) - V_q^\star(s) = Q_p^\star(s, a_p) - Q_q^\star(s, a_q) > 2(H - h)\epsilon$. Since $Q_q^\star(s, a_p) \ge Q_p^\star(s, a_p) - 2(H - h)\epsilon$, it follows that $Q_q^\star(s, a_p) > Q_q^\star(s, a_q)$, which contradicts the fact that $a_q = \mathrm{argmax}_{a \in \mathcal{A}} Q_q^\star(s, a)$.

We now return to the inductive proof, and we show that given the inductive hypothesis, for every $(s, a) \in \mathcal{S}_h \times \mathcal{A}$ and $p, q \in [M]$,

$$\left| Q_p^\star(s, a) - Q_q^\star(s, a) \right|$$

$$\le \left| R_p(s, a) - R_q(s, a) \right| + \left| \sum_{s' \in \mathcal{S}_{h+1}} \left[ \mathbb{P}_p(s' \mid s, a) V_p^\star(s') - \mathbb{P}_q(s' \mid s, a) V_q^\star(s') \right] \right|$$

$$\le \epsilon + \left| \sum_{s' \in \mathcal{S}_{h+1}} \left[ \mathbb{P}_p(s' \mid s, a) V_p^\star(s') - \mathbb{P}_q(s' \mid s, a) V_p^\star(s') \right] \right| + \left| \sum_{s' \in \mathcal{S}_{h+1}} \mathbb{P}_q(s' \mid s, a) \left( V_p^\star(s') - V_q^\star(s') \right) \right|$$

$$\le \epsilon + \| \mathbb{P}_p(\cdot \mid s, a) - \mathbb{P}_q(\cdot \mid s, a)) \|_1 \left( \max_{s' \in \mathcal{S}_{h+1}} \left| V_p^\star(s') \right| \right) + \| \mathbb{P}_q(\cdot \mid s, a) \|_1 \left( \max_{s' \in \mathcal{S}_{h+1}} \left| V_p^\star(s') - V_q^\star(s') \right| \right)$$

$$\le \epsilon + \frac{\epsilon}{H} \cdot H + 2(H - h)\epsilon$$

$$= 2(H - h + 1)\epsilon,$$

where the first inequality follows from Eq. (1) and the triangle inequality; the second inequality follows from Definition 1 and the triangle inequality; the third inequality follows from Hölder's inequality; and the fourth inequality uses Definition 1 and Eq. (6).

For the second claim, we note that from the first claim, we have for any $p, q, s$,

$$\left| V_p^\star(s) - V_q^\star(s) \right| = \left| \max_{a \in \mathcal{A}} Q_p^\star(s, a) - \max_{a \in \mathcal{A}} Q_p^\star(s, a) \right| \le 2H\epsilon,$$

therefore, for any $p, q, s, a$,

$$\left| \mathrm{gap}_p(s, a) - \mathrm{gap}_q(s, a) \right| \le \left| V_p^\star(s) - V_q^\star(s) \right| + \left| Q_p^\star(s, a) - Q_p^\star(s, a) \right| \le 4H\epsilon. \qquad \square$$

## A.2 Proof of Lemma 4

**Lemma 4.** *For any $(s,a) \in \mathcal{I}_\epsilon$, we have that: (1) for all $p \in [M]$, $(s,a) \notin Z_{p,\mathrm{opt}}$, where we recall that $Z_{p,\mathrm{opt}} = \left\{ (s,a) : \mathrm{gap}_p(s,a) = 0 \right\}$ is the set of optimal state-action pairs with respect to p; (2) for all $p, q \in [M]$, $\mathrm{gap}_p(s,a) \geq \frac{1}{2}\mathrm{gap}_q(s,a)$.*

*Proof.* For any $(s,a) \in \mathcal{I}_\epsilon$, there exists some $p_0$ such that $\mathrm{gap}_{p_0}(s,a) \geq 96H\epsilon$. Therefore, for every $p \in [M]$,

$$\mathrm{gap}_p(s,a) \geq \mathrm{gap}_{p_0}(s,a),$$

From Lemma 2 we know that $\left|\mathrm{gap}_p(s,a) - \mathrm{gap}_{p_0}(s,a)\right| \leq 4H\epsilon$. Therefore, for all $p$,

$$\mathrm{gap}_p(s,a) \geq \mathrm{gap}_{p_0}(s,a) - 4H\epsilon \geq 92H\epsilon > 0.$$

This proves the first item.

For the second item, for all $p, q \in [M]$,

$$\frac{\mathrm{gap}_p(s,a)}{\mathrm{gap}_q(s,a)} = \frac{\mathrm{gap}_q(s,a) - 4H\epsilon}{\mathrm{gap}_q(s,a)} \geq 1 - \frac{4H\epsilon}{\mathrm{gap}_q(s,a)} \geq 1 - \frac{4}{92} \geq \frac{1}{2}. \qquad \square$$

## B   Additional Definitions Used in the Proofs

In this section, we define a few useful notations that will be used in our proofs. For state-action pair $(s,a) \in \mathcal{S} \times \mathcal{A}$, player $p \in [M]$, episode $k \in [K]$:

1. Define $n^k(s,a)$ (resp. $n_p^k(s,a)$, $\hat{\mathbb{P}}^k$, $\hat{\mathbb{P}}_p^k$, $\hat{R}^k$, $\hat{R}_p^k$) to be the value of $n(s,a)$ (resp. $n_p(s,a)$, $\hat{\mathbb{P}}$, $\hat{\mathbb{P}}_p$, $\hat{R}$, $\hat{R}_p$) at the *beginning* of episode $k$ of MULTI-TASK-EULER.

2. Denote by $\overline{Q}_p^k$ (resp. $\underline{Q}_p^k, \overline{V}_p^k, \underline{V}_p^k$, ind-$b_p^k(s,a)$, agg-$b_p^k(s,a)$) the values of $\overline{Q}_p$ (resp. $\underline{Q}_p, \overline{V}_p, \underline{V}_p$, ind-$b_p(s,a)$, agg-$b_p(s,a)$) right after MULTI-TASK-EULER finishes its optimistic value iteration (line 15) at episode $k$.

3. Define the *surplus* [36] (also known as the Bellman error) of $(s,a)$ at episode $k$ and player $p$ as:
$$E_p^k(s,a) := \overline{Q}_p^k(s,a) - R_p(s,a) - (\mathbb{P}_p \overline{V}_p^k)(s,a).$$

4. Define $w_p^k(s,a) := \frac{n_p^k(s,a)}{n^k(s,a)}$ be the proportion of player $p$ on $(s,a)$ at the beginning of episode $k$; this induces $(s,a)$'s *mixture expected reward*:
$$\bar{R}^k(s,a) := \sum_{q=1}^{M} w_q^k(s,a) R_q(s,a),$$
and *mixture transition probability*:
$$\bar{\mathbb{P}}^k(\cdot \mid s,a) := \sum_{q=1}^{M} w_q^k(s,a) \mathbb{P}_q(\cdot \mid s,a).$$

5. Define $\rho_p^k(s,a) := \mathbb{P}((s_h, a_h) = (s,a) \mid \pi^k(p), \mathcal{M}_p)$ to be the occupancy measure of $\pi^k(p)$ over $\mathcal{M}_p$ on $(s,a)$, where $h \in [H]$ is the layer $s$ is in (so that $s \in \mathcal{S}_h$). It can be seen that $\rho_p^k$, when restricted to $\mathcal{S}_h \times \mathcal{A}$, is a probability distribution on this set.

   Define $\rho^k(s,a) := \sum_{p=1}^{M} \rho_p^k(s,a)$; it can be seen that $\rho^k(s,a) \in [0, M]$. Define $\bar{n}_p^k(s,a) := \sum_{j=1}^{k} \rho_p^j(s,a)$, and $\bar{n}^k(s,a) := \sum_{j=1}^{k} \rho^j(s,a)$.[3]

---

[3]These are the cumulative occupancy measures up to episode $k$, inclusively; this is in contrast with the definition of $n^k(s,a)$ and $n_p^k(s,a)$, which do not count the trajectories observed at episode $k$.

6. Define $N^k(s) := \sum_{a \in \mathcal{A}} n^k(s, a)$ and $N_p^k(s) := \sum_{a \in \mathcal{A}} n_p^k(s, a)$ to be the total number of encounters of state $s$ by all players, and by player $p$ only, respectively, at the beginning of episode $k$.

7. Define $N_1 \asymp M \ln(\frac{SAK}{\delta})$, and $N_2 \asymp \ln(\frac{MSAK}{\delta})$; define $\tau(s, a) := \min \{ k : \bar{n}^k(s, a) \geq N_1 \}$, and $\tau_p(s, a) := \min \{ k : \bar{n}_p^k(s, a) \geq N_2 \}$. With high probability, so long as $k \geq \tau(s, a)$ (resp. $k \geq \tau_p(s, a)$), $n^k(s, a)$ and $\bar{n}^k(s, a)$ (resp. $n_p^k(s, a)$ and $\bar{n}_p^k(s, a)$) are within a constant factor of each other; see Lemma 11.

8. Define $\mathrm{g\breve{a}p}_p(s, a) := \frac{\mathrm{gap}_p(s,a)}{4H} \vee \frac{\mathrm{gap}_{p,\min}}{4H}$; recall the definitions of $\mathrm{gap}_p(s, a)$ and $\mathrm{gap}_{p,\min}$ in Section 2.

Define $\mathrm{Reg}(K, p) := \sum_{k=1}^{K} \left( V_{0,p}^{\star} - V_{0,p}^{\pi^k(p)} \right)$ as player $p$'s contribution to the collective regret; in this notation, $\mathrm{Reg}(K) = \sum_{p=1}^{M} \mathrm{Reg}(K, p)$.

Define the clipping function $\mathrm{clip}(\alpha, \Delta) := \alpha \mathbf{1}(\alpha \geq \Delta)$.

We also adopt the following conventions in our proofs:

1. As $\epsilon$-dissimilarity with $\epsilon > 2H$ does not impose any constraints on $\{\mathcal{M}_p\}_{p=1}^{M}$, throughout the proof, we only focus on the regime that $\epsilon \leq 2H$.

2. We will use $\pi^k(p)$ and $\pi_p^k$ interchangeably. To avoid notational clutter, we will also sometimes slightly abuse notation and use $V_{p,h}^{\pi^k}$, $V_p^{\pi^k}$ to denote $V_{p,h}^{\pi^k(p)}$, $V_p^{\pi^k(p)}$ respectively.

## C   Proof of the Upper Bounds

This section establishes the regret guarantees of MULTI-TASK-EULER (Theorems 5 and 6). The proof follows a similar outline as STRONG-EULER's analysis [36], with important modifications tailored to the multitask setting. The proof has the following structure:

1. Subsection C.1 defines a "clean" event $E$ that we show happens with probability $1 - \delta$. When $E$ happens, the observed samples are representative enough so that standard concentration inequalities apply. This will serve as the basis of our subsequent arguments.

2. Subsection C.2 shows that when $E$ happens, the value function upper and lower bounds are valid; furthermore, MULTI-TASK-EULER satisfies strong optimism [36], in that all players' surpluses are always nonnegative for all state-action pairs at all time steps.

3. Subsection C.3 establishes a distribution-dependent upper bound on MULTI-TASK-EULER's surpluses when $E$ happens, which is key to our regret theorems. In comparison with STRONG-EULER [36] in the single task setting, MULTI-TASK-EULER exploits inter-task similarity, so that its surpluses on state-action pair $(s, a)$ for player $p$ are further controlled by a new term that depends on the dissimilarity parameter $\epsilon$, along with $n^k(s, a)$, the total visitation counts of $(s, a)$ *by all players*.

4. Subsection C.4 uses the strong optimism property and the surplus bounds established in the previous two subsections to conclude our final gap-independent and gap-dependent regret guarantees, via the clipping lemma of [36] (see also Lemma 20).

5. Finally, Subsection C.5 collects miscellaneous technical lemmas used in the proofs.

### C.1   A clean event

Below we define a "clean" event $E$ in which all concentration bounds used in the analysis hold, which we will show happens with high probability. Specifically, we will define $E = E_{\mathrm{ind}} \cap E_{\mathrm{agg}} \cap E_{\mathrm{sample}}$, where $E_{\mathrm{ind}}, E_{\mathrm{agg}}, E_{\mathrm{sample}}$ are defined respectively below.

In subsequent definitions of events, we will abbreviate $\forall k \in [K], h \in [H], p \in [M], s \in \mathcal{S}_h, a \in \mathcal{A}, s' \in \mathcal{S}_{h+1}$ as $\forall k, h, p, s, a, s'$. Also, recall that $L(n) \asymp \ln(\frac{MSAn}{\delta})$.

Define event $E_{\text{ind}}$ as:

$$E_{\text{ind}} = E_{\text{ind,rw}} \cap E_{\text{ind,val}} \cap E_{\text{ind,prob}} \cap E_{\text{ind,var}}, \tag{7}$$

$$E_{\text{ind,rw}} = \left\{ \forall k, h, p, s, a \, . \, \left| \hat{R}_p^k(s,a) - R_p(s,a) \right| \leq \sqrt{\frac{L(n^k(s,a))}{2n^k(s,a)}} \right\}, \tag{8}$$

$$E_{\text{ind,val}} = \left\{ \forall k, h, p, s, a \, . \, \left| (\hat{\mathbb{P}}_p^k V_p^\star - \mathbb{P}_p V_p^\star)(s,a) \right| \leq 4\sqrt{\frac{\text{var}_{\mathbb{P}_p(\cdot|s,a)}[V_p^\star]L(n_p^k(s,a))}{n_p^k(s,a)}} + \frac{2HL(n_p^k(s,a))}{n_p^k(s,a)} \right\}, \tag{9}$$

$$E_{\text{ind,prob}} = \left\{ \forall k, h, p, s, a, s' \, . \, \left| (\hat{\mathbb{P}}_p^k - \mathbb{P}_p)(s' \mid s,a) \right| \leq 4\sqrt{\frac{L(n_p^k(s,a)) \cdot \mathbb{P}_p(s' \mid s,a)}{n_p^k(s,a)}} + \frac{2L(n_p^k(s,a))}{n_p^k(s,a)} \right\}, \tag{10}$$

$$E_{\text{ind,var}} = \left\{ \forall k, h, p, s, a \, . \, \left| \frac{1}{n_p^k(s,a)} \sum_{i=1}^{n_p^k(s,a)} (V_p^\star(s_i') - (\mathbb{P}_p V_p^\star)(s,a))^2 - \text{var}_{\mathbb{P}_p(\cdot|s,a)}[V_p^\star] \right|, \tag{11} \right.$$

$$\left. \leq 4\sqrt{\frac{H^2\text{var}_{\mathbb{P}_p(\cdot|s,a)}[V_p^\star]L(n_p^k(s,a))}{n_p^k(s,a)}} + \frac{2H^2L(n_p^k(s,a))}{n_p^k(s,a)} \right\},$$

where in Equation (11), $s_i'$ denotes the next state player $p$ transitions to, for the $i$-th episode it experiences $(s,a)$. $E_{\text{ind}}$ captures the concentration behavior of each player's individual model estimates.

**Lemma 9.** $\mathbb{P}(E_{\text{ind}}) \geq 1 - \frac{\delta}{3}$.

*Proof.* The proof follows a similar reasoning as the proof of e.g., [36, Proposition F.9] using Freedman's Inequality. We would like to show that each of $E_{\text{ind,rw}}, E_{\text{ind,val}}, E_{\text{ind,prob}}, E_{\text{ind,var}}$ happens with probability $1 - \frac{\delta}{12}$, which would give the lemma statement by a union bound. For brevity, we only show that $\mathbb{P}(E_{\text{ind,var}}) \geq 1 - \frac{\delta}{12}$, and the other probability statements follow from a similar reasoning.

Fix $h \in [H]$, $(s,a) \in \mathcal{S}_h \times \mathcal{A}$, and $p \in [M]$. We will show

$$\mathbb{P}\left( \exists k \in [K] . \left| \frac{1}{n_p^k(s,a)} \sum_{i=1}^{n_p^k(s,a)} (V_p^\star(s_i') - (\mathbb{P}_p V_p^\star)(s,a))^2 - \text{var}_{\mathbb{P}_p(\cdot|s,a)}[V_p^\star] \right| \right.$$

$$\left. \geq 4\sqrt{\frac{H^2\text{var}_{\mathbb{P}_p(\cdot|s,a)}[V_p^\star]L(n_p^k(s,a))}{n_p^k(s,a)}} + \frac{2H^2L(n_p^k(s,a))}{n_p^k(s,a)} \right) \leq \frac{\delta}{12MSA}. \tag{12}$$

For every $j \in \mathbb{N}_+$, define stopping time $k_j$ as the $j$-th episode when $(s,a)$ is experienced by player $p$, if such episode exists; otherwise, $k_j$ is defined as $\infty$. it suffices to show that

$$\mathbb{P}\left( \exists j \in \mathbb{N}_+ . \; k_j < \infty \wedge \left| \frac{1}{j} \sum_{i=1}^{j} (V_p^\star(s_i') - (\mathbb{P}_p V_p^\star)(s,a))^2 - \text{var}_{\mathbb{P}_p(\cdot|s,a)}[V_p^\star] \right| \right.$$

$$\left. \geq 4\sqrt{\frac{H^2\text{var}_{\mathbb{P}_p(\cdot|s,a)}[V_p^\star]L(j)}{j}} + \frac{2H^2L(j)}{j} \right) \leq \frac{\delta}{12MSA}. \tag{13}$$

Define $\mathcal{G}_j$ as the $\sigma$-algebra generated by all observations up to time step $k_j$. We have that $\{\mathcal{G}_j\}_{j=0}^{\infty}$ is a filtration. It can be seen that the sequence $\left\{X_j := (V_p^\star(s_j') - (\mathbb{P}_p V_p^\star)(s,a))^2 - \mathrm{var}_{\mathbb{P}_p(\cdot|s,a)}[V_p^\star]\right\}_{j=1}^{\infty}$ is a martingale difference sequence adapted to $\{\mathcal{G}_j\}_{j=0}^{\infty}$; in addition, for every $j$, $|X_j| \leq H^2$, and $\mathbb{E}\left[X_j^2 \mid \mathcal{G}_{j-1}\right] \leq \mathbb{E}\left[(V_p^\star(s_j') - (\mathbb{P}_p V_p^\star)(s,a))^4 \mid \mathcal{G}_{j-1}\right] \leq H^2 \mathrm{var}_{\mathbb{P}_p(\cdot|s,a)}[V_p^\star]$. This implies that for any $\lambda \geq 0$,

$$\left\{Y_j(\lambda) = \exp\left(\lambda \frac{1}{H^2}(\sum_{i=1}^{j} X_i) - \left((e^\lambda - \lambda - 1)\frac{j}{H^2}\mathrm{var}_{\mathbb{P}_p(\cdot|s,a)}[V_p^\star]\right)\right)\right\}_{j=0}^{\infty}$$

is a nonnegative supermartingale [14], and by optional sampling theorem, $\mathbb{E}\left[Y_j(\lambda)\mathbf{1}(k_j < \infty)\right] \leq \mathbb{E}\left[Y_0(\lambda)\right] = 1$. As a result, for any fixed thresholds $a, v \geq 0$ [see 14, Theorem 1.6],

$$\mathbb{P}\left(\sum_{i=1}^{j} X_i \geq a \wedge \sum_{i=1}^{j} H^2 \mathrm{var}_{\mathbb{P}_p(\cdot|s,a)}[V_p^\star] \leq v \wedge k_j < \infty\right) \leq \exp\left(-\frac{a^2}{2v + 2aH^2/3}\right)$$

Now, by the doubling argument of [4, Lemma 2] (observe that $\sum_{i=1}^{j} \mathbb{E}\left[X_i^2 \mid \mathcal{G}_{i-1}\right] \in [0, H^4 j]$), we have that for all $j \in \mathbb{N}_+$:

$$\mathbb{P}\left(k_j < \infty \wedge \left|\frac{1}{j}\sum_{i=1}^{j}(V_p^\star(s_i') - (\mathbb{P}_p V_p^\star)(s,a))^2 - \mathrm{var}_{\mathbb{P}_p(\cdot|s,a)}[V_p^\star]\right| \right.$$
$$\left. \geq 4\sqrt{\frac{H^2 \mathrm{var}_{\mathbb{P}_p(\cdot|s,a)}[V_p^\star] L(j)}{j} + \frac{2H^2 L(j)}{j}}\right) \leq \ln(4j) \cdot \frac{\delta}{48j^2 MSA}.$$

A union bound over all $j \in \mathbb{N}_+$ yields Equation (13). $\qquad\square$

Define event $E_{\mathrm{agg}}$ as:

$$E_{\mathrm{agg}} = E_{\mathrm{agg,rw}} \cap E_{\mathrm{agg,val}} \cap E_{\mathrm{agg,prob}} \cap E_{\mathrm{agg,var}}, \tag{14}$$

$$E_{\mathrm{agg,rw}} = \left\{\forall k, h, p, s, a \centerdot \left|\hat{R}^k(s,a) - \bar{R}^k(s,a)\right| \leq \sqrt{\frac{L(n^k(s,a))}{2n^k(s,a)}}\right\}, \tag{15}$$

$$E_{\mathrm{agg,val}} = \left\{\forall k, h, p, s, a \centerdot \left|(\hat{\mathbb{P}}^k V_p^\star - \bar{\mathbb{P}}^k V_p^\star)(s,a)\right|, \right. \tag{16}$$

$$\left. \leq 4\sqrt{\frac{\left(\sum_{q=1}^{M} w_q^k(s,a)\mathrm{var}_{\mathbb{P}_q(\cdot|s,a)}[V_p^\star]\right) L(n^k(s,a))}{n^k(s,a)} + \frac{2HL(n^k(s,a))}{n^k(s,a)}}\right\}, \tag{17}$$

$$E_{\mathrm{agg,prob}} = \left\{\forall k, h, p, s, a, s' \centerdot \left|(\hat{\mathbb{P}}^k - \bar{\mathbb{P}}^k)(s' \mid s,a)\right| \leq 4\sqrt{\frac{\bar{\mathbb{P}}^k(s' \mid s,a) \cdot L(n^k(s,a))}{n^k(s,a)}} + \frac{2L(n^k(s,a))}{n^k(s,a)}\right\}, \tag{18}$$

$$E_{\mathrm{agg,var}} = \left\{\forall k, h, p, s, a \centerdot \left|\frac{1}{n^k(s,a)}\sum_{i=1}^{n^k(s,a)}(V_p^\star(s_i') - (\mathbb{P}_{p_i} V_q^\star)(s,a))^2 - \sum_{q=1}^{M} w_q^k(s,a)\mathrm{var}_{\mathbb{P}_q(\cdot|s,a)}[V_p^\star]\right|, \right. \tag{19}$$

$$\left. \leq 4\sqrt{\frac{H^2\left(\sum_{q=1}^{M} w_q^k(s,a)\mathrm{var}_{\mathbb{P}_q(\cdot|s,a)}[V_p^\star]\right) L(n^k(s,a))}{n^k(s,a)} + \frac{2H^2 L(n^k(s,a))}{n^k(s,a)}}\right\},$$

where in Equation (19), $s'_i$ denotes the next state for the $i$-th time some player experiences $(s, a)$. $E_{\text{agg}}$ captures the concentration behavior of the aggregate model estimates.

**Lemma 10.** $\mathbb{P}(E_{\text{agg}}) \geq 1 - \frac{\delta}{3}$.

*Proof.* The proof follows a similar reasoning as the proof of e.g., [36, Proposition F.9] using Freedman's Inequality. We would like to show that each of $E_{\text{agg,rw}}, E_{\text{agg,val}}, E_{\text{agg,prob}}, E_{\text{agg,var}}$ happen with probability $1 - \frac{\delta}{12}$, which would give the lemma statement by a union bound. For brevity, we show that $\mathbb{P}(E_{\text{agg,var}}) \geq 1 - \frac{\delta}{12}$, and the other probability statements follow from a similar reasoning.

Fix $h \in [H]$, $(s, a) \in \mathcal{S}_h \times \mathcal{A}$ and $p \in [M]$; denote by $p_i$ the identity of the player when $(s, a)$ is experienced for the $i$-th time for some player. It suffices to show that

$$
\mathbb{P}\left( \exists k \in [K]. \left| \frac{1}{n^k(s, a)} \sum_{i=1}^{n^k(s,a)} \left( (V_p^\star(s'_i) - (\mathbb{P}_{p_i} V_p^\star)(s, a))^2 - \text{var}_{\mathbb{P}_{p_i}(\cdot|s,a)}[V_p^\star] \right) \right| \right.
$$
$$
\left. \geq 4 \sqrt{ \frac{H^2 \left( \sum_{i=1}^{n^k(s,a)} \text{var}_{\mathbb{P}_{p_i}(\cdot|s,a)}[V_p^\star] \right) L(n^k(s,a))}{(n^k(s,a))^2} } + \frac{2H^2 L(n^k(s,a))}{n^k(s,a)} \right) \leq \frac{\delta}{12MSA},
$$
(20)

because $\frac{1}{n^k(s,a)} \sum_{i=1}^{n^k(s,a)} \text{var}_{\mathbb{P}_{p_i}(\cdot|s,a)}[V_p^\star] = \sum_{q=1}^M w_q^k(s, a) \text{var}_{\mathbb{P}_q(\cdot|s,a)}[V_p^\star]$.

For episode $k$ and player index $p$, denote its corresponding *micro-episode* index as $(k-1)M + p$. For every $j \in \mathbb{N}_+$, define stopping time $k_j$ as follows: it is the index of the $j$-th micro-episode when $(s, a)$ is experienced by some player, if such micro-episode exists; and $k_j$ is defined to be $\infty$ otherwise. With this notation, it suffices to show:

$$
\mathbb{P}\left( \exists j \in \mathbb{N}_+. \; k_j < \infty \wedge \left| \frac{1}{j} \sum_{i=1}^{j} \left( (V_p^\star(s'_i) - (\mathbb{P}_{p_i} V_p^\star)(s, a))^2 - \text{var}_{\mathbb{P}_{p_i}(\cdot|s,a)}[V_p^\star] \right) \right| \right.
$$
$$
\left. \geq 4 \sqrt{ \frac{H^2 \left( \sum_{i=1}^{j} \text{var}_{\mathbb{P}_{p_i}(\cdot|s,a)}[V_p^\star] \right) L(j)}{j^2} } + \frac{2H^2 L(j)}{j} \right) \leq \frac{\delta}{12MSA},
$$
(21)

Define $\mathcal{G}_j$ as the $\sigma$-algebra generated by all observations up to micro-episode $k_j$. We have that $\{\mathcal{G}_j\}_{j=0}^{\infty}$ is a filtration. It can be seen that $\left\{ X_j := (V_p^\star(s'_j) - (\mathbb{P}_{p_j} V_p^\star)(s, a))^2 - \text{var}_{\mathbb{P}_{p_j}(\cdot|s,a)}[V_p^\star] \right\}_{j=1}^{\infty}$ is a martingale difference sequence adapted to $\{\mathcal{G}_j\}_{j=0}^{\infty}$; in addition, for every $j$, $|X_j| \leq H^2$, and $\mathbb{E}\left[ X_j^2 \mid \mathcal{G}_{j-1} \right] \leq \mathbb{E}\left[ (V_p^\star(s'_j) - (\mathbb{P}_{p_j} V_p^\star)(s, a))^4 \mid \mathcal{G}_{j-1} \right] \leq H^2 \text{var}_{\mathbb{P}_{p_j}(\cdot|s,a)}[V_p^\star]$. Using the same reasoning as in the proof of Lemma 9 (and observing that $\sum_{i=1}^{j} \mathbb{E}\left[ X_i^2 \mid \mathcal{G}_{i-1} \right] \in [0, H^4 j]$), we have that for all $j \in \mathbb{N}_+$:

$$
\mathbb{P}\left( k_j < \infty \wedge \left| \frac{1}{j} \sum_{i=1}^{j} \left( (V_p^\star(s'_i) - (\mathbb{P}_{p_i} V_p^\star)(s, a))^2 - \text{var}_{\mathbb{P}_p(\cdot|s,a)}[V_p^\star] \right) \right| \right.
$$
$$
\left. \geq 4 \sqrt{ \frac{H \sum_{i=1}^{j} \text{var}_{\mathbb{P}_{p_i}(\cdot|s,a)}[V_p^\star] L(j)}{j^2} } + \frac{2H^2 L(j)}{j} \right) \leq \ln(4j) \cdot \frac{\delta}{48j^2 MSA}.
$$

A union bound over all $j \in \mathbb{N}_+$ implies that Equation (21) holds. $\square$

Define

$$E_{\text{sample}} = E_{\text{ind,sample}} \cap E_{\text{agg,sample}},$$

$$E_{\text{agg,sample}} = \left\{ \forall s, a, k \centerdot \bar{n}^k(s,a) \geq N_1 \implies n^k(s,a) \geq \frac{1}{2}\bar{n}^k(s,a) \right\},$$

$$E_{\text{ind,sample}} = \left\{ \forall s, a, k, p \centerdot \bar{n}_p^k(s,a) \geq N_2 \implies n_p^k(s,a) \geq \frac{1}{2}\bar{n}_p^k(s,a) \right\},$$

where we recall from Section B that $N_1 \asymp M \ln(\frac{SAK}{\delta})$, and $N_2 \asymp \ln(\frac{MSAK}{\delta})$.

**Lemma 11.** $\mathbb{P}(E_{\text{sample}}) \geq 1 - \frac{\delta}{3}$.

*Proof.* We first show $\mathbb{P}(E_{\text{agg,sample}}) \geq 1 - \frac{\delta}{6}$. Specifically, fix $h \in [H]$ and $(s,a) \in \mathcal{S}_h \times \mathcal{A}$, define random variable $X_k = \sum_{p=1}^{M} \left( \mathbf{1}\left( (s_{h,p}^k, a_{h,p}^k) = (s,a) \right) - \rho_p^k(s,a) \right)$. Also, define $\mathcal{G}_k$ as the $\sigma$-algebra generated by all observations up to episode $k$. It can be readily seen that $\{X_k\}_{k=1}^{K}$ is a martingale difference sequence adapted to filtration $\{\mathcal{G}_k\}_{k=0}^{K}$. Freedman's inequality (specifically, Lemma 2 of [4]) implies that for every fixed $k$, with probability $1 - \frac{\delta}{6K}$,

$$\left| n^k(s,a) - \bar{n}^{k-1}(s,a) \right| \leq 4\sqrt{\bar{n}^{k-1}(s,a) \cdot M \ln\left( \frac{6SAK^2}{\delta} \right)} + 4M \ln\left( \frac{6SAK^2}{\delta} \right), \tag{22}$$

If Equation (22) happens, then by AM-GM inequality that $\sqrt{\bar{n}^{k-1}(s,a) \cdot M \ln\left( \frac{6SAK^2}{\delta} \right)} \leq \frac{1}{4}\bar{n}^{k-1}(s,a) + 16M \ln\left( \frac{6SAK^2}{\delta} \right)$, we have

$$\bar{n}^{k-1}(s,a) - n^k(s,a) \leq \frac{1}{4}\bar{n}^{k-1}(s,a) + 20M \ln\left( \frac{6SAK^2}{\delta} \right),$$

implying that

$$n^k(s,a) \geq \frac{3}{4}\bar{n}^{k-1}(s,a) - 20M \ln\left( \frac{6SAK^2}{\delta} \right).$$

Additionally, as $\bar{n}^{k-1}(s,a) \geq \bar{n}^k(s,a) - M$ always holds, we have

$$n^k(s,a) \geq \frac{3}{4}\bar{n}^k(s,a) - 21M \ln\left( \frac{6SAK^2}{\delta} \right).$$

In summary, for any fixed $k$, with probability $1 - \frac{\delta}{6K}$, if $\bar{n}^k(s,a) \geq N_1 := 84M \ln\left( \frac{6SAK^2}{\delta} \right)$,

$$n^k(s,a) \geq \frac{1}{2}\bar{n}^k(s,a).$$

Taking a union bound over all $k \in [K]$, we have $\mathbb{P}(E_{\text{agg,sample}}) \geq 1 - \frac{\delta}{6}$.

It follows similarly that $\mathbb{P}(E_{\text{ind,sample}}) \geq 1 - \frac{\delta}{6}$; the only difference in the proof is that, we need to take an extra union bound over all $p \in [M]$ - hence an additional factor of $M$ within $\ln(\cdot)$ in the definition of $N_2$. The lemma statement follows from a union bound over these two statements. $\square$

**Lemma 12.** $\mathbb{P}(E) \geq 1 - \delta$.

*Proof.* Follows from Lemmas 9, 10, and 11, along with a union bound. $\square$

## C.2 Validity of value function bounds

In this section, we show that if the clean event $E$ happens, then for all $k$ and $p$, the value function estimates $\overline{Q}_p^k, \underline{Q}_p^k, \overline{V}_p^k, \underline{V}_p^k$ are valid upper and lower bounds of the optimal value functions $Q_p^\star, V_p^\star$ (Lemma 15). As a by-product, we also give a general bound on the surplus (Lemma 14) which will be refined and used in the subsequent regret bound calculations. Before going into the proof of the above two lemmas, we need a technical lemma below (Lemma 13) that gives necessary concentration results which motivate the bonus constructions; its proof can be found at Section C.2.1.

**Lemma 13.** *Fix $p \in [M]$. Suppose $E$ happens, and suppose that for episode $k$ and step $h$, we have that for all $s' \in \mathcal{S}_{h+1}, \underline{V}_p^k(s') \le V^\star(s') \le \overline{V}_p^k(s')$. Then, for all $(s, a) \in \mathcal{S}_h \times \mathcal{A}$:*

*1.*

$$\left| \hat{R}_p^k(s, a) - R_p(s, a) \right| \le b_{\mathrm{rw}} \left( n_p^k(s, a), 0 \right), \tag{23}$$

$$\left| \hat{R}^k(s, a) - R_p(s, a) \right| \le b_{\mathrm{rw}} \left( n^k(s, a), \epsilon \right). \tag{24}$$

*2.*

$$\left| (\hat{\mathbb{P}}_p^k - \mathbb{P}_p)(V_p^\star)(s, a) \right| \le b_{\mathrm{prob}} \left( \hat{\mathbb{P}}_p^k(\cdot \mid s, a), n_p^k(s, a), \overline{V}_p^k, \underline{V}_p^k, 0 \right), \tag{25}$$

$$\left| (\hat{\mathbb{P}}^k - \mathbb{P}_p)(V_p^\star)(s, a) \right| \le b_{\mathrm{prob}} \left( \hat{\mathbb{P}}^k(\cdot \mid s, a), n^k(s, a), \overline{V}_p^k, \underline{V}_p^k, \epsilon \right). \tag{26}$$

*3. For any $V_1, V_2 : \mathcal{S}_{h+1} \to \mathbb{R}$ such that $\overline{V}_p^k \le V_1 \le V_2 \le \underline{V}_p^k$,*

$$\left| (\hat{\mathbb{P}}_p^k - \mathbb{P}_p)(V_2 - V_1)(s, a) \right| \le b_{\mathrm{str}} \left( \hat{\mathbb{P}}_p^k(\cdot \mid s, a), n_p^k(s, a), \overline{V}_p^k, \underline{V}_p^k, 0 \right), \tag{27}$$

$$\left| (\hat{\mathbb{P}}^k - \mathbb{P}_p)(V_2 - V_1)(s, a) \right| \le b_{\mathrm{str}} \left( \hat{\mathbb{P}}^k(\cdot \mid s, a), n^k(s, a), \overline{V}_p^k, \underline{V}_p^k, \epsilon \right). \tag{28}$$

**Lemma 14.** *If event $E$ happens, and suppose that for episode $k$ and step $h$, we have that for all $s' \in \mathcal{S}_{h+1}, \underline{V}_p^k(s') \le V_p^\star(s') \le \overline{V}_p^k(s')$. Then, for $(s, a) \in \mathcal{S}_h \times \mathcal{A}$,*

$$\overline{Q}_p^k(s, a) - \left( R_p(s, a) + (\mathbb{P}_p \overline{V}_p^k)(s, a) \right) \in \left[ 0, (H - h + 1) \wedge 2\textit{ind-}b_p^k(s, a) \wedge 2\textit{agg-}b_p^k(s, a) \right], \tag{29}$$

*and*

$$\left( R_p(s, a) + (\mathbb{P}_p \underline{V}_p^k)(s, a) \right) - \underline{Q}_p^k(s, a) \in \left[ 0, (H - h + 1) \wedge 2\textit{ind-}b_p^k(s, a) \wedge 2\textit{agg-}b_p^k(s, a) \right], \tag{30}$$

*where we recall that*

$$\textit{ind-}b_p^k(s, a) = b_{\mathrm{rw}} \left( n_p^k(s, a), 0 \right) + b_{\mathrm{prob}} \left( \hat{\mathbb{P}}_p^k(\cdot \mid s, a), n_p^k(s, a), \overline{V}_p^k, \underline{V}_p^k, 0 \right) + b_{\mathrm{str}} \left( \hat{\mathbb{P}}_p^k(\cdot \mid s, a), n_p^k(s, a), \overline{V}_p^k, \underline{V}_p^k, 0 \right),$$

$$\textit{agg-}b_p^k(s, a) = b_{\mathrm{rw}} \left( n^k(s, a), \epsilon \right) + b_{\mathrm{prob}} \left( \hat{\mathbb{P}}^k(\cdot \mid s, a), n^k(s, a), \overline{V}_p^k, \underline{V}_p^k, \epsilon \right) + b_{\mathrm{str}} \left( \hat{\mathbb{P}}^k(\cdot \mid s, a), n^k(s, a), \overline{V}_p^k, \underline{V}_p^k, \epsilon \right).$$

*Proof.* We only show Equation (29) for brevity; Equation (30) follows from an exact symmetrical reasoning.

Recall that $\overline{Q}_p^k(s, a) = \min \left( \overline{\mathrm{ind}\text{-}Q}_p^k(s, a), \overline{\mathrm{agg}\text{-}Q}_p^k(s, a), H \right)$. We compare each term in the $\min(\cdot)$ operator with $(R_p(s, a) + (\mathbb{P}_p \overline{V}_p^k)(s, a))$:

- For $\overline{\mathrm{ind}\text{-}Q}_p^k(s, a)$, using Lemma 13 and our assumption on $\overline{V}_p^k$ and $\underline{V}_p^k$ on $\mathcal{S}_{h+1}$, we have:

$$\overline{\mathrm{ind}\text{-}Q}_p^k(s, a) - \left( R_p(s, a) + (\mathbb{P}_p \overline{V}_p^k)(s, a) \right)$$

$$= (\hat{R}_p^k - R_p)(s, a) + b_{\mathrm{rw}} \left( n_p^k(s, a), 0 \right)$$

$$+ ((\hat{\mathbb{P}}_p^k - \mathbb{P}_p)V_p^\star)(s, a) + b_{\mathrm{prob}} \left( \hat{\mathbb{P}}_p^k(\cdot \mid s, a), n_p^k(s, a), \overline{V}_p^k, \underline{V}_p^k, 0 \right)$$

$$+ (\hat{\mathbb{P}}_p^k - \mathbb{P}_p)(\overline{V}_p^k - V_p^\star))(s, a) + b_{\mathrm{str}} \left( \hat{\mathbb{P}}_p^k(\cdot \mid s, a), n_p^k(s, a), \overline{V}_p^k, \underline{V}_p^k, 0 \right)$$

$$\in [0, 2\mathrm{ind}\text{-}b_p^k(s, a)].$$

- For $\overline{\text{agg-}Q}_p^k(s,a)$, using Lemma 13 and our assumptions on $\overline{V}_p^k$ and $\underline{V}_p^k$ over $\mathcal{S}_{h+1}$, we have:

$$\overline{\text{agg-}Q}_p^k(s,a) - \left( R_p(s,a) + (\mathbb{P}_p \overline{V}_p^k)(s,a) \right)$$

$$= (\hat{R}_p^k - R_p)(s,a) + b_{\text{rw}}\left( n^k(s,a), \epsilon \right)$$

$$+ ((\hat{\mathbb{P}}^k - \mathbb{P}_p)V_p^\star)(s,a) + b_{\text{prob}}\left( \hat{\mathbb{P}}^k(\cdot \mid s,a), n^k(s,a), \overline{V}_p^k, \underline{V}_p^k, \epsilon \right)$$

$$+ ((\hat{\mathbb{P}}^k - \mathbb{P}_p)(\overline{V}_p^k - V_p^\star))(s,a) + b_{\text{str}}\left( \hat{\mathbb{P}}^k(\cdot \mid s,a), n^k(s,a), \overline{V}_p^k, \underline{V}_p^k, \epsilon \right)$$

$$\in [0, 2\text{agg-}b_p^k(s,a)],$$

- For $H - h + 1$, we have:

$$(H - h + 1) - (R_p(s,a) + (\mathbb{P}_p \overline{V}_p^k)(s,a)) \in [0, H - h + 1],$$

where we use the observation that $R(s,a) \in [0,1]$, and $(\mathbb{P}_p \overline{V}_p^k)(s,a) \in [0, H-h]$, and their sum is in $[0, H]$.

Combining the above three establishes that

$$\overline{Q}_p^k(s,a) - (R(s,a) + (\mathbb{P}_p \overline{V}_p^k)(s,a)) \in \left[ 0, (H-h+1) \wedge 2\text{ind-}b_p^k(s,a) \wedge 2\text{agg-}b_p^k(s,a) \right]. \quad \square$$

**Lemma 15.** *Under event $E$, for every $k \in [K]$, and every $p \in [M]$, and for every $h \in [H]$, For all $(s,a) \in \mathcal{S}_h \times \mathcal{A}$,*

$$\underline{Q}_p^k(s,a) \leq Q_p^{\pi^k}(s,a) \leq Q_p^\star(s,a) \leq \overline{Q}_p^k(s,a), \tag{31}$$

*and*

$$\underline{V}_p^k(s) \leq V_p^{\pi^k}(s) \leq V_p^\star(s) \leq \overline{V}_p^k(s), \tag{32}$$

*Proof.* The proof of this lemma extends [36, Proposition F.1] to our multitask setting.

For every $k$ and $p$, we show the above holds for all layers $h \in [H]$ and every $(s,a) \in \mathcal{S}_h \times \mathcal{A}$; to this end, we do backward induction on layer $h$.

**Base case:** For layer $h = H + 1$, we have $\underline{V}_p^k(\bot) = V_p^{\pi^k}(\bot) = V_p^\star(\bot) = \overline{V}_p^k(\bot) = 0$.

**Inductive case:** By our inductive hypothesis, for layer $h + 1$ and every $s \in \mathcal{S}_{h+1}$,

$$\underline{V}_p^k(s) \leq V_p^{\pi^k}(s) \leq V_p^\star(s) \leq \overline{V}_p^k(s).$$

We will show that Equations (31) and (32) holds holds for all $(s,a) \in \mathcal{S}_h \times \mathcal{A}$.

We first show Equation (31). First, $Q_p^{\pi^k}(s,a) \leq Q_p^\star(s,a)$ for all $(s,a) \in \mathcal{S}_h \times \mathcal{A}$ is trivial.

To show $Q_p^\star(s,a) \leq \overline{Q}_p^k(s,a)$ for all $(s,a) \in \mathcal{S}_h \times \mathcal{A}$, by Lemma 14 and inductive hypothesis, we have:

$$Q_p^\star(s,a) = R_p(s,a) + (\mathbb{P}_p V_p^\star)(s,a) \leq R_p(s,a) + (\mathbb{P}_p \overline{V}_p^k)(s,a) \leq \overline{Q}_p^k(s,a).$$

Likewise, we show $Q_p^{\pi^k}(s,a) \geq \underline{Q}_p^k(s,a)$ for all $(s,a) \in \mathcal{S}_h \times \mathcal{A}$, using Lemma 14 and inductive hypothesis:

$$Q_p^{\pi^k}(s,a) = R_p(s,a) + (\mathbb{P}_p V_p^{\pi^k})(s,a) \geq R_p(s,a) + (\mathbb{P}_p \overline{V}_p^k)(s,a) \geq \underline{Q}_p^k(s,a).$$

This completes the proof of Equation (31) for layer $h$.

We now show Equation (32) for layer $h$. Again $V_p^{\pi^k}(s) \leq V_p^\star(s)$ for all $s \in \mathcal{S}_h$ is trivial.

To show $V_p^\star(s) \le \overline{V}_p^k(s)$ for all $s \in \mathcal{S}_h$, observe that

$$V_p^\star(s) = \max_{a \in \mathcal{A}} Q_p^\star(s,a) \le \max_{a \in \mathcal{A}} \overline{Q}_p^k(s,a) = \overline{V}_p^k(s).$$

To show $V_p^{\pi^k}(s) \ge \underline{V}_p^k(s)$ for all $s \in \mathcal{S}_h$, observe that

$$V_p^{\pi^k}(s) = Q_p^{\pi^k}(s, \pi^k(p)(s)) \ge \underline{Q}_p^k(s, \pi^k(p)(s)) = \underline{V}_p^k(s).$$

This completes the induction. $\qquad\square$

### C.2.1 Proof of Lemma 13

*Proof of Lemma 13.* Equations (23), (25), and (27) essentially follow the same reasoning as in [36]; we still include their proofs for completeness. Equations (24), (26), and (28) are new, and require a more involved analysis. Our proof also relies on a technical lemma, namely Lemma 16; we defer its statement and proof to the end of this subsection.

1. Equation (23) follows directly from the definition of $E_{\mathrm{ind,rw}}$. Equation (24) follows from the definition of $E_{\mathrm{agg,rw}}$, and the fact that $\left| \bar{R}^k(s,a) - R_p(s,a) \right| \le \epsilon$.

2. We prove Equation (25) as follows:

$$\left| (\hat{\mathbb{P}}_p^k V^\star - \mathbb{P}_p V_p^\star)(s,a) \right|$$

$$\le O\left( \sqrt{\frac{\mathrm{var}_{\mathbb{P}_p(\cdot|s,a)}[V^\star] L(n_p^k(s,a))}{n_p^k(s,a)}} + \frac{HL(n_p^k(s,a))}{n_p^k(s,a)} \right)$$

$$\le O\left( \sqrt{\frac{\mathrm{var}_{\hat{\mathbb{P}}_p^k(\cdot|s,a)}[V^\star] L(n_p^k(s,a))}{n_p^k(s,a)}} + \frac{HL(n_p^k(s,a))}{n_p^k(s,a)} \right)$$

$$\le O\left( \sqrt{\frac{\mathrm{var}_{\hat{\mathbb{P}}_p^k(\cdot|s,a)}[\overline{V}_p^k] L(n_p^k(s,a))}{n_p^k(s,a)}} + \sqrt{\frac{\|V_p^\star - \overline{V}_p^k\|_{\hat{\mathbb{P}}_p^k(\cdot|s,a)}^2 L(n_p^k(s,a))}{n_p^k(s,a)}} + \frac{HL(n_p^k(s,a))}{n_p^k(s,a)} \right)$$

$$\le O\left( \sqrt{\frac{\mathrm{var}_{\hat{\mathbb{P}}_p^k(\cdot|s,a)}[\overline{V}_p^k] L(n_p^k(s,a))}{n_p^k(s,a)}} + \sqrt{\frac{\|\overline{V}_p^k - \underline{V}_p^k\|_{\hat{\mathbb{P}}_p^k(\cdot|s,a)}^2 L(n_p^k(s,a))}{n_p^k(s,a)}} + \frac{HL(n_p^k(s,a))}{n_p^k(s,a)} \right)$$

$$\le b_{\mathrm{prob}}\left( \hat{\mathbb{P}}_p^k(\cdot \mid s,a), n_p^k(s,a), \overline{V}_p^k, \underline{V}_p^k, 0 \right),$$

where the first inequality is from the definition of $E_{\mathrm{ind,val}}$; the second inequality is from Equation (33) of Lemma 16; the third inequality is from Lemma 23; the fourth inequality is from our assumption that for all $s' \in \mathcal{S}_{h+1}$, $\underline{V}_p^k(s') \le V^\star(s') \le \overline{V}_p^k(s')$, and thus $\left| (V_p^\star - \underline{V}_p^k)(s') \right| \le \left| (\overline{V}_p^k - \underline{V}_p^k)(s') \right|$ for all $s'$ in the support of $\hat{\mathbb{P}}_p^k(\cdot \mid s,a)$.

We prove Equation (26) as follows:

$$\left|(\hat{\mathbb{P}}^k - \mathbb{P}_p)(V_p^\star)(s,a)\right|$$

$$\leq \epsilon + \left|(\hat{\mathbb{P}}^k - \bar{\mathbb{P}}^k)(V_p^\star)(s,a)\right|$$

$$\leq \epsilon + O\left(\sqrt{\frac{\left(\sum_{q=1}^M w_q^k(s,a)\mathrm{var}_{\mathbb{P}_q(\cdot|s,a)}[V_p^\star]\right)L(n^k(s,a))}{n^k(s,a)}} + \frac{HL(n^k(s,a))}{n^k(s,a)}\right)$$

$$\leq \epsilon + O\left(\sqrt{\frac{\mathrm{var}_{\hat{\mathbb{P}}^k(\cdot|s,a)}[V_p^\star]\,L(n^k(s,a))}{n^k(s,a)}} + \sqrt{\frac{L(n^k(s,a))}{n^k(s,a)}}\cdot \epsilon H + \frac{HL(n^k(s,a))}{n^k(s,a)} + \frac{HL(n^k(s,a))}{n^k(s,a)}\right)$$

$$\leq 2\epsilon + O\left(\sqrt{\frac{\mathrm{var}_{\hat{\mathbb{P}}^k(\cdot|s,a)}[\overline{V}_p^k]\,L(n^k(s,a))}{n^k(s,a)}} + \sqrt{\frac{\|\overline{V}_p^k - V_p^\star\|_{\hat{\mathbb{P}}^k(\cdot|s,a)}^2\,L(n^k(s,a))}{n^k(s,a)}} + \frac{HL(n^k(s,a))}{n^k(s,a)}\right)$$

$$\leq 2\epsilon + O\left(\sqrt{\frac{\mathrm{var}_{\hat{\mathbb{P}}^k(\cdot|s,a)}[\overline{V}_p^k]\,L(n^k(s,a))}{n^k(s,a)}} + \sqrt{\frac{\|\overline{V}_p^k - \underline{V}_p^k\|_{\hat{\mathbb{P}}^k(\cdot|s,a)}^2\,L(n^k(s,a))}{n^k(s,a)}} + \frac{HL(n^k(s,a))}{n^k(s,a)}\right)$$

$$\leq b_{\mathrm{prob}}\left(\hat{\mathbb{P}}^k(\cdot\mid s,a), n^k(s,a), \overline{V}_p^k, \underline{V}_p^k, \epsilon\right),$$

where the first inequality is from the observation that $\|\bar{\mathbb{P}}^k(\cdot\mid s,a) - \mathbb{P}_p(\cdot\mid s,a)\|_1 \leq \frac{\epsilon}{H}$ and Lemma 24; the second inequality is from the definition of $E_{\mathrm{agg,val}}$; the third inequality is from Equation (34) of Lemma 16; the fourth inequality is from Lemma 23 and the observation that for constant $c > 0$, $c\sqrt{\frac{L(n^k(s,a))}{n^k(s,a)}\cdot \epsilon H} \leq \epsilon + \frac{c^2}{4}\frac{L(n^k(s,a))}{n^k(s,a)}$ by AM-GM inequality; the fifth inequality is from our assumption that for all $s' \in \mathcal{S}_{h+1}$, $\underline{V}_p^k(s') \leq V^\star(s') \leq \overline{V}_p^k(s')$, and thus $\left|(V_p^\star - \underline{V}_p^k)(s')\right| \leq \left|(\overline{V}_p^k - \underline{V}_p^k)(s')\right|$ for all $s'$ in the support of $\hat{\mathbb{P}}^k(\cdot\mid s,a)$.

3. We prove Equation (27) as follows:

$$\left|(\hat{\mathbb{P}}_p^k - \mathbb{P}_p)(V_2 - V_1)(s,a)\right|$$

$$\leq \sum_{s'\in\mathcal{S}_{h+1}}\left|(\hat{\mathbb{P}}_p^k - \mathbb{P}_p)(s'\mid s,a)\right|\cdot (V_2 - V_1)(s')$$

$$\leq O\left(\sum_{s'\in\mathcal{S}_{h+1}}\left(\sqrt{\frac{L(n_p^k(s,a))\cdot \mathbb{P}_p(s'\mid s,a)}{n_p^k(s,a)}} + \frac{L(n_p^k(s,a))}{n_p^k(s,a)}\right)\cdot (V_2 - V_1)(s')\right)$$

$$\leq O\left(\sum_{s'\in\mathcal{S}_{h+1}}\left(\sqrt{\frac{L(n_p^k(s,a))\cdot \hat{\mathbb{P}}_p^k(s'\mid s,a)}{n_p^k(s,a)}} + \frac{L(n_p^k(s,a))}{n_p^k(s,a)}\right)\cdot (V_2 - V_1)(s')\right)$$

$$\leq O\left(\sum_{s'\in\mathcal{S}_{h+1}}\sqrt{\hat{\mathbb{P}}_p^k(s'\mid s,a)}(\overline{V}_p^k - \underline{V}_p^k)(s')\cdot \sqrt{\frac{L(n_p^k(s,a))}{n_p^k(s,a)}} + \sum_{s'\in\mathcal{S}_{h+1}}\frac{HL(n_p^k(s,a))}{n_p^k(s,a)}\right)$$

$$\leq O\left(\sqrt{\frac{S\|\overline{V}_p^k - \underline{V}_p^k\|_{\hat{\mathbb{P}}_p^k(\cdot|s,a)}^2\,L(n_p^k(s,a))}{n_p^k(s,a)}} + \frac{SHL(n_p^k(s,a))}{n_p^k(s,a)}\right)$$

$$\leq b_{\mathrm{str}}\left(\hat{\mathbb{P}}_p^k(\cdot\mid s,a), n(s,a), \overline{V}_p^k, \underline{V}_p^k, 0\right),$$

where the first inequality is from the elementary fact that $\left|\sum_{i=1}^n a_i\right| \leq \sum_{i=1}^n |a_i|$; the second inequality is from the definition of $E_{\mathrm{ind,prob}}$; the third inequality is from the definition of $E_{\mathrm{ind,prob}}$ and Lemma 25; the fourth inequality is by algebra and $0 \leq (V_2 - V_1)(s') \leq \min(H, (\overline{V}_p^k - \underline{V}_p^k)(s'))$ for all $s' \in \mathcal{S}_{h+1}$; the fifth inequality is by Cauchy-Schwarz.

We now prove Equation (28):

$$\left|(\hat{\mathbb{P}}^k - \mathbb{P}_p)(V_2 - V_1)(s,a)\right|$$

$$\leq \left|(\bar{\mathbb{P}}^k - \mathbb{P}_p)(V_2 - V_1)(s,a)\right| + \left|(\hat{\mathbb{P}}^k - \bar{\mathbb{P}}^k)(V_2 - V_1)(s,a)\right|$$

$$\leq \epsilon + \sum_{s' \in \mathcal{S}_{h+1}} \left|(\hat{\mathbb{P}}^k - \bar{\mathbb{P}}^k)(s' \mid s,a)\right| \cdot (V_2 - V_1)(s')$$

$$\leq \epsilon + O\left(\sum_{s' \in \mathcal{S}_{h+1}} \left(\sqrt{\frac{L(n^k(s,a)) \cdot \bar{\mathbb{P}}^k(s' \mid s,a)}{n^k(s,a)}} + \frac{L(n^k(s,a))}{n^k(s,a)}\right) \cdot (V_2 - V_1)(s')\right)$$

$$\leq \epsilon + O\left(\sum_{s' \in \mathcal{S}_{h+1}} \left(\sqrt{\frac{L(n^k(s,a)) \cdot \hat{\mathbb{P}}^k(s' \mid s,a)}{n^k(s,a)}} + \frac{L(n^k(s,a))}{n^k(s,a)}\right) \cdot (V_2 - V_1)(s')\right)$$

$$\leq \epsilon + O\left(\sum_{s' \in \mathcal{S}_{h+1}} \sqrt{\hat{\mathbb{P}}^k(s' \mid s,a)}(\overline{V}_p^k - \underline{V}_p^k)(s') \cdot \sqrt{\frac{L(n^k(s,a))}{n^k(s,a)}} + \sum_{s' \in \mathcal{S}_{h+1}} \frac{HL(n^k(s,a))}{n^k(s,a)}\right)$$

$$\leq \epsilon + O\left(\sqrt{\frac{S\|\overline{V}_p^k - \underline{V}_p^k\|_{\hat{\mathbb{P}}^k(\cdot|s,a)}^2 \, L(n^k(s,a))}{n^k(s,a)}} + \frac{SHL(n^k(s,a))}{n^k(s,a)}\right)$$

$$\leq b_{\mathrm{str}}\left(\hat{\mathbb{P}}^k(\cdot \mid s,a), n(s,a), \overline{V}_p^k, \underline{V}_p^k, \epsilon\right),$$

where the first inequality is triangle inequality; the second inequality is from the elementary fact that $\left|\sum_{i=1}^n a_i\right| \leq \sum_{i=1}^n |a_i|$, along with $\|\bar{\mathbb{P}}_k(\cdot \mid s,a) - \mathbb{P}_p(\cdot \mid s,a)\|_1 \leq \frac{\epsilon}{H}$ and Lemma 24; the third inequality is from the definition of $E_{\mathrm{agg,prob}}$; the fourth inequality is from the definition of $E_{\mathrm{agg,prob}}$ and Lemma 25; the fifth inequality is by algebra and $0 \leq (V_2 - V_1)(s') \leq \min(H, (\overline{V}_p^k - \underline{V}_p^k)(s'))$ for all $s' \in \mathcal{S}_{h+1}$; the last inequality is by Cauchy-Schwarz. □

Lemma 13 relies on the following technical lemma on the concentrations of the conditional variances. Specifically, Equation (33) is well-known (see, e.g., [2, 30]); Equations (34) and (35) are new, and allow for heterogeneous data aggregation in the multi-task RL setting. We still include the proof of Equation (33) here, as it helps illustrate our ideas for proving the two new inequalities.

**Lemma 16.** *If event $E$ happens, then for any $s, a, k, p$, we have:*

*1.*

$$\left|\sqrt{\mathrm{var}_{\hat{\mathbb{P}}_p^k(\cdot|s,a)}\left[V_p^\star\right]} - \sqrt{\mathrm{var}_{\mathbb{P}_p(\cdot|s,a)}\left[V_p^\star\right]}\right| \lesssim H\sqrt{\frac{L(n_p^k(s,a))}{n_p^k(s,a)}}, \qquad (33)$$

*2.*

$$\left|\sqrt{\mathrm{var}_{\hat{\mathbb{P}}^k(\cdot|s,a)}\left[V_p^\star\right]} - \sqrt{\sum_{q=1}^M w_q^k(s,a)\mathrm{var}_{\mathbb{P}_q(\cdot|s,a)}\left[V_p^\star\right]}\right| \lesssim \sqrt{H\epsilon} + H\sqrt{\frac{L(n^k(s,a))}{n^k(s,a)}}, \qquad (34)$$

*and*

$$\left|\sqrt{\mathrm{var}_{\hat{\mathbb{P}}^k(\cdot|s,a)}\left[V_p^\star\right]} - \sqrt{\mathrm{var}_{\mathbb{P}_p(\cdot|s,a)}\left[V_p^\star\right]}\right| \lesssim \sqrt{H\epsilon} + H\sqrt{\frac{L(n^k(s,a))}{n^k(s,a)}}, \qquad (35)$$

*Proof.* 1. By the definition of $E$, we have

$$\left|\frac{1}{n_p^k(s,a)}\sum_{i=1}^{n_p^k(s,a)}(V_p^\star(s_i')-(\mathbb{P}_pV_p^\star)(s,a))^2-\mathrm{var}_{\mathbb{P}_p(\cdot|s,a)}[V_p^\star]\right|\lesssim\sqrt{\frac{H^2\mathrm{var}_{\mathbb{P}_p(\cdot|s,a)}[V_p^\star]L(n_p^k(s,a))}{n_p^k(s,a)}}+\frac{H^2L(n_p^k(s,a))}{n_p^k(s,a)};$$

this, when combined with Lemma 25, implies that

$$\left|\sqrt{\frac{1}{n_p^k(s,a)}\sum_{i=1}^{n_p^k(s,a)}(V_p^\star(s_i')-(\mathbb{P}_pV_p^\star)(s,a))^2}-\sqrt{\mathrm{var}_{\mathbb{P}_p(\cdot|s,a)}[V_p^\star]}\right|\le H\sqrt{\frac{L(n_p^k(s,a))}{n_p^k(s,a)}}. \quad (36)$$

Now, observe that

$$\mathrm{var}_{\hat{\mathbb{P}}_p^k(\cdot|s,a)}\left[V_p^\star\right]=\frac{1}{n_p^k(s,a)}\sum_{i=1}^{n_p^k(s,a)}(V_p^\star(s_i')-(\mathbb{P}_pV_p^\star)(s,a))^2-((\hat{\mathbb{P}}_p^kV_p^\star)(s,a)-(\mathbb{P}_pV_p^\star)(s,a))^2.$$

Recall that by the definition of event $E$, we have

$$\left|(\hat{\mathbb{P}}_p^kV_p^\star)(s,a)-(\mathbb{P}_pV_p^\star)(s,a)\right|\le H\wedge\left(\sqrt{\frac{H^2L(n_p^k(s,a))}{n_p^k(s,a)}}+\frac{HL(n_p^k(s,a))}{n_p^k(s,a)}\right)\le 2H\sqrt{\frac{L(n_p^k(s,a))}{n_p^k(s,a)}},$$

where the second inequality uses Lemma 26. Using the elementary fact that $|A-B|\le C\Rightarrow\sqrt{A}\le\sqrt{B}+\sqrt{C}$, we get that

$$\left|\sqrt{\mathrm{var}_{\hat{\mathbb{P}}_p^k(\cdot|s,a)}\left[V_p^\star\right]}-\sqrt{\frac{1}{n^k(s,a)}\sum_{i=1}^{n^k(s,a)}(V_p^\star(s_i')-(\mathbb{P}_pV_p^\star)(s,a))^2}\right|$$
$$\le\left|(\hat{\mathbb{P}}_p^kV_p^\star)(s,a)-(\mathbb{P}_pV_p^\star)(s,a)\right|\lesssim H\sqrt{\frac{L(n_p^k(s,a))}{n_p^k(s,a)}}. \quad (37)$$

Combining Equations (36) and (37), using algebra, we get

$$\left|\sqrt{\mathrm{var}_{\hat{\mathbb{P}}_p^k(\cdot|s,a)}\left[V_p^\star\right]}-\sqrt{\mathrm{var}_{\mathbb{P}_p(\cdot|s,a)}\left[V_p^\star\right]}\right|\lesssim H\sqrt{\frac{L(n_p^k(s,a))}{n_p^k(s,a)}},$$

establishing Equation (33).

2. We first show Equation (34). By the definition of $E$, we have

$$\left|\frac{1}{n^k(s,a)}\sum_{i=1}^{n^k(s,a)}(V_p^\star(s_i')-(\mathbb{P}_{p_i}V_p^\star)(s,a))^2-\sum_{p=1}^{M}w_p^k(s,a)\mathrm{var}_{\mathbb{P}_p(\cdot|s,a)}[V_p^\star]\right|$$
$$\lesssim\sqrt{\frac{H^2\left(\sum_{p=1}^{M}w_p^k(s,a)\mathrm{var}_{\mathbb{P}_p(\cdot|s,a)}[V_p^\star]\right)L(n^k(s,a))}{n^k(s,a)}}+\frac{H^2L(n^k(s,a))}{n^k(s,a)},$$

this, combined with Lemma 25, implies that

$$\left|\sqrt{\frac{1}{n^k(s,a)}\sum_{i=1}^{n^k(s,a)}(V_p^\star(s_i')-(\mathbb{P}_{p_i}V_p^\star)(s,a))^2}-\sqrt{\sum_{p=1}^{M}w_p^k(s,a)\mathrm{var}_{\mathbb{P}_p(\cdot|s,a)}[V_p^\star]}\right|\lesssim H\sqrt{\frac{L(n^k(s,a))}{n^k(s,a)}}. \quad (38)$$

For the first term on the left hand side, observe that for each $i$, $|(\mathbb{P}_{p_i}V_p^\star)(s,a)-(\mathbb{P}_pV_p^\star)(s,a)|\le H\frac{\epsilon}{H}=\epsilon$, we therefore have $\left|(V_p^\star(s_i')-(\mathbb{P}_{p_i}V_p^\star)(s,a))^2-(V_p^\star(s_i')-(\mathbb{P}_pV_p^\star)(s,a))^2\right|\le 2H\epsilon$ by

$2H$-Lipschitzness of function $f(x) = x^2$ on $[-H, H]$. By averaging over all $i$'s and taking square root, we have

$$\left| \sqrt{\frac{1}{n^k(s,a)} \sum_{i=1}^{n^k(s,a)} (V_p^\star(s_i') - (\mathbb{P}_{p_i} V_p^\star)(s,a))^2} - \sqrt{\frac{1}{n^k(s,a)} \sum_{i=1}^{n^k(s,a)} (V_p^\star(s_i') - (\mathbb{P}_p V_p^\star)(s,a))^2} \right| \lesssim \sqrt{H\epsilon}.$$

(39)

Furthermore,

$$\mathrm{var}_{\hat{\mathbb{P}}^k(\cdot|s,a)}\left[V_p^\star\right] = \frac{1}{n^k(s,a)} \sum_{i=1}^{n^k(s,a)} (V_p^\star(s_i') - (\mathbb{P}_p V_p^\star)(s,a))^2 - ((\hat{\mathbb{P}}^k V_p^\star)(s,a) - (\mathbb{P}_p V_p^\star)(s,a))^2,$$

and

$$\left| (\hat{\mathbb{P}}^k V_p^\star)(s,a) - (\mathbb{P}_p V_p^\star)(s,a) \right| \lesssim \epsilon + H\sqrt{\frac{L(n^k(s,a))}{n^k(s,a)}}$$

Together with our assumption that $\epsilon \leq 2H$ (which implies that $\epsilon \lesssim \sqrt{H\epsilon}$), this gives

$$\left| \sqrt{\mathrm{var}_{\hat{\mathbb{P}}^k(\cdot|s,a)}\left[V_p^\star\right]} - \sqrt{\frac{1}{n^k(s,a)} \sum_{i=1}^{n^k(s,a)} (V_p^\star(s_i') - (\mathbb{P}_p V_p^\star)(s,a))^2} \right| \lesssim \sqrt{H\epsilon} + H\sqrt{\frac{L(n^k(s,a))}{n^k(s,a)}}.$$

(40)

Equation (34) is a direct consequence of Equations (38), (39) and (40) along with algebra.

We now show Equation (35) using Equation (34). By Lemma 24, for every $q$, $\left| \mathrm{var}_{\mathbb{P}_q(\cdot|s,a)}\left[V_p^\star\right] - \mathrm{var}_{\mathbb{P}_p(\cdot|s,a)}\left[V_p^\star\right] \right| \leq 3H^2 \cdot \frac{\epsilon}{H} = 3H\epsilon$. Therefore, $\left| \sum_{q=1}^M w_q^k(s,a)\mathrm{var}_{\mathbb{P}_p(\cdot|s,a)}[V_p^\star] - \mathrm{var}_{\mathbb{P}_p(\cdot|s,a)}\left[V_p^\star\right] \right| \leq 3H^2 \cdot \frac{\epsilon}{H} = 3H\epsilon$, and

$$\left| \sqrt{\sum_{q=1}^M w_q^k(s,a)\mathrm{var}_{\mathbb{P}_q(\cdot|s,a)}[V_p^\star]} - \sqrt{\mathrm{var}_{\mathbb{P}_p(\cdot|s,a)}\left[V_p^\star\right]} \right| \lesssim \sqrt{H\epsilon}$$

This, together with Equation (34), implies

$$\left| \sqrt{\mathrm{var}_{\hat{\mathbb{P}}^k(\cdot|s,a)}\left[V_p^\star\right]} - \sqrt{\mathrm{var}_{\mathbb{P}_p(\cdot|s,a)}\left[V_p^\star\right]} \right| \lesssim \sqrt{H\epsilon} + H\sqrt{\frac{L(n^k(s,a))}{n^k(s,a)}},$$

establishing Equation (35). □

## C.3   Simplifying the surplus bounds

In this section, we show a distribution-dependent bound on the surplus terms, namely Lemma 19, which is key to establishing our regret bound. It can be seen as an extension of Proposition B.4 of [36] to our multitask setting using the MULTI-TASK-EULER algorithm, under the $\epsilon$-dissimilarity assumption. Before we present Lemma 19 (Section C.3.1), we first show and prove two auxiliary lemmas, Lemma 17 and Lemma 18.

**Lemma 17** (Bounds on $\overline{V}_p^k - \underline{V}_p^k$, generalization of [36], Lemma F.8). *If E happens, then for all $p \in [M]$, $k \in [K]$, $h \in [H+1]$ and $s \in \mathcal{S}_h$,*

$$(\overline{V}_p^k - \underline{V}_p^k)(s) \leq 4\mathbb{E}\left[ \sum_{t=h}^H \left( H \wedge \mathit{ind\text{-}b}_p^k(s_t, a_t) \wedge \mathit{agg\text{-}b}_p^k(s_t, a_t) \right) \mid s_h = s, \pi^k(p), \mathcal{M}_p \right]; \quad (41)$$

*consequently,*

$$(\overline{V}_p^k - \underline{V}_p^k)(s) \lesssim H \sum_{t=h}^H \mathbb{E}\left[ \left( 1 \wedge \sqrt{\frac{SL(n_p^k(s_t, a_t))}{n_p^k(s_t, a_t)}} \right) \mid s_h = s, \pi^k(p), \mathcal{M}_p \right]. \quad (42)$$

*Proof.* First, Lemmas 15 and 14 together imply that if $E$ holds, Equations (29) and (30) holds for all $p, k, s, a$. Under this premise, we show Equation (41) by backward induction.

**Base case:** for $h = H + 1$, we have that LHS is $(\overline{V}_p^k - \underline{V}_p^k)(\perp) = 0$ which is equal to the RHS.

**inductive case:** Suppose Equation (41) holds for all $s \in \mathcal{S}_{h+1}$. Now consider $s \in \mathcal{S}_h$. By the definitions of $\overline{V}_p^k$ and $\underline{V}_p^k$,

$$
\begin{aligned}
&(\overline{V}_p^k - \underline{V}_p^k)(s) \\
=&\overline{Q}_p^k(s, \pi_p^k(s)) - \underline{Q}_p^k(s, \pi_p^k(s)) \\
\leq&(\mathbb{P}_p(\overline{V}_p^k - \underline{V}_p^k))(s, \pi_p^k(s)) + 4(H \wedge \text{ind-}b_p^k(s, \pi_p^k(s)) \wedge \text{agg-}b_p^k(s, \pi_p^k(s))) \\
=&\mathbb{E}\left[ 4\min(H, \text{ind-}b_p^k(s, a), \text{agg-}b_p^k(s, a)) + (\overline{V}_p^k - \underline{V}_p^k)(s_{h+1}) \mid s_h = s, \pi_p^k, \mathcal{M}_p \right] \\
\leq&\mathbb{E}\left[ 4(H \wedge \text{ind-}b_p^k(s, a) \wedge \text{agg-}b_p^k(s, a)) + \mathbb{E}\left[ \sum_{t=h+1}^{H} \left( H \wedge 2\text{ind-}b_p^k(s_t, a_t) \wedge 2\text{agg-}b_p^k(s_t, a_t) \right) \mid s_{h+1} \right] \mid s_h = s, \pi_p^k, \mathcal{M}_p \right] \\
\leq&4\mathbb{E}\left[ \sum_{t=h}^{H} \left( H \wedge \text{ind-}b_p^k(s_t, a_t) \wedge \text{agg-}b_p^k(s_t, a_t) \right) \mid s_h = s, \pi_p^k, \mathcal{M}_p \right],
\end{aligned}
$$

where the first inequality is from Equations (29) and (30) for $(s, a)$ and player $p$ at episode $k$, and the second inequality is from the inductive hypothesis; the third inequality is by algebra. This completes the induction.

We now show Equation (42). By the definition of ind-$b_p^k(s, a)$ and algebra,

$$
\begin{aligned}
&\text{ind-}b_p^k(s, a) \\
\lesssim&\sqrt{\frac{\text{var}_{\hat{\mathbb{P}}_p^k(\cdot|s,a)}\left[\overline{V}_p^k\right] L(n_p^k(s, a))}{n_p^k(s, a)}} + \sqrt{\frac{L(n_p^k(s, a))}{n_p^k(s, a)}} + \sqrt{\frac{S\|\overline{V}_p^k - \underline{V}_p^k\|_{\hat{\mathbb{P}}_p^k(\cdot|s,a)}^2 L(n_p^k(s, a))}{n_p^k(s, a)}} + \frac{HSL(n_p^k(s, a))}{n_p^k(s, a)} \\
\lesssim&H\sqrt{\frac{SL(n_p^k(s_t, a_t))}{n_p^k(s_t, a_t)}} + \frac{HSL(n_p^k(s_t, a_t))}{n_p^k(s_t, a_t)},
\end{aligned}
$$

where the second inequality uses $\text{var}_{\hat{\mathbb{P}}_p^k(\cdot|s,a)}\left[\overline{V}_p^k\right] \leq H^2$ and $\|\overline{V}_p^k - \underline{V}_p^k\|_{\hat{\mathbb{P}}_p^k}^2 \leq H^2$.

As a consequence, using Lemma 26,

$$
\begin{aligned}
H \wedge \text{ind-}b_p^k(s_t, a_t) \wedge \text{agg-}b_p^k(s_t, a_t) \lesssim&H \wedge \left( H\sqrt{\frac{SL(n_p^k(s_t, a_t))}{n_p^k(s_t, a_t)}} + \frac{HSL(n_p^k(s_t, a_t))}{n_p^k(s_t, a_t)} \right) \\
\lesssim&H\left( 1 \wedge \sqrt{\frac{SL(n_p^k(s_t, a_t))}{n_p^k(s_t, a_t)}} \right). \qquad \square
\end{aligned}
$$

**Lemma 18.** *If $E$ happens, we have the following statements holding for all $p, k, s, a$:*

*1. For two terms that appear in ind-$b_p^k(s, a)$, they are bounded respectively as:*

$$
\|\overline{V}_p^k - \underline{V}_p^k\|_{\hat{\mathbb{P}}_p^k(\cdot|s,a)}^2 \lesssim \|\overline{V}_p^k - \underline{V}_p^k\|_{\mathbb{P}_p(\cdot|s,a)}^2 + \frac{H^2 SL(n_p^k(s, a))}{n_p^k(s, a)} \tag{43}
$$

$$\sqrt{\frac{\mathrm{var}_{\hat{\mathbb{P}}_p^k(\cdot|s,a)}\left[\overline{V}_p^k\right]L(n_p^k(s,a))}{n_p^k(s,a)}} \lesssim \sqrt{\frac{\mathrm{var}_{\mathbb{P}_p(\cdot|s,a)}\left[V_p^{\pi^k}\right]L(n_p^k(s,a))}{n_p^k(s,a)}}$$
$$+ \sqrt{\frac{\|\overline{V}_p^k - \underline{V}_p^k\|_{\mathbb{P}_p(\cdot|s,a)}^2 L(n_p^k(s,a))}{n_p^k(s,a)}} + \frac{H\sqrt{S}L(n_p^k(s,a))}{n_p^k(s,a)}$$

(44)

2. *For two terms that appear in agg-$b_p^k(s,a)$, they are bounded respectively as:*

$$\|\overline{V}_p^k - \underline{V}_p^k\|_{\hat{\mathbb{P}}^k(\cdot|s,a)}^2 \lesssim 2\|\overline{V}_p^k - \underline{V}_p^k\|_{\mathbb{P}_p(\cdot|s,a)}^2 + \frac{H^2 S L(n_p^k(s,a))}{n_p^k(s,a)} + H\epsilon \qquad (45)$$

$$\sqrt{\frac{\mathrm{var}_{\hat{\mathbb{P}}^k(\cdot|s,a)}\left[\overline{V}_p^k\right]L(n^k(s,a))}{n^k(s,a)}} \lesssim \sqrt{\frac{\mathrm{var}_{\mathbb{P}_p(\cdot|s,a)}\left[V_p^{\pi^k}\right]L(n^k(s,a))}{n^k(s,a)}} + \sqrt{\frac{\|\overline{V}_p^k - \underline{V}_p^k\|_{\mathbb{P}_p(\cdot|s,a)}^2 L(n^k(s,a))}{n^k(s,a)}}$$
$$+ \frac{H\sqrt{S}L(n^k(s,a))}{n^k(s,a)} + \sqrt{\frac{H\epsilon L(n^k(s,a))}{n^k(s,a)}}$$

(46)

*Proof.* First, Lemmas 15 and 14 together imply that if $E$ happens, the value function upper and lower bounds are valid. Conditioned on $E$ happening, we prove the two items respectively.

1. For Equation (43), using the definition of $E_{\mathrm{ind,prob}}$ and AM-GM inequality, when $E$ happens, we have for all $p,k,s,a,s'$,

$$\hat{\mathbb{P}}_p^k(s' \mid s,a) \lesssim \mathbb{P}_p(s' \mid s,a) + \frac{L(n_p^k(s,a))}{n_p^k(s,a)}. \qquad (47)$$

This implies that

$$\|\overline{V}_p^k - \underline{V}_p^k\|_{\hat{\mathbb{P}}_p^k(\cdot|s,a)}^2$$
$$= \sum_{s'\in\mathcal{S}_{h+1}} \hat{\mathbb{P}}_p^k(s' \mid s,a)(\overline{V}_p^k(s') - \underline{V}_p^k(s'))^2$$
$$\lesssim \sum_{s'\in\mathcal{S}_{h+1}} \mathbb{P}_p^k(s' \mid s,a)(\overline{V}_p^k(s') - \underline{V}_p^k(s'))^2 + \sum_{s'\in\mathcal{S}_{h+1}} \frac{L(n_p^k(s,a))}{n_p^k(s,a)} \cdot H^2$$
$$\lesssim \|\overline{V}_p^k - \underline{V}_p^k\|_{\mathbb{P}_p(\cdot|s,a)}^2 + \frac{SH^2 L(n_p^k(s,a))}{n_p^k(s,a)},$$

where the first inequality is from Equation (47), and the fact that $\overline{V}_p^k(s') - \underline{V}_p^k(s') \in [0,H]$ for any $s' \in \mathcal{S}_{h+1}$; the second inequality is by algebra.

For Equation (44), we have:

$$\sqrt{\frac{\mathrm{var}_{\hat{\mathbb{P}}_p^k(\cdot|s,a)}\left[\overline{V}_p^k\right]L(n_p^k(s,a))}{n_p^k(s,a)}}$$
$$\lesssim \sqrt{\frac{\mathrm{var}_{\hat{\mathbb{P}}_p^k(\cdot|s,a)}\left[V_p^\star\right]L(n_p^k(s,a))}{n_p^k(s,a)}} + \sqrt{\frac{\|\overline{V}_p^k - \underline{V}_p^k\|_{\hat{\mathbb{P}}_p^k(\cdot|s,a)}^2 L(n_p^k(s,a))}{n_p^k(s,a)}}$$
$$\lesssim \sqrt{\frac{\mathrm{var}_{\mathbb{P}_p(\cdot|s,a)}\left[V_p^\star\right]L(n_p^k(s,a))}{n_p^k(s,a)}} + \sqrt{\frac{\|\overline{V}_p^k - \underline{V}_p^k\|_{\mathbb{P}_p(\cdot|s,a)}^2 L(n_p^k(s,a))}{n_p^k(s,a)}} + \frac{\sqrt{S}HL(n_p^k(s,a))}{n_p^k(s,a)}$$
$$\lesssim \sqrt{\frac{\mathrm{var}_{\mathbb{P}_p(\cdot|s,a)}\left[V_p^{\pi^k}\right]L(n_p^k(s,a))}{n_p^k(s,a)}} + \sqrt{\frac{\|\overline{V}_p^k - \underline{V}_p^k\|_{\mathbb{P}_p(\cdot|s,a)}^2 L(n_p^k(s,a))}{n_p^k(s,a)}} + \frac{\sqrt{S}HL(n_p^k(s,a))}{n_p^k(s,a)}$$

where the first inequality is from Lemma 23 and the observation that when $E$ happens, $\left|(\overline{V}_p^k - V_p^\star)(s')\right| \le \left|(\overline{V}_p^k - \underline{V}_p^k)(s')\right|$ for all $s' \in \mathcal{S}_{h+1}$; the second inequality is from Equation (33) of Lemma 16 and Equation (43); the third inequality again uses Lemma 23 and the observation that when $E$ happens, $\left|(V_p^\star - V_p^{\pi^k})(s')\right| \le \left|(\overline{V}_p^k - \underline{V}_p^k)(s')\right|$ for all $s' \in \mathcal{S}_{h+1}$.

2. For Equation (45), using the definition of $E_{\text{agg,prob}}$ and AM-GM inequality, when $E$ happens, we have for all $p, k, s, a, s'$,

$$\hat{\mathbb{P}}^k(s' \mid s, a) \lesssim \bar{\mathbb{P}}^k(s' \mid s, a) + \frac{L(n^k(s,a))}{n^k(s,a)}. \tag{48}$$

This implies that

$$\|\overline{V}_p^k - \underline{V}_p^k\|_{\hat{\mathbb{P}}^k(\cdot|s,a)}^2$$

$$= \sum_{s' \in \mathcal{S}_{h+1}} \hat{\mathbb{P}}^k(s' \mid s, a)(\overline{V}_p^k(s') - \underline{V}_p^k(s'))^2$$

$$\lesssim 2 \sum_{s' \in \mathcal{S}_{h+1}} \bar{\mathbb{P}}^k(s' \mid s, a)(\overline{V}_p^k(s') - \underline{V}_p^k(s'))^2 + \sum_{s' \in \mathcal{S}_{h+1}} \frac{L(n_p^k(s,a))}{n_p^k(s,a)} \cdot H^2$$

$$\lesssim 2 \sum_{s' \in \mathcal{S}_{h+1}} \mathbb{P}_p(s' \mid s, a)(\overline{V}_p^k(s') - \underline{V}_p^k(s'))^2 + \epsilon H + \frac{SH^2 L(n_p^k(s,a))}{n_p^k(s,a)}$$

$$\lesssim \|\overline{V}_p^k - \underline{V}_p^k\|_{\mathbb{P}_p(\cdot|s,a)}^2 + \frac{SH^2 L(n_p^k(s,a))}{n_p^k(s,a)} + \epsilon H,$$

where the first inequality is from Equation (48) and the fact that $\overline{V}_p^k(s') - \underline{V}_p^k(s') \in [0, H]$ for any $s' \in \mathcal{S}_{h+1}$; the second inequality is from the observation that $\|\mathbb{P}_p(\cdot \mid s, a) - \bar{\mathbb{P}}^k(\cdot \mid s, a)\|_1 \le \frac{\epsilon}{H}$; the third inequality is by algebra.

For Equation (46), we have:

$$\sqrt{\frac{\text{var}_{\hat{\mathbb{P}}^k(\cdot|s,a)}\left[\overline{V}_p^k\right] L(n_p^k(s,a))}{n_p^k(s,a)}}$$

$$\lesssim \sqrt{\frac{\text{var}_{\hat{\mathbb{P}}^k(\cdot|s,a)}\left[V_p^\star\right] L(n_p^k(s,a))}{n_p^k(s,a)}} + \sqrt{\frac{\|\overline{V}_p^k - \underline{V}_p^k\|_{\hat{\mathbb{P}}^k(\cdot|s,a)}^2 L(n_p^k(s,a))}{n_p^k(s,a)}}$$

$$\lesssim \sqrt{\frac{\text{var}_{\mathbb{P}_p(\cdot|s,a)}\left[V_p^\star\right] L(n_p^k(s,a))}{n_p^k(s,a)}} + \sqrt{\frac{\|\overline{V}_p^k - \underline{V}_p^k\|_{\mathbb{P}_p(\cdot|s,a)}^2 L(n_p^k(s,a))}{n_p^k(s,a)}} + \frac{\sqrt{S} H L(n_p^k(s,a))}{n_p^k(s,a)} + \sqrt{\frac{H \epsilon L(n_p^k(s,a))}{n_p^k(s,a)}}$$

$$\lesssim \sqrt{\frac{\text{var}_{\mathbb{P}_p(\cdot|s,a)}\left[V_p^{\pi^k}\right] L(n_p^k(s,a))}{n_p^k(s,a)}} + \sqrt{\frac{\|\overline{V}_p^k - \underline{V}_p^k\|_{\mathbb{P}_p(\cdot|s,a)}^2 L(n_p^k(s,a))}{n_p^k(s,a)}} + \frac{\sqrt{S} H L(n_p^k(s,a))}{n_p^k(s,a)} + \sqrt{\frac{H \epsilon L(n_p^k(s,a))}{n_p^k(s,a)}},$$

where the first inequality is from Lemma 23 and the observation that when $E$ happens, $\left|(\overline{V}_p^k - V_p^\star)(s')\right| \le \left|(\overline{V}_p^k - \underline{V}_p^k)(s')\right|$ for $s' \in \mathcal{S}_{h+1}$; the second inequality uses Equation (35) of Lemma 16 and Equation (45); the third inequality is from Lemma 23 and the observation that when $E$ happens, $\left|(\overline{V}_p^\star - V_p^{\pi^k})(s')\right| \le \left|(\overline{V}_p^k - \underline{V}_p^k)(s')\right|$ for $s' \in \mathcal{S}_{h+1}$. $\qquad\square$

### C.3.1 Distribution-dependent bound on the surplus terms

**Lemma 19** (Surplus bound). *If $E$ happens, then for all $p, k, s, a$:*

$$E_p^k(s,a) \lesssim B_p^{k,\text{lead}}(s,a) + \mathbb{E}\left[\sum_{t=h}^{H} B_p^{k,\text{fut}}(s_t, a_t) \mid (s_h, a_h) = (s,a), \pi^k(p), \mathcal{M}_p\right],$$

*where*

$$B_p^{k,\text{lead}}(s,a) = H \wedge \left( 5\epsilon + O\left( \sqrt{\frac{\left(1 + \text{var}_{\mathbb{P}_p(\cdot|s,a)}[V_p^{\pi^k}]\right) L(n^k(s,a))}{n^k(s,a)}} \right) \right)$$

$$\wedge\, O\left( \sqrt{\frac{\left(1 + \text{var}_{\mathbb{P}_p(\cdot|s,a)}[V_p^{\pi^k}]\right) L(n_p^k(s,a))}{n_p^k(s,a)}} \right),$$

$$B_p^{k,\text{fut}}(s,a) = H^3 \wedge O\left( \frac{H^3 S L(n_p^k(s,a))}{n_p^k(s,a)} \right).$$

*Proof of Lemma 19.* First, Lemmas 15 and 14 together imply that if $E$ holds, for all $p, k, s, a$, $E_p^k(s,a) \leq 2\left( H \wedge \text{ind-}b_p^k(s,a) \wedge \text{agg-}b_p^k(s,a) \right)$. We now bound $\text{ind-}b_p^k(s,a)$ and $\text{agg-}b_p^k(s,a)$ respectively.

**Bounding ind-$b_p^k(s,a)$:**   We have

$$\text{ind-}b_p^k(s,a)$$
$$=O\left( \sqrt{\frac{\text{var}_{\hat{\mathbb{P}}_p^k(\cdot|s,a)}[\overline{V}_p^k] L(n_p^k(s,a))}{n_p^k(s,a)}} + \sqrt{\frac{L(n_p^k(s,a))}{n_p^k(s,a)}} + \sqrt{\frac{S\|\overline{V}_p^k - \underline{V}_p^k\|_{\hat{\mathbb{P}}_p^k(\cdot|s,a)}^2\, L(n_p^k(s,a))}{n_p^k(s,a)}} + \frac{SHL(n_p^k(s,a))}{n_p^k(s,a)} \right)$$

$$\leq O\left( \sqrt{\frac{\text{var}_{\mathbb{P}_p(\cdot|s,a)}[V_p^{\pi^k}] L(n_p^k(s,a))}{n_p^k(s,a)}} + \sqrt{\frac{L(n_p^k(s,a))}{n_p^k(s,a)}} + \sqrt{\frac{S\|\overline{V}_p^k - \underline{V}_p^k\|_{\mathbb{P}_p(\cdot|s,a)}^2\, L(n_p^k(s,a))}{n_p^k(s,a)}} + \frac{SHL(n_p^k(s,a))}{n_p^k(s,a)} \right)$$

$$\leq O\left( \sqrt{\frac{\left(1 + \text{var}_{\mathbb{P}_p(\cdot|s,a)}[V_p^{\pi^k}]\right) L(n_p^k(s,a))}{n_p^k(s,a)}} + \sqrt{\frac{S\|\overline{V}_p^k - \underline{V}_p^k\|_{\mathbb{P}_p(\cdot|s,a)}^2\, L(n_p^k(s,a))}{n_p^k(s,a)}} + \frac{SHL(n_p^k(s,a))}{n_p^k(s,a)} \right)$$

$$\leq O\left( \sqrt{\frac{\left(1 + \text{var}_{\mathbb{P}_p(\cdot|s,a)}[V_p^{\pi^k}]\right) L(n_p^k(s,a))}{n_p^k(s,a)}} + \|\overline{V}_p^k - \underline{V}_p^k\|_{\mathbb{P}_p(\cdot|s,a)}^2 + \frac{SHL(n_p^k(s,a))}{n_p^k(s,a)} \right)$$

where the first inequality is by expanding the definition of $\text{ind-}b_p^k(s,a)$ and algebra; the second inequality is from Equations Equation (43) and (44) of Lemma 18, along with algebra; the third inequality is by the basic fact that $\sqrt{A} + \sqrt{B} \lesssim \sqrt{A + B}$; the fourth inequality is by AM-GM inequality.

**Bounding agg-$b_p^k(s,a)$:** We have:

$$\text{agg-}b_p^k(s,a)$$

$$\lesssim 4\epsilon + O\left(\sqrt{\frac{\text{var}_{\hat{\mathbb{P}}^k(\cdot|s,a)}[\overline{V}_p^k]\ L(n^k(s,a))}{n^k(s,a)}} + \sqrt{\frac{L(n^k(s,a))}{n^k(s,a)}} + \sqrt{\frac{S\|\overline{V}_p^k - \underline{V}_p^k\|_{\hat{\mathbb{P}}^k(\cdot|s,a)}^2\ L(n^k(s,a))}{n^k(s,a)}} + \frac{SHL(n^k(s,a))}{n^k(s,a)}\right)$$

$$\lesssim 5\epsilon + O\left(\sqrt{\frac{\text{var}_{\mathbb{P}_p(\cdot|s,a)}[V_p^{\pi^k}]\ L(n^k(s,a))}{n^k(s,a)}} + \sqrt{\frac{L(n^k(s,a))}{n^k(s,a)}} + \sqrt{\frac{S\|\overline{V}_p^k - \underline{V}_p^k\|_{\mathbb{P}_p(\cdot|s,a)}^2\ L(n^k(s,a))}{n^k(s,a)}} + \frac{SHL(n^k(s,a))}{n^k(s,a)}\right)$$

$$\lesssim 5\epsilon + O\left(\sqrt{\frac{\left(1 + \text{var}_{\mathbb{P}_p(\cdot|s,a)}[V_p^{\pi^k}]\right)\ L(n^k(s,a))}{n^k(s,a)}} + \sqrt{\frac{S\|\overline{V}_p^k - \underline{V}_p^k\|_{\mathbb{P}_p(\cdot|s,a)}^2\ L(n^k(s,a))}{n^k(s,a)}} + \frac{SHL(n^k(s,a))}{n^k(s,a)}\right)$$

$$\leq 5\epsilon + O\left(\sqrt{\frac{\left(1 + \text{var}_{\mathbb{P}_p(\cdot|s,a)}[V_p^{\pi^k}]\right)\ L(n^k(s,a))}{n^k(s,a)}} + \|\overline{V}_p^k - \underline{V}_p^k\|_{\mathbb{P}_p(\cdot|s,a)}^2 + \frac{SHL(n^k(s,a))}{n^k(s,a)}\right)$$

where the first inequality is by expanding the definition of agg-$b_p^k(s,a)$ and algebra; the second inequality is from Equations (46) and Equation (45) of Lemma 18, along with the observation that $\sqrt{\frac{S\epsilon HL(n^k(s,a))}{n^k(s,a)}} \leq \frac{SHL(n^k(s,a))}{n^k(s,a)} + \epsilon$ by AM-GM inequality; the third inequality is by the basic fact that $\sqrt{A} + \sqrt{B} \lesssim \sqrt{A+B}$; the fourth inequality is from AM-GM inequality.

Combining the above upper bounds, and using the observation that $\frac{L(n^k(s,a))}{n^k(s,a)} \leq \frac{L(n_p^k(s,a))}{n_p^k(s,a)}$, we get

$$\text{ind-}b_p^k(s,a) \wedge \text{agg-}b_p^k(s,a) \wedge H$$

$$\leq O\left(\sqrt{\frac{\left(1 + \text{var}_{\mathbb{P}_p(\cdot|s,a)}[V_p^{\pi^k}]\right)\ L(n_p^k(s,a))}{n_p^k(s,a)}}\right) \wedge \left(5\epsilon + O\left(\sqrt{\frac{\left(1 + \text{var}_{\mathbb{P}_p(\cdot|s,a)}[V_p^{\pi^k}]\right)\ L(n^k(s,a))}{n^k(s,a)}}\right)\right) \wedge H$$

$$+ O\left(\|\overline{V}_p^k - \underline{V}_p^k\|_{\mathbb{P}_p(\cdot|s,a)}^2 + \left(\frac{SHL(n_p^k(s,a))}{n_p^k(s,a)} \wedge H\right)\right)$$

$$\leq B^{k,\text{lead}}(s,a) + O\left(\|\overline{V}_p^k - \underline{V}_p^k\|_{\mathbb{P}_p(\cdot|s,a)}^2 + \left(\frac{SHL(n_p^k(s,a))}{n_p^k(s,a)} \wedge H\right)\right).$$

We now show that

$$\|\overline{V}_p^k - \underline{V}_p^k\|_{\mathbb{P}_p(\cdot|s,a)}^2 + \left(\frac{SHL(n_p^k(s,a))}{n_p^k(s,a)} \wedge H\right) \lesssim \mathbb{E}\left[\sum_{t=h}^{H} B^{k,\text{fut}}(s_t, a_t) \mid (s_h, a_h) = (s,a), \pi^k(p), \mathcal{M}_p\right],$$

$$(49)$$

which will conclude the proof. To this end, we simplify the left hand side of Equation (49) using Lemma 17:

$$\|\overline{V}_p^k - \underline{V}_p^k\|_{\mathbb{P}_p(\cdot|s,a)}^2 + \left(\frac{SHL(n^k(s,a))}{n^k(s,a)} \wedge H\right)$$

$$\lesssim \mathbb{E}\left[\left(H\sum_{t=h+1}^{H} \mathbb{E}\left[\left(1 \wedge \sqrt{\frac{SL(n_p^k(s_t,a_t))}{n_p^k(s_t,a_t)}}\right) \mid s_{h+1}\right]\right)^2 \mid (s_h,a_h) = (s,a), \pi^k(p), \mathcal{M}_p\right] + \left(\frac{SHL(n^k(s,a))}{n^k(s,a)} \wedge H\right)$$

$$\lesssim H^3 \mathbb{E}\left[\sum_{t=h+1}^{H} \mathbb{E}\left[\left(1 \wedge \sqrt{\frac{SL(n_p^k(s_t,a_t))}{n_p^k(s_t,a_t)}}\right)^2 \mid s_{h+1}\right] \mid (s_h,a_h) = (s,a), \pi^k(p), \mathcal{M}_p\right] + \left(\frac{SHL(n^k(s,a))}{n^k(s,a)} \wedge H\right)$$

$$\lesssim \mathbb{E}\left[\sum_{t=h}^{H} H^3 \wedge \frac{H^3 SL(n_p^k(s_t,a_t))}{n_p^k(s_t,a_t)} \mid (s_h,a_h) = (s,a), \pi^k(p), \mathcal{M}_p\right]$$

$$\lesssim \mathbb{E}\left[\sum_{t=h}^{H} B^{k,\mathrm{fut}}(s_t,a_t) \mid (s_h,a_h) = (s,a), \pi^k(p), \mathcal{M}_p\right],$$

where the first inequality is from Equation (42) of Lemma 17; the second inequality is by Cauchy-Schwarz and $\mathbb{E}[X]^2 \leq \mathbb{E}[X^2]$ for any random variable $X$; and the third inequality is by the law of total expectation and algebra. □

### C.4 Concluding the regret bounds

In this section, we present the proofs of Theorems 5 and 6.

To bound the collective regret of MULTI-TASK-EULER, we first recall the following general result from [36], which is useful to establish instance-dependent regret guarantees for episodic RL.

**Lemma 20** (Clipping lemma, [36], Lemma B.6). *Fix player $p \in [M]$; suppose for each episode $k$, it follows $\pi^k(p)$, the greedy policy with respect to $\overline{Q}_p^k$. In addition, there exists some event $E$ and a collection of functions $\left\{B_p^{k,\mathrm{lead}}, B_p^{k,\mathrm{fut}}\right\}_{k \in [K]} \subset (\mathcal{S} \times \mathcal{A} \to \mathbb{R})$, such that if $E$ happens, then for all $k \in [K]$, $h \in [H]$ and $(s,a) \in \mathcal{S}_h \times \mathcal{A}$, the surplus of $\overline{Q}_p^k$ satisfies that*

$$0 \leq E_p^k(s,a) \lesssim B_p^{k,\mathrm{lead}}(s,a) + \mathbb{E}\left[\sum_{t=h}^{H} B_p^{k,\mathrm{fut}}(s_t,a_t) \mid (s_h,a_h) = (s,a), \pi^k(p), \mathcal{M}_p\right],$$

*then, on $E$:*

$$\mathrm{Reg}(K,p) \lesssim \sum_{s,a}\sum_k \rho_p^k(s,a) \, \mathrm{clip}\left(B_p^{k,\mathrm{lead}}(s,a), \mathrm{g\breve{a}p}_p(s,a)\right) + H\sum_{s,a}\sum_k \rho_p^k(s,a) \, \mathrm{clip}\left(B_p^{k,\mathrm{fut}}(s,a), \frac{\mathrm{gap}_{p,\min}}{8SAH^2}\right),$$

*here, recall that $\mathrm{clip}(\alpha, \Delta) = \alpha\mathbf{1}(\alpha \geq \Delta)$, and $\mathrm{g\breve{a}p}_p(s,a) = \frac{\mathrm{gap}_p(s,a)}{4H} \vee \frac{\mathrm{gap}_{p,\min}}{4H}$.*

**Remark 21.** *Our presentation of the clipping lemma is slightly different than the original one [36, Lemma B.6], in that:*

1. *We consider layered MDPs, while [36] consider general stationary MDPs where one state may be experienced at multiple different steps in $[H]$. Specifically, in a layered MDP, the occupancy distributions $\omega_{k,h}$ defined in [36] is only supported over $\mathcal{S}_h \times \mathcal{A}$. As a result, in the presentation here, we no longer need to sum over $h$ – this is already captured in the sum over all $s$ across all layers.*

2. *Our presentation here is in the context of multitask RL, which is with respect to a player $p \in [M]$, its corresponding MDP $\mathcal{M}_p$, and its policies used throughout the process $\left\{\pi^k(p)\right\}_{k=1}^{K}$. As a result, all quantities have $p$ as subscripts.*

We are now ready to prove Theorems 5 and 6, MULTI-TASK-EULER's main regret theorems.

### C.4.1 Proof of Theorem 5

*Proof of Theorem 5.* From Lemma 20 and Lemma 19, we have that when $E$ happens,

$$
\begin{aligned}
\mathrm{Reg}(K) &= \sum_{p=1}^{M} \mathrm{Reg}(K,p) \\
&\leq \underbrace{\sum_{s,a} \sum_{k,p} \rho_p^k(s,a)\,\mathrm{clip}\left( B^{k,\mathrm{lead}}(s,a), \mathrm{g\check{a}p}_p(s,a) \right)}_{(A)} + \underbrace{H \sum_{s,a} \sum_{k,p} \rho_p^k(s,a)\,\mathrm{clip}\left( B^{k,\mathrm{fut}}(s,a), \frac{\mathrm{gap}_{p,\min}}{8SAH^2} \right)}_{(B)},
\end{aligned}
\tag{50}
$$

We bound each term separately. We can directly use Lemma 22 to bound term $(B)$ as:

$$
H \sum_{s,a} \sum_{k,p} \rho_p^k(s,a)\,\mathrm{clip}\left( B^{k,\mathrm{fut}}(s,a), \frac{\mathrm{gap}_{p,\min}}{8SAH^2} \right) \lesssim MH^4 S^2 A \left( \ln\left( \frac{MSAK}{\delta} \right) \right)^2.
\tag{51}
$$

For term $(A)$, we will group the sum by $(s,a) \in \mathcal{I}_\epsilon$ and $(s,a) \notin \mathcal{I}_\epsilon$ separately.

**Case 1: $(s,a) \in \mathcal{I}_\epsilon$.** In this case, we have that for all $p$, $\mathrm{g\check{a}p}_p(s,a) = \frac{\mathrm{gap}_p(s,a)}{4H} \geq 24\epsilon$. We simplify the corresponding term as follows:

$$
\begin{aligned}
&\sum_{(s,a)\in\mathcal{I}_\epsilon} \sum_{k,p} \rho_p^k(s,a)\,\mathrm{clip}\left( B^{k,\mathrm{lead}}(s,a), \mathrm{g\check{a}p}_p(s,a) \right) \\
&\leq \sum_{(s,a)\in\mathcal{I}_\epsilon} \sum_{k,p} \rho_p^k(s,a)\,\mathrm{clip}\left( H \wedge \left( 5\epsilon + O\left( \sqrt{\frac{(1+\mathrm{var}_{\mathbb{P}_p(\cdot|s,a)}[V_p^{\pi^k}]) L(n^k(s,a))}{n^k(s,a)}} \right) \right), \frac{\min_p \mathrm{gap}_p(s,a)}{4H} \right) \\
&\leq \sum_{(s,a)\in\mathcal{I}_\epsilon} \sum_{k,p} \rho_p^k(s,a) \left( H \wedge \mathrm{clip}\left( 5\epsilon + O\left( \sqrt{\frac{(1+\mathrm{var}_{\mathbb{P}_p(\cdot|s,a)}[V_p^{\pi^k}]) L(n^k(s,a))}{n^k(s,a)}} \right), \frac{\min_p \mathrm{gap}_p(s,a)}{4H} \right) \right) \\
&\lesssim \sum_{(s,a)\in\mathcal{I}_\epsilon} \sum_{k,p} \rho_p^k(s,a) \left( H \wedge \sqrt{\frac{(1+\mathrm{var}_{\mathbb{P}_p(\cdot|s,a)}[V_p^{\pi^k}]) L(n^k(s,a))}{n^k(s,a)}} \right)
\end{aligned}
$$

where the first inequality is from the definition of $B^{k,\mathrm{lead}}$; the second inequality is from the basic fact that $\mathrm{clip}(A \wedge B, C) \leq A \wedge \mathrm{clip}(B,C)$; the third inequality uses Lemma 27 with $a_1 = 5\epsilon$, $a_2 = \sqrt{\frac{(1+\mathrm{var}_{\mathbb{P}_p(\cdot|s,a)}[V_p^{\pi^k}]) L(n^k(s,a))}{n^k(s,a)}}$, and $\Delta = \frac{\min_p \mathrm{gap}_p(s,a)}{4H}$, along with the observation that $\mathrm{clip}(5\epsilon, \frac{\min_p \mathrm{gap}_p(s,a)}{16H}) = 0$, since for all $(s,a) \in \mathcal{I}_\epsilon$ and all $p \in [M]$, $\mathrm{gap}_p(s,a) \geq 96\epsilon H$.

We now decompose the inner sum over $k$, $\sum_{k=1}^{K}$, to $\sum_{k=1}^{\tau(s,a)-1}$ and $\sum_{k=\tau(s,a)}^{K}$. The first part is bounded by:

$$
\sum_{(s,a)\in\mathcal{I}_\epsilon} \sum_{k=1}^{\tau_p(s,a)-1} \sum_{p=1}^{M} \rho_p^k(s,a) \left( H \wedge \sqrt{\frac{(1+\mathrm{var}_{\mathbb{P}_p(\cdot|s,a)}[V_p^{\pi^k}]) L(n^k(s,a))}{n^k(s,a)}} \right) \leq \sum_{(s,a)\in\mathcal{I}_\epsilon} \sum_{k=1}^{\tau_p(s,a)-1} \sum_{p=1}^{M} \rho_p^k(s,a) H \leq SAHN_1,
$$

which is $\lesssim MHSA \ln\left( \frac{SAK}{\delta} \right)$.

For the second part,

$$\sum_{(s,a)\in\mathcal{I}_\epsilon}\sum_{k=\tau(s,a)}^{K}\sum_{p=1}^{M}\rho_p^k(s,a)\left(H\wedge\sqrt{\frac{(1+\mathrm{var}_{\mathbb{P}_p(\cdot|s,a)}[V_p^{\pi^k}])L(n^k(s,a))}{n^k(s,a)}}\right)$$

$$\lesssim\sum_{(s,a)\in\mathcal{I}_\epsilon}\sum_{k=\tau(s,a)}^{K}\sum_{p=1}^{M}\rho_p^k(s,a)\sqrt{\frac{(1+\mathrm{var}_{\mathbb{P}_p(\cdot|s,a)}[V_p^{\pi^k}])L(\bar{n}^k(s,a))}{\bar{n}^k(s,a)}}$$

$$\lesssim\sqrt{\sum_{(s,a)\in\mathcal{I}_\epsilon}\sum_{k=\tau(s,a)}^{K}\sum_{p=1}^{M}\rho_p^k(s,a)\cdot\frac{L(\bar{n}^k(s,a))}{\bar{n}^k(s,a)}}\cdot\sqrt{\sum_{(s,a)\in\mathcal{I}_\epsilon}\sum_{k=1}^{K}\sum_{p=1}^{M}\rho_p^k(s,a)\left(1+\mathrm{var}_{\mathbb{P}_p(\cdot|s,a)}[V_p^{\pi^k}]\right)},$$

where the first inequality is by dropping the "$H\wedge$" operator; the second inequality is by Cauchy-Schwarz.

We bound each factor as follows: for the first factor,

$$\sum_{(s,a)\in\mathcal{I}_\epsilon}\sum_{k=\tau(s,a)}^{K}\sum_{p=1}^{M}\rho_p^k(s,a)\cdot\frac{L(\bar{n}^k(s,a))}{\bar{n}^k(s,a)}=\sum_{(s,a)\in\mathcal{I}_\epsilon}\sum_{k=\tau(s,a)}^{K}\rho^k(s,a)\cdot\frac{L(\bar{n}^k(s,a))}{\bar{n}^k(s,a)}$$

$$\leq L(MK)\sum_{(s,a)\in\mathcal{I}_\epsilon}\sum_{k=\tau(s,a)}^{K}\frac{\rho^k(s,a)}{\bar{n}^k(s,a)}$$

$$\leq\sum_{(s,a)\in\mathcal{I}_\epsilon}L(MK)\cdot\int_1^{\bar{n}^K(s,a)}\frac{1}{u}du$$

$$\leq|\mathcal{I}_\epsilon|\,L(MK)^2\lesssim|\mathcal{I}_\epsilon|\left(\ln\left(\frac{MSAK}{\delta}\right)\right)^2,$$

where the first inequality is because $L$ is monotonically increasing, and $\bar{n}^k(s,a)\leq MK$; the second inequality is from the observation that $\rho^k(s,a)\in[0,M]$, $\bar{n}^k(s,a)\geq 2M$, and $u\mapsto\frac{1}{u}$ is monotonically decreasing; the last two inequalities are by algebra.

For the second factor,

$$\sum_{(s,a)\in\mathcal{I}_\epsilon}\sum_{k=1}^{K}\sum_{p=1}^{M}\rho_p^k(s,a)\left(1+\mathrm{var}_{\mathbb{P}_p(\cdot|s,a)}[V_p^{\pi^k}]\right)\lesssim MKH+\sum_{p=1}^{M}\sum_{k=1}^{K}\sum_{(s,a)\in\mathcal{S}\times\mathcal{A}}\rho_p^k(s,a)\mathrm{var}_{\mathbb{P}_p(\cdot|s,a)}[V_p^{\pi^k}]$$

$$\lesssim MKH+\sum_{p=1}^{M}\sum_{k=1}^{K}\mathrm{Var}\left[\sum_{h=1}^{H}r_{h,p}^k\mid\pi^k(p)\right]$$

$$\lesssim MKH^2.$$

(52)

where the first inequality is by the fact that $\rho_p^k$ are probability distributions over every layer $h\in[H]$; the last two inequalities are by a law of total variance identity (see, e.g., [3, Equation (26)]). To summarize, the second part is at most

$$\sum_{(s,a)\in\mathcal{I}_\epsilon}\sum_{k=\tau(s,a)}^{K}\sum_{p=1}^{M}\rho_p^k(s,a)\left(H\wedge\sqrt{\frac{(1+\mathrm{var}_{\mathbb{P}_p(\cdot|s,a)}[V_p^{\pi^k}])L(n^k(s,a))}{n^k(s,a)}}\right)\lesssim\sqrt{MKH^2|\mathcal{I}_\epsilon|}\ln\left(\frac{MSAK}{\delta}\right).$$

Combining the bounds for the first and the second parts, we have:

$$\sum_{(s,a)\in\mathcal{I}_\epsilon}\sum_{k,p}\rho_p^k(s,a)\,\mathrm{clip}\left(B^{k,\mathrm{lead}}(s,a),\breve{\mathrm{gap}}_p(s,a)\right)\lesssim\left(\sqrt{MKH^2|\mathcal{I}_\epsilon|}+MHSA\right)\ln\left(\frac{MSAK}{\delta}\right).$$

**Case 2:** $(s, a) \notin \mathcal{I}_\epsilon$. We simplify the corresponding term as follows:

$$\sum_{(s,a) \notin \mathcal{I}_\epsilon} \sum_{k,p} \rho_p^k(s,a) \operatorname{clip}\left(B^{k,\mathrm{lead}}(s,a), \mathrm{g\breve{a}p}_p(s,a)\right)$$

$$\lesssim \sum_{(s,a) \notin \mathcal{I}_\epsilon} \sum_{k,p} \rho_p^k(s,a) \operatorname{clip}\left(H \wedge \left(\sqrt{\frac{\left(1 + \operatorname{var}_{\mathbb{P}_p(\cdot|s,a)}[V_p^{\pi^k}]\right) L(n_p^k(s,a))}{n_p^k(s,a)}}\right), \frac{\mathrm{g\breve{a}p}_p(s,a)}{4H}\right)$$

$$\lesssim \sum_{(s,a) \notin \mathcal{I}_\epsilon} \sum_{k,p} \left(H \wedge \sqrt{\frac{\left(1 + \operatorname{var}_{\mathbb{P}_p(\cdot|s,a)}[V_p^{\pi^k}]\right) L(n_p^k(s,a))}{n_p^k(s,a)}}\right)$$

For each $p$ and $(s, a)$, we now decompose the inner sum over $k$, $\sum_{k=1}^K$, to $\sum_{k=1}^{\tau_p(s,a)-1}$ and $\sum_{k=\tau_p(s,a)}^K$. The first part is bounded by:

$$\sum_{(s,a) \notin \mathcal{I}_\epsilon} \sum_{p=1}^M \sum_{k=1}^{\tau_p(s,a)-1} \rho_p^k(s,a) \left(H \wedge \sqrt{\frac{(1 + \operatorname{var}_{\mathbb{P}_p(\cdot|s,a)}[V_p^{\pi^k}]) L(n_p^k(s,a))}{n_p^k(s,a)}}\right) \leq \sum_{(s,a) \notin \mathcal{I}_\epsilon} \sum_{p=1}^M \sum_{k=1}^{\tau_p(s,a)-1} \rho_p^k(s,a) H$$

$$\leq MHSAN_2,$$

which is $\lesssim MHSA \ln\left(\frac{MSAK}{\delta}\right)$.

For the second part,

$$\sum_{(s,a) \notin \mathcal{I}_\epsilon} \sum_{p=1}^M \sum_{k=\tau_p(s,a)}^K \rho_p^k(s,a) \left(H \wedge \sqrt{\frac{\left(1 + \operatorname{var}_{\mathbb{P}_p(\cdot|s,a)}[V_p^{\pi^k}]\right) L(n_p^k(s,a))}{n_p^k(s,a)}}\right)$$

$$\lesssim \sum_{(s,a) \notin \mathcal{I}_\epsilon} \sum_{p=1}^M \sum_{k=\tau_p(s,a)}^K \rho_p^k(s,a) \sqrt{\frac{\left(1 + \operatorname{var}_{\mathbb{P}_p(\cdot|s,a)}[V_p^{\pi^k}]\right) L(\bar{n}_p^k(s,a))}{\bar{n}_p^k(s,a)}}$$

$$\leq \sqrt{\sum_{(s,a) \notin \mathcal{I}_\epsilon} \sum_{p=1}^M \sum_{k=\tau_p(s,a)}^K \rho_p^k(s,a) \cdot \frac{L(\bar{n}_p^k(s,a))}{\bar{n}_p^k(s,a)}} \cdot \sqrt{\sum_{(s,a) \notin \mathcal{I}_\epsilon} \sum_{k=1}^K \sum_{p=1}^M \rho_p^k(s,a) \left(1 + \operatorname{var}_{\mathbb{P}_p(\cdot|s,a)}[V_p^{\pi^k}]\right)}$$

We bound each factor as follows: for the first factor,

$$\sum_{(s,a) \notin \mathcal{I}_\epsilon} \sum_{p=1}^M \sum_{k=\tau_p(s,a)}^K \rho_p^k(s,a) \cdot \frac{L(\bar{n}^k(s,a))}{\bar{n}^k(s,a)} \leq L(K) \cdot \sum_{(s,a) \notin \mathcal{I}_\epsilon} \sum_{p=1}^M \sum_{k=\tau_p(s,a)}^K \frac{\rho_p^k(s,a)}{\bar{n}^k(s,a)}$$

$$\leq L(K) \cdot \sum_{(s,a) \notin \mathcal{I}_\epsilon} \sum_{p=1}^M \int_1^{\bar{n}_p^K(s,a)} \frac{1}{u} du$$

$$\leq \left|\mathcal{I}_\epsilon^C\right| ML(K)^2 \leq \left|\mathcal{I}_\epsilon^C\right| M \left(\ln\left(\frac{MSAK}{\delta}\right)\right)^2.$$

where the first inequality is because $L$ is monotonically increasing, and $\bar{n}_p^k(s,a) \leq K$; the second inequality is from the observation that $\rho^k(s,a) \in [0, 1]$, $\bar{n}^k(s,a) \geq 2$, and $u \mapsto \frac{1}{u}$ is monotonically decreasing; the last two inequalities are by algebra.

The second factor is again bounded by (52). Therefore, the second part of the sum is at most

$$\sum_{(s,a)\notin\mathcal{I}_\epsilon}\sum_{p=1}^{M}\sum_{k=\tau_p(s,a)}^{K}\rho_p^k(s,a)\left(H\wedge\sqrt{\frac{\left(1+\text{var}_{\mathbb{P}_p(\cdot|s,a)}[V_p^{\pi^k}]\right)L(n_p^k(s,a))}{n_p^k(s,a)}}\right)$$

$$\leq\left(M\sqrt{KH^2|\mathcal{I}_\epsilon^C|}+MHSA\right)\ln\left(\frac{MSAK}{\delta}\right).$$

Combining the bounds for the first and the second parts, we have:

$$\sum_{(s,a)\notin\mathcal{I}_\epsilon}\sum_{k,p}\rho_p^k(s,a)\,\text{clip}\left(B^{k,\text{lead}}(s,a),\check{\text{gap}}_p(s,a)\right)\lesssim\left(M\sqrt{KH^2|\mathcal{I}_\epsilon^C|}+MHSA\right)\ln\left(\frac{MSAK}{\delta}\right).$$

Now, combining the bounds for cases 1 and 2, we have that

$$(A)\leq\left(\sqrt{MKH^2|\mathcal{I}_\epsilon|}+M\sqrt{KH^2|\mathcal{I}_\epsilon^C|}+MHSA\right)\cdot\ln\left(\frac{MSAK}{\delta}\right).\tag{53}$$

In conclusion, by the regret decomposition Equation (50), and Equations (53) and (51), we have:

$$\text{Reg}(K)\leq\left(\sqrt{MH^2|\mathcal{I}_\epsilon|\,K}+M\sqrt{H^2|\mathcal{I}_\epsilon^C|\,K}+MH^4S^2A\ln\left(\frac{MSAK}{\delta}\right)\right)\ln\left(\frac{MSAK}{\delta}\right).$$

$\square$

### C.4.2 Proof of Theorem 6

*Proof of Theorem 6.* From Lemma 20, we have that when $E$ happens,

$$\text{Reg}(K)=\sum_{p=1}^{M}\text{Reg}(K,p)$$

$$\leq\underbrace{\sum_{s,a}\sum_{k,p}\rho_p^k(s,a)\,\text{clip}\left(B^{k,\text{lead}}(s,a),\check{\text{gap}}_p(s,a)\right)}_{(A)}+\underbrace{H\sum_{s,a}\sum_{k,p}\rho_p^k(s,a)\,\text{clip}\left(B^{k,\text{fut}}(s,a),\frac{\text{gap}_{p,\min}}{8SAH^2}\right)}_{(B)},$$

We focus on each term separately. We directly use Lemma 22 to bound term (B) as:

$$H\sum_{s,a}\sum_{k,p}\rho_p^k(s,a)\,\text{clip}\left(B^{k,\text{fut}}(s,a),\frac{\text{gap}_{p,\min}}{8SAH^2}\right)\lesssim MH^4S^2A\ln\left(\frac{MSAK}{\delta}\right)\cdot\ln\frac{MSA}{\text{gap}_{\min}}.\tag{54}$$

For the $(s,a)$-th term in term (A), we will consider the cases of $(s,a)\in\mathcal{I}_\epsilon$ and $(s,a)\notin\mathcal{I}_\epsilon$ separately.

**Case 1:** $(s, a) \in \mathcal{I}_\epsilon$. In this case, we have that for all $p$, $\text{gǎp}_p(s, a) = \frac{\text{gap}_p(s,a)}{4H} \geq 24\epsilon$. We simplify the corresponding term as follows:

$$\sum_{k,p} \rho_p^k(s, a) \, \text{clip}\left(B^{k,\text{lead}}(s, a), \text{gǎp}_p(s, a)\right)$$

$$\leq \sum_{k=1}^{K} \sum_{p=1}^{M} \rho_p^k(s, a) \, \text{clip}\left(H \wedge \left(5\epsilon + O\left(\sqrt{\frac{(1 + \text{var}_{\mathbb{P}_p(\cdot|s,a)}[V_p^{\pi^k}]) L(n^k(s, a))}{n^k(s, a)}}\right)\right), \frac{\min_p \text{gap}_p(s, a)}{4H}\right)$$

$$\leq \sum_{k=1}^{K} \rho^k(s, a) \, \text{clip}\left(H \wedge \left(5\epsilon + O\left(\sqrt{\frac{H^2 L(n^k(s, a))}{n^k(s, a)}}\right)\right), \frac{\min_p \text{gap}_p(s, a)}{4H}\right)$$

$$\leq \sum_{k=1}^{k} \rho^k(s, a) \left(H \wedge \text{clip}\left(5\epsilon + O\left(\sqrt{\frac{H^2 L(n^k(s, a))}{n^k(s, a)}}\right), \frac{\min_p \text{gap}_p(s, a)}{4H}\right)\right)$$

$$\lesssim \sum_{k=1}^{K} \rho^k(s, a) \left(H \wedge \text{clip}\left(\sqrt{\frac{H^2 L(n^k(s, a))}{n^k(s, a)}}, \frac{\min_p \text{gap}_p(s, a)}{16H}\right)\right)$$

where the first inequality is by the definition of $B^{k,\text{lead}}$; the second inequality is from that $\text{var}_{\mathbb{P}_p(\cdot|s,a)}[V_p^{\pi^k}] \leq H^2$; the third inequality is from that $\text{clip}(A \wedge B, C) \leq A \wedge \text{clip}(B, C)$; the third inequality uses Lemma 27 with $a_1 = 5\epsilon$, $a_2 = \sqrt{\frac{H^2 L(n^k(s,a))}{n^k(s,a)}}$, and $\Delta = \frac{\min_p \text{gap}_p(s,a)}{4H}$, along with the observation that $\text{clip}(5\epsilon, \frac{\min_p \text{gap}_p(s,a)}{16H}) = 0$, since for all $(s, a) \in \mathcal{I}_\epsilon$ and all $p \in [M]$, $\text{gap}_p(s, a) \geq 96\epsilon H$.

We now decompose the inner sum over $k$, $\sum_{k=1}^{K}$, to $\sum_{k=1}^{\tau(s,a)-1}$ and $\sum_{k=\tau(s,a)}^{K}$. The first part's contribution is at most $N_1 \cdot H \lesssim MH \ln\left(\frac{SAK}{\delta}\right)$. For the second part, its contribution is at most:

$$\sum_{k=\tau(s,a)}^{K} \rho^k(s, a) \left(H \wedge \text{clip}\left(\sqrt{\frac{H^2 L(n^k(s, a))}{n^k(s, a)}}, \frac{\min_p \text{gap}_p(s, a)}{16H}\right)\right)$$

$$\lesssim MH + \int_{1}^{\bar{n}^K(s,a)} \left(H \wedge \text{clip}\left(\sqrt{\frac{H^2 L(u)}{u}}, \frac{\min_p \text{gap}_p(s, a)}{16H}\right)\right) du$$

$$\lesssim MH + \frac{H^3}{\min_p \text{gap}_p(s, a)} \ln\left(\frac{MSAK}{\delta}\right)$$

where the second inequality is from Lemma 28 with $f_{\max} = H$, $C = H^2$, $\Delta = \frac{\min_p \text{gap}_p(s,a)}{16H}$, $N = MSA$, $\xi = \delta$, $\Gamma = 1$, $n = \bar{n}^K(s, a) \leq K$. In summary, for all $(s, a) \in \mathcal{I}_\epsilon$,

$$\sum_{k,p} \rho_p^k(s, a) \, \text{clip}\left(B^{k,\text{lead}}(s, a), \text{gǎp}_p(s, a)\right) \leq \left(MH + \frac{H^3}{\min_p \text{gap}_p(s, a)}\right) \ln\left(\frac{MSAK}{\delta}\right).$$

**Case 2:** $(s, a) \notin \mathcal{I}_\epsilon$. In this case, for each $p \in [M]$, we simplify the corresponding term as follows:

$$\sum_{k} \rho_p^k(s, a) \, \text{clip}\left(B^{k,\text{lead}}(s, a), \text{gǎp}_p(s, a)\right)$$

$$\lesssim \sum_{k=1}^{K} \rho_p^k(s, a) \left(H \wedge \text{clip}\left(\sqrt{\frac{H^2 L(n_p^k(s, a))}{n_p^k(s, a)}}, \frac{\text{gǎp}_p(s, a)}{16H}\right)\right)$$

We now decompose the inner sum over $k$, $\sum_{k=1}^{K}$, to $\sum_{k=1}^{\tau_p(s,a)-1}$ and $\sum_{k=\tau_p(s,a)}^{K}$. The first part's contribution is at most $N_2 \cdot H \lesssim H \ln\left(\frac{MSAK}{\delta}\right)$.

For the second part, its contribution is at most:

$$\sum_{k=\tau_p(s,a)}^{K} \rho_p^k(s,a) \left( H \wedge \text{clip}\left( \sqrt{\frac{H^2 L(n^k(s,a))}{n^k(s,a)}}, \frac{\text{gǎp}_p(s,a)}{16H} \right) \right)$$

$$\lesssim H + \int_{1}^{\bar{n}_p^K(s,a)} \left( H \wedge \text{clip}\left( \sqrt{\frac{H^2 L(u)}{u}}, \frac{\text{gǎp}_p(s,a)}{16H} \right) \right) du$$

$$\lesssim H + \frac{H^3}{\text{gǎp}_p(s,a)} \ln\left( \frac{MSAK}{\delta} \right)$$

where the second inequality is from Lemma 28 with $f_{\max} = H$, $C = H^2$, $\Delta = \frac{\text{gǎp}_p(s,a)}{16H}$, $N = MSA$, $\xi = \delta$, $\Gamma = 1$, $n = \bar{n}_p^K(s,a) \leq K$. In summary, for any $(s,a) \in \mathcal{I}_\epsilon^C$ and $p \in [M]$,

$$\sum_{k} \rho_p^k(s,a) \, \text{clip}\left( B^{k,\text{lead}}(s,a), \text{gǎp}_p(s,a) \right) \lesssim \left( H + \frac{H^3}{\text{gǎp}_p(s,a)} \right) \ln\left( \frac{MSAK}{\delta} \right),$$

summing over $p$, we get:

$$\sum_{k,p} \rho_p^k(s,a) \, \text{clip}\left( B^{k,\text{lead}}(s,a), \text{gǎp}_p(s,a) \right) \lesssim \left( MH + \sum_{p=1}^{M} \frac{H^3}{\text{gǎp}_p(s,a)} \right) \ln\left( \frac{MSAK}{\delta} \right),$$

In summary, combining the regret bounds of cases 1 and 2 for term $(A)$, along with Equation (54) for term $(B)$, and observe that $\text{gǎp}_p(s,a) = \text{gap}_{p,\min}$ if $(s,a) \in Z_{p,\text{opt}}$, and $\text{gǎp}_p(s,a) = \text{gap}_p(s,a)$ otherwise, we have that on event $E$, MULTI-TASK-EULER satisfies:

$$\text{Reg}(K) \lesssim \ln\left( \frac{MSAK}{\delta} \right) \left( \sum_{p \in [M]} \left( \sum_{(s,a) \in Z_{p,\text{opt}}} \frac{H^3}{\text{gap}_{p,\min}} + \sum_{(s,a) \in (\mathcal{I}_\epsilon \cup Z_{p,\text{opt}})^C} \frac{H^3}{\text{gap}_p(s,a)} \right) + \right.$$

$$\left. \sum_{(s,a) \in \mathcal{I}_\epsilon} \frac{H^3}{\min_p \text{gap}_p(s,a)} \right) + \ln\left( \frac{MSAK}{\delta} \right) \cdot MS^2AH^4 \ln \frac{MSA}{\text{gap}_{\min}}. \;\square$$

**Lemma 22** (Bounding the lower order terms). *If $E$ happens, then*

$$\sum_{s,a} \sum_{k,p} \rho_p^k(s,a) \, \text{clip}\left( B^{k,\text{fut}}(s,a), \frac{\text{gap}_{p,\min}}{8SAH^2} \right) \lesssim MH^3 S^2 A \ln\left( \frac{MSAK}{\delta} \right) \left( \ln\left( \frac{MSAK}{\delta} \right) \wedge \ln\left( \frac{MSA}{\text{gap}_{\min}} \right) \right).$$

*Proof.* We expand the left hand side using the definition of $B^{k,\text{fut}}$, and the fact that $\text{gap}_{p,\min} \geq \text{gap}_{\min}$:

$$\sum_{k=1}^{K} \rho_p^k(s,a) \, \text{clip}\left( B^{k,\text{fut}}(s,a), \frac{\text{gap}_{p,\min}}{8SAH^2} \right) \qquad (55)$$

$$\lesssim \sum_{k=1}^{K} \rho_p^k(s,a) \left( H^3 \wedge \text{clip}\left( \frac{H^3 SL(n_p^k(s,a))}{n_p^k(s,a)}, \frac{\text{gap}_{\min}}{8SAH^2} \right) \right) \qquad (56)$$

We now decompose the sum $\sum_{k=1}^{K}$ to $\sum_{k=1}^{\tau_p(s,a)-1}$ and $\sum_{k=\tau_p(s,a)}^{K}$. The first part can be bounded by

$$\sum_{k=1}^{\tau_p(s,a)-1} \rho_p^k(s,a) \left( H^3 \wedge \text{clip}\left( \frac{H^3 SL(n_p^k(s,a))}{n_p^k(s,a)}, \frac{\text{gap}_{\min}}{8SAH^2} \right) \right) \leq \sum_{k=1}^{\tau_p(s,a)-1} H^3 \rho_p^k(s,a) \leq H^3 N_2,$$

which is at most $O\left(H^3 \cdot \ln\left(\frac{MSAK}{\delta}\right)\right)$. For the second part, it can be bounded by:

$$\sum_{k=\tau_p(s,a)}^{K} \rho_p^k(s,a)\left(H^3 \wedge \mathrm{clip}\left(\frac{H^3 SL(n_p^k(s,a))}{n_p^k(s,a)}, \frac{\mathrm{gap_{min}}}{8SAH^2}\right)\right)$$

$$\leq H^3 \cdot 1 + \int_1^{\bar{n}_p^K(s,a)}\left(H^3 \wedge \mathrm{clip}\left(\frac{H^3 SL(u)}{u}, \frac{\mathrm{gap_{min}}}{8SAH^2}\right)\right) du$$

$$\lesssim H^3 + H^3 \ln\left(\frac{MSA}{\delta}\right) + H^3 S \ln\left(\frac{MSAK}{\delta}\right)\left(\ln\left(\frac{MSAK}{\delta}\right) \wedge \ln\left(\frac{MHSA}{\mathrm{gap_{min}}}\right)\right),$$

where the second inequality is from Lemma 28 with $f_{\max} = H^3$, $C = H^3 S$, $\Delta = \frac{\mathrm{gap_{min}}}{8SAH^2}$, $N = MSA$, $\xi = \delta$, $\Gamma = 1$, $n = \bar{n}_p^K(s,a) \leq K$. In addition, observe that $H \leq S$ by our layered MDP assumption, we have

$$\sum_k \rho_p^k(s,a)\,\mathrm{clip}\left(B^{k,\mathrm{lead}}(s,a), \frac{\mathrm{gap_{min}}}{8SAH^2}\right) \lesssim H^3 S \ln\left(\frac{MSAK}{\delta}\right)\left(\ln\left(\frac{MSAK}{\delta}\right) \wedge \ln\left(\frac{MSA}{\mathrm{gap_{min}}}\right)\right)$$

Summing over $s \in \mathcal{S}$, $a \in \mathcal{A}$, and $p \in [M]$, we get

$$\sum_{s,a}\sum_{k,p} \rho_p^k(s,a)\,\mathrm{clip}\left(B^{k,\mathrm{lead}}(s,a), \frac{\mathrm{gap_{min}}}{8SAH^2}\right) \lesssim MH^3 S^2 A \ln\left(\frac{MSAK}{\delta}\right)\left(\ln\left(\frac{MSAK}{\delta}\right) \wedge \ln\left(\frac{MSA}{\mathrm{gap_{min}}}\right)\right).$$

$\square$

### C.5 Miscellaneous lemmas

This subsection collects a few miscellaneous lemmas used throughout the upper bound proofs.

**Lemma 23** ([36], Lemma F.5). *For random variables $X$ and $Y$, $\left|\sqrt{\mathrm{var}[X]} - \sqrt{\mathrm{var}[Y]}\right| \leq \sqrt{\mathbb{E}\left[(X-Y)^2\right]}$.*

**Lemma 24.** *Suppose distributions $P$ and $Q$ are supported over $[0, B]$, and $\|P - Q\|_1 \leq \epsilon \leq 2$. Then:*

$$\left|\mathbb{E}_{X\sim P}[X] - \mathbb{E}_{X\sim Q}[X]\right| \leq B\epsilon,$$
$$\left|\mathrm{var}_{X\sim P}[X] - \mathrm{var}_{X\sim Q}[X]\right| \leq 3B^2\epsilon.$$

*Proof.* First,

$$\left|\mathbb{E}_{X\sim P}[X] - \mathbb{E}_{X\sim Q}[X]\right| = \left|\int_0^B x(p_X(x) - q_X(x))dx\right| \leq \int_0^B |x||p_X(x) - q_X(x)|\,dx \leq B\|P-Q\|_1 \leq B\epsilon.$$

Second, observe that

$$\left|\mathbb{E}_{X\sim P}[X^2] - \mathbb{E}_{X\sim Q}[X^2]\right| \leq B^2\epsilon.$$

Meanwhile,

$$\left|(\mathbb{E}_{X\sim P}[X])^2 - (\mathbb{E}_{X\sim Q}[X])^2\right| \leq \left|\mathbb{E}_{X\sim P}[X] - \mathbb{E}_{X\sim Q}[X]\right|\cdot\left|\mathbb{E}_{X\sim P}[X] + \mathbb{E}_{X\sim Q}[X]\right| \leq 2B\cdot B\epsilon = 2B^2\epsilon.$$

Combining the above, we have

$$\left|\mathrm{var}_{X\sim P}[X] - \mathrm{var}_{X\sim Q}[X]\right| \leq 3B^2\epsilon.$$

$\square$

**Lemma 25.** *For $A, B, C, D, E, F \geq 0$:*

1. If $|A - B| \leq \sqrt{BC} + C$, then we have $\left|\sqrt{A} - \sqrt{B}\right| \leq 2\sqrt{C}$.

2. If $D \leq E + F\sqrt{D}$, then $\sqrt{D} \leq \sqrt{E} + F$.

*Proof.*      1. First, $A - B \leq |A - B| \leq \sqrt{BC} + C$; this implies that $A \leq B + 2\sqrt{BC} + C$, and therefore $\sqrt{A} \leq \sqrt{B} + \sqrt{C}$.

On the other hand, $B \leq A + C + \sqrt{BC}$; therefore, applying item 1 with $D = B$, $E = A + C$, and $F = \sqrt{C}$, we have $\sqrt{B} \leq \sqrt{A + C} + \sqrt{C} \leq \sqrt{A} + 2\sqrt{C}$.

2. The roots of $x^2 - Fx - E = 0$ are $\frac{F \pm \sqrt{F^2 + 4E}}{2}$, and therefore $D$ must satisfy $\sqrt{D} \leq \frac{F + \sqrt{F^2 + 4E}}{2} \leq \frac{F + F + 2\sqrt{E}}{2} = F + \sqrt{E}$.

$\square$

**Lemma 26.** *For $a \geq 0$, $1 \wedge (a + \sqrt{a}) \leq 1 \wedge 2\sqrt{a}$.*

*Proof.* We consider the cases of $a \geq 1$ and $a < 1$ respectively. If $a \geq 1$, LHS $= 1 =$ RHS. Otherwise, $a \leq 1$; in this case, LHS $= 1 \wedge (a + \sqrt{a}) \leq 1 \wedge (\sqrt{a} + \sqrt{a}) =$ RHS. $\square$

**Lemma 27** (Special case of [36], Lemma B.5)**.** *For $a_1, a_2, \Delta \geq 0$, $\mathrm{clip}(a_1 + a_2, \Delta) \leq 2\,\mathrm{clip}(a_1, \Delta/4) + 2\,\mathrm{clip}(a_2, \Delta/4)$.*

**Lemma 28** (Integral calculation, [36], Lemma B.9)**.** *Let $f(u) \leq \min(f_{\max}, \mathrm{clip}(g(u), \Delta))$, where $\Delta \in [0, \Gamma]$, and $g(u)$ is nonincreasing. Let $N \geq 1$ and $\xi \in (0, \frac{1}{2})$. Then:*

1. *If $g(u) \lesssim \sqrt{\frac{C \log \frac{Nu}{\xi}}{u}}$ for some $C > 0$ such that $\ln C \lesssim \ln N$, then*

$$\int_{\Gamma}^{n} f(u/4)du \lesssim \sqrt{Cn \ln \frac{Nn}{\xi}} \wedge \frac{C}{\Delta} \ln\left(\frac{Nn}{\xi}\right).$$

2. *If $g(u) \lesssim \frac{C \ln \frac{Nu}{\xi}}{u}$ for some $C > 0$ such that $\ln C \lesssim \ln N$, then*

$$\int_{\Gamma}^{n} f(u/4)du \lesssim f_{\max} \ln \frac{N}{\xi} + C \ln \frac{Nn}{\xi} \cdot \left(\ln \frac{Nn}{\xi} \wedge \ln \frac{N\Gamma}{\Delta}\right).$$

# D    Proof of the Lower Bounds

## D.1    Auxiliary Lemmas

**Lemma 29** (Regret decomposition, [36], Section H.2)**.** *For any MPERL problem instance and any algorithm, we have*

$$\mathbb{E}\left[\mathrm{Reg}(K)\right] \geq \sum_{p=1}^{M} \sum_{(s,a) \in \mathcal{S}_1 \times \mathcal{A}} \mathbb{E}\left[n_p^{K+1}(s,a)\right] \mathrm{gap}_p(s,a), \tag{57}$$

*where we recall that $n_p^{K+1}(s,a)$ is the number of visits of $(s,a)$ by player $p$ at the beginning of the $(K+1)$-th episode (after the first $K$ episodes). Furthermore, for any $(s,a) \in \mathcal{S}_1 \times \mathcal{A}$, we have*

$$\sum_{p=1}^{M} \mathbb{E}\left[n_p^{K+1}(s,a)\right] \mathrm{gap}_p(s,a) \geq \mathbb{E}\left[n^{K+1}(s,a)\right]\left(\min_{p \in [M]} \mathrm{gap}_p(s,a)\right), \tag{58}$$

*where we recall that $n^{K+1}(s,a) = \sum_{p=1}^{M} n_p^{K+1}(s,a)$.*

*Proof.* Eq. (58) follows straightforwardly from the fact that for every $(s,a,p) \in \mathcal{S}_1 \times \mathcal{A} \times [M]$, $\min_{p' \in [M]} \mathrm{gap}_{p'}(s,a) \leq \mathrm{gap}_p(s,a)$.

We now prove Eq. (57). Let $\pi_p^k$ denote $\pi^k(p)$. We have

$$
\begin{aligned}
\mathbb{E}\left[\text{Reg}(K)\right] &= \mathbb{E}\left[\sum_{p=1}^{M}\sum_{k=1}^{K}\sum_{s\in\mathcal{S}_1} p_0(s_{1,p}^k = s)\left(V_p^\star(s) - V_p^{\pi_p^k}(s)\right)\right] \\
&\geq \mathbb{E}\left[\sum_{p=1}^{M}\sum_{k=1}^{K}\sum_{s\in\mathcal{S}_1} p_0(s_{1,p}^k = s)\left(V_p^\star(s) - Q_p^\star(s, \pi_p^k(s))\right)\right] \\
&= \mathbb{E}\left[\sum_{p=1}^{M}\sum_{k=1}^{K}\sum_{s\in\mathcal{S}_1} p_0(s)\text{gap}_p(s, \pi_p^k(s))\right] \\
&= \mathbb{E}\left[\sum_{p=1}^{M}\sum_{k=1}^{K}\sum_{s\in\mathcal{S}_1} \mathbf{1}\left(s_{1,p}^k = s\right)\text{gap}_p(s, \pi_p^k(s))\right] \\
&= \mathbb{E}\left[\sum_{p=1}^{M}\sum_{k=1}^{K}\sum_{(s,a)\in\mathcal{S}_1\times\mathcal{A}} \mathbf{1}\left(s_{1,p}^k, \pi_p^k(s) = (s,a)\right)\text{gap}_p(s,a)\right] \\
&= \sum_{p=1}^{M}\sum_{(s,a)\in\mathcal{S}_1\times\mathcal{A}} \mathbb{E}\left[n_p^K(s,a)\right]\text{gap}_p(s,a)
\end{aligned}
$$

$$(59)$$

where the first equality is from the definition of collective regret; the first inequality is from the simple fact that $V_p^\pi(s) = Q_p^\pi(s, \pi(s)) \leq Q_p^\star(s, \pi(s))$ for any policy $\pi$; the second equality is from the definition of suboptimality gaps; and the third equality is from the basic observation that $s_{1,p}^k \sim p_0$. $\qquad\square$

**Lemma 30** (Divergence decomposition [23, 44]). *For two MPERL problem instances, $\mathfrak{M}$ and $\mathfrak{M}'$, which only differ in the transition probabilities $\left\{\mathbb{P}_p(\cdot \mid s,a)\right\}_{p\in[M],(s,a)\in\mathcal{S}\times\mathcal{A}}$, and for a fixed algorithm, let $\mathbb{P}_{\mathfrak{M}}$ and $\mathbb{P}_{\mathfrak{M}'}$ be the probability measures on the outcomes of running the algorithm on $\mathfrak{M}$ and $\mathfrak{M}'$, respectively. Then,*

$$
\text{KL}(\mathbb{P}_{\mathfrak{M}}, \mathbb{P}_{\mathfrak{M}'}) = \sum_{p=1}^{M}\sum_{(s,a)\in\mathcal{S}\times\mathcal{A}} \mathbb{E}_{\mathfrak{M}}\left[n_p^{K+1}(s,a)\right]\text{KL}\left(\mathbb{P}_p^{\mathfrak{M}}(\cdot \mid s,a), \mathbb{P}_p^{\mathfrak{M}'}(\cdot \mid s,a)\right),
$$

*where $\mathbb{P}_p^{\mathfrak{M}}(\cdot \mid s,a)$ and $\mathbb{P}_p^{\mathfrak{M}'}(\cdot \mid s,a)$ are the transition probabilities of the problem instance $\mathfrak{M}$ and $\mathfrak{M}'$, respectively.*

**Lemma 31** (Bretagnolle-Huber inequality, [23], Theorem 14.2). *Let $\mathbb{P}$ and $\mathbb{Q}$ be two distributions on the same measurable space, and $A$ be an event. Then,*

$$
\mathbb{P}(A) + \mathbb{Q}(A^C) \geq \frac{1}{2}\exp\left(-\text{KL}(\mathbb{P}, \mathbb{Q})\right).
$$

**Lemma 32** (see, e.g., [43], Lemma 25). *For any $x, y \in [\frac{1}{4}, \frac{3}{4}]$, $\text{KL}\left(\text{Ber}(x), \text{Ber}(y)\right) \leq 3(x-y)^2$.*

**Lemma 33.** *Let $X$ be a Binomial random variable and $X \sim \text{Bin}(n,p)$, where $n \geq \frac{1}{p}$. Then,*

$$
\mathbb{E}\left[X^{\frac{3}{2}}\right] \leq 2(np)^{\frac{3}{2}}.
$$

*Proof.* Let $Y = X^2$, and $f(Y) = Y^{\frac{3}{4}}$. We have $\mathbb{E}[Y] = \mathbb{E}\left[X^2\right] = \text{var}\left[X\right] + \mathbb{E}\left[X\right]^2 = (np)^2 + np(1-p) \leq (np)^2 + np \leq 2(np)^2$, where the last inequality follows from the assumption that $n \geq \frac{1}{p}$. By Jensen's inequality, we have $\mathbb{E}\left[X^{\frac{3}{2}}\right] = \mathbb{E}\left[f(Y)\right] \leq f\left(\mathbb{E}[Y]\right) \leq \left(2n^2p^2\right)^{\frac{3}{4}} \leq 2(np)^{\frac{3}{2}}$. $\qquad\square$

## D.2 Gap independent lower bounds

**Theorem 34** (Restatement of Theorem 7). *For any $A \geq 2$, $H \geq 2$, $S \geq 4H$, $K \geq SA$, $M \in \mathbb{N}$, and $l, l^C \in \mathbb{N}$ such that $l + l^C = SA$ and $l \leq SA - 4(S + HA)$, there exists some $\epsilon$ such that for any algorithm Alg, there exists an $\epsilon$-MPERL problem instance with $S$ states, $A$ actions, $M$ players and an episode length of $H$ such that $\left| \mathcal{I}_{\frac{\epsilon}{192H}} \right| \geq l$, and*

$$\mathbb{E}\left[ \mathrm{Reg}_{\mathrm{Alg}}(K) \right] \geq \Omega\left( M\sqrt{H^2 l^C K} + \sqrt{MH^2 lK} \right).$$

*Proof.* The construction and techniques in this proof are inspired by [43, Section E.1] and [36].

Fix any algorithm Alg; we consider two cases:

1. $l > Ml^C$;

2. $Ml^C \geq l$.

**Case 1:** $l > Ml^C$. Let $S_1 = S - 2(H - 1)$, and $b = \lceil \frac{l}{S_1} \rceil \geq 1$. Let $\Delta = \sqrt{\frac{l+1}{384MK}}$, and let $\epsilon = \frac{1}{2}H\Delta$. We note that under the assumption that $K \geq SA$, and the observation that $l \leq SA$, we have $\Delta \leq \frac{1}{4}$. We define $(b+1)^{S_1}$ $\epsilon$-MPERL problem instances, each indexed by an element in $[b+1]^{S_1}$. It suffices to show that, on at least one of the problem instances, $\mathbb{E}\left[ \mathrm{Reg}_{\mathrm{Alg}}(K) \right] \geq \Omega\left( \sqrt{MH^2 lK} \right)$.

**Construction.** For $\mathbf{a} = (a_1, \ldots, a_{S_1}) \in [b+1]^{S_1}$, we define the following $\epsilon$-MPERL problem instance, $\mathfrak{M}(\mathbf{a}) = \{\mathcal{M}_p\}_{p=1}^{M}$, with $S$ states, $A$ actions, and an episode length of $H$, such that for each $p \in [M]$, $\mathcal{M}_p$ is constructed as follows:

- $\mathcal{S}_1 = [S_1]$, and $p_0$ is a uniform distribution over the states in $\mathcal{S}_1$.

- For $h \in [2, H]$, $\mathcal{S}_h = \{S_1 + 2h - 3, S_1 + 2h - 2\}$.

- $\mathcal{A} = [A]$.

- For each $(s, a) \in \mathcal{S} \times \mathcal{A}$, the reward distribution $r_p(s, a)$ is a Bernoulli distribution, $\mathrm{Ber}(R_p(s, a))$, and we will specify $R_p(s, a)$ subsequently.

- For each state $s \in [S_1]$,

$$\mathbb{P}_p(S_1 + 1 \mid s, a) = \begin{cases} \frac{1}{2} + \Delta, & \text{if } a = a_s; \\ \frac{1}{2}, & \text{if } a \in [b+1] \setminus a_s; \\ 0, & \text{if } a \notin [b+1]; \end{cases}$$

  and for each $a \in \mathcal{A}$, $\mathbb{P}_p(S_1 + 2 \mid s, a) = 1 - \mathbb{P}_p(S_1 + 1 \mid s, a)$, and $R_p(s, a) = 0$.

- For $h \in [2, H]$, and $a \in \mathcal{A}$, let

  - $\mathbb{P}_p\left(S_1 + 2h - 1 \mid S_1 + 2h - 3, a\right) = 1$, $\mathbb{P}_p\left(S_1 + 2h \mid S_1 + 2h - 3, a\right) = 0$, and $R_p(S_1 + 2h - 3, a) = 1$.
  - $\mathbb{P}_p\left(S_1 + 2h \mid S_1 + 2h - 2, a\right) = 0$, $\mathbb{P}_p\left(S_1 + 2h - 1 \mid S_1 + 2h - 2, a\right) = 1$, and $R_p(S_1 + 2h - 2, a) = 0$.

It can be easily verified that $\mathfrak{M}(\mathbf{a}) = \{\mathcal{M}_p\}_{p=1}^{M}$ is a 0-MPERL problem instance, and hence an $\epsilon$-MPERL problem instance—the reward distributions and the transition probabilities are the same for all players, i.e., for every $p, q \in [M]$, and every $(s, a) \in \mathcal{S} \times \mathcal{A}$,

$$\left| R_p(s, a) - R_q(s, a) \right| = 0 \leq \epsilon, \quad \left| \mathbb{P}_p(\cdot \mid s, a) - \mathbb{P}_q(\cdot \mid s, a) \right| = 0 \leq \frac{\epsilon}{H}.$$

**Suboptimality gaps.** We now calculate the suboptimality gaps of the state-action pairs in the above MDPs. For each $p \in [M]$ and each $(s, a) \in \mathcal{S} \times \mathcal{A}$,

$$\text{gap}_p(s, a) = V_p^\star(s) - Q_p^\star(s, a) = \max_{a'} Q_p^\star(s, a') - Q_p^\star(s, a).$$

In $\mathfrak{M}(\mathbf{a})$, it can be easily observed that for every $p \in [M]$, and every $(s, a) \in (\mathcal{S} \setminus \mathcal{S}_1) \times \mathcal{A}$, $\text{gap}_p(s, a) = 0$. Now, for every $p \in [M]$, $(s, a) \in \mathcal{S}_1 \times \mathcal{A}$, we have

$$\text{gap}_p(s, a) = \max_{a'} Q_p^\star(s, a') - Q_p^\star(s, a) = (H-1)\left(\max_{a'} \mathbb{P}_p(S_1 + 1 \mid s, a') - \mathbb{P}_p(S_1 + 1 \mid s, a)\right).$$

It follows that, for every $p \in [M]$ and every state $s \in [S_1]$,

$$\text{gap}_p(s, a) = \begin{cases} 0, & \text{if } a = a_s; \\ (H-1)\Delta, & \text{if } a \in [b+1] \setminus a_s; \\ (H-1)\left(\frac{1}{2} + \Delta\right), & \text{if } a \notin [b+1]. \end{cases}$$

**Subpar state-action pairs.** It can be verified that in $\mathfrak{M}(\mathbf{a})$, $\left|\mathcal{I}_{\frac{\epsilon}{192H}}\right| \geq l$. Specifically, since $(H-1)\Delta = (H-1)\frac{2\epsilon}{H} \geq \epsilon \geq \frac{\epsilon}{2} = 96H\frac{\epsilon}{192H}$, we have that the number of subpar state-action pairs is at least $S_1 b = S_1 \lceil \frac{l}{S_1} \rceil \geq l$.

It suffices to prove that

$$\mathbb{E}_{\mathbf{a} \sim \text{Unif}\left([b+1]^{S_1}\right)} \mathbb{E}_{\mathfrak{M}(\mathbf{a})}\left[\text{Reg}_{\text{Alg}}(K)\right] \geq \frac{1}{640}\sqrt{MH^2 lK},$$

where we recall that $\mathbf{a} = (a_1, \ldots, a_{S_1})$; furthermore, it suffices to show that, for any $s' \in [S_1]$,

$$\mathbb{E}_{\mathbf{a} \sim \text{Unif}\left([b+1]^{S_1}\right)} \mathbb{E}_{\mathfrak{M}(\mathbf{a})}\left[N^{K+1}(s') - n^{K+1}(s', a_{s'})\right] \geq \frac{MK}{4S_1}, \tag{60}$$

where $N^{K+1}(s') = \sum_{a \in \mathcal{A}} n^{K+1}(s', a)$; this is because it follows from Eq. (60) that

$$\mathbb{E}_{\mathbf{a} \sim \text{Unif}\left([b+1]^{S_1}\right)} \mathbb{E}_{\mathfrak{M}(\mathbf{a})}\left[\text{Reg}_{\text{Alg}}(K)\right] \geq \sum_{s' \in \mathcal{S}_1} (H-1)\frac{\Delta}{4} \cdot \mathbb{E}_{\mathbf{a} \sim \text{Unif}\left([b+1]^{S_1}\right)} \mathbb{E}_{\mathfrak{M}(\mathbf{a})}\left[N^{K+1}(s') - n^{K+1}(s', a_{s'})\right]$$

$$\geq \sum_{s' \in \mathcal{S}_1} \frac{H}{2} \cdot \frac{\Delta}{4} \cdot \frac{MK}{4S_1}$$

$$\geq \frac{1}{640}\sqrt{MH^2 lK},$$

where the first inequality uses Lemma 29 (the regret decomposition lemma).

Without loss of generality, let us choose $s' = 1$. To prove Eq. (60), we use a standard technique and define a set of helper problem instances. Specifically, for any $(a_2, a_3, \ldots, a_{S_1}) \in [b+1]^{S_1-1}$, we define a problem instance $\mathfrak{M}(0, a_2, \ldots, a_{s_1})$ such that it agrees with $\mathfrak{M}(a_1, a_2, \ldots, a_{s_1})$ on everything but $\mathbb{P}_p(\cdot \mid 1, a_1)$'s, i.e., in $\mathfrak{M}(0, a_2, \ldots, a_{s_1})$, for every $p \in [M]$,

$$\mathbb{P}_p(S_1 + 1 \mid 1, a_1) = \frac{1}{2}.$$

Now, for each $(j, a_2, \ldots, a_{s_1}) \in ([0] \cup [b+1]) \times [b+1]^{S_1-1}$, let $\mathbb{P}_{j,a_2,\ldots,a_{S_1}}$ denote the probability measure on the outcomes of running Alg on the problem instance $\mathfrak{M}(j, a_2, \ldots, a_{s_1})$. Further, for each $j \in \{0\} \cup [b+1]$, we define

$$\mathbb{P}_j = \frac{1}{(b+1)^{S_1-1}} \sum_{a_2, \ldots, a_{S_1} \in [b+1]^{S_1-1}} \mathbb{P}_{j,a_2,\ldots,a_{S_1}};$$

and we use $\mathbb{E}_j$ to denote the expectation with respect to $\mathbb{P}_j$.

In subsequent calculations, for any index $m \in \big([0] \cup [b+1]\big) \times [b+1]^{S_1-1}$, we also denote by $\mathbb{P}_m \big(\cdot \mid N^{K+1}(1)\big)$ and $\mathbb{E}_m \big[\cdot \mid N^{K+1}(1)\big]$ the probability and expectation, respectively, conditional on a realization of $N^{K+1}(1)$ under $\mathbb{P}_m$. Observe that, for any $j \in \{0\} \cup [b+1]$,

$$
\begin{aligned}
\mathbb{P}_j(\cdot \mid N^{K+1}(1)) &= \frac{\mathbb{P}_j(\cdot, N^{K+1}(1))}{\mathbb{P}_j(N^{K+1}(1))} \\
&= \frac{\frac{1}{(b+1)^{S_1-1}} \sum_{a_2,\dots,a_{S_1} \in [b+1]^{S_1-1}} \mathbb{P}_{j,a_2,\dots,a_{S_1}}(\cdot, N^{K+1}(1))}{\mathbb{P}_j(N^{K+1}(1))} \\
&= \frac{1}{(b+1)^{S_1-1}} \sum_{a_2,\dots,a_{S_1} \in [b+1]^{S_1-1}} \frac{\mathbb{P}_{j,a_2,\dots,a_{S_1}}(\cdot, N^{K+1}(1))}{\mathbb{P}_{j,a_2,\dots,a_{S_1}}(N^{K+1}(1))} \\
&= \frac{1}{(b+1)^{S_1-1}} \sum_{a_2,\dots,a_{S_1} \in [b+1]^{S_1-1}} \mathbb{P}_{j,a_2,\dots,a_{S_1}}(\cdot \mid N^{K+1}(1)), \qquad (61)
\end{aligned}
$$

where the first equality is from the definition of conditional probability; the second equality is from the definition of $\mathbb{P}_j$; the third equality uses the fact that $\mathbb{P}_j(N^{K+1}(1)) = \mathbb{P}_{j,a_2,\dots,a_{S_1}}(N^{K+1}(1))$ for any $a_2,\dots,a_{S_1}$, which is true because $N^{K+1}(1)$ is independent of $a_2,\dots,a_{S_1}$ conditional on $j$; and the last equality, again, is from the definition of conditional probability.

We have, for each $j \in [b+1]$,

$$
\begin{aligned}
&\mathbb{E}_j\Big[n^{K+1}(1,j) \mid N^{K+1}(1)\Big] - \mathbb{E}_0\Big[n^{K+1}(1,j) \mid N^{K+1}(1)\Big] \\
\leq & N^{K+1}(1) \left\| \mathbb{P}_j\Big(\cdot \mid N^{K+1}(1)\Big) - \mathbb{P}_0\Big(\cdot \mid N^{K+1}(1)\Big) \right\|_1 \\
\leq & N^{K+1}(1) \cdot \frac{1}{(b+1)^{S_1-1}} \sum_{a_2,\dots,a_{S_1} \in [b+1]^{S_1-1}} \left\| \mathbb{P}_{j,a_2,\dots,a_{S_1}}\Big(\cdot \mid N^{K+1}(1)\Big) - \mathbb{P}_{0,a_2,\dots,a_{S_1}}\Big(\cdot \mid N^{K+1}(1)\Big) \right\|_1 \\
\leq & N^{K+1}(1) \cdot \frac{1}{(b+1)^{S_1-1}} \sum_{a_2,\dots,a_{S_1} \in [b+1]^{S_1-1}} \sqrt{2\,\mathrm{KL}\Big(\mathrm{Ber}(\tfrac{1}{2}+\Delta), \mathrm{Ber}(\tfrac{1}{2})\Big) \mathbb{E}_{0,a_2,\dots,a_{S_1}}\Big[n^{K+1}(1,j) \mid N^{K+1}(1)\Big]} \\
\leq & N^{K+1}(1) \cdot \frac{1}{(b+1)^{S_1-1}} \sum_{a_2,\dots,a_{S_1} \in [b+1]^{S_1-1}} \sqrt{6\Delta^2 \mathbb{E}_{0,a_2,\dots,a_{S_1}}\Big[n^{K+1}(1,j) \mid N^{K+1}(1)\Big]} \\
\leq & N^{K+1}(1) \sqrt{(6)\frac{l+1}{384 MK} \cdot \mathbb{E}_0\Big[n^{K+1}(1,j) \mid N^{K+1}(1)\Big]} \\
= & \frac{1}{8} N^{K+1}(1) \sqrt{\frac{l+1}{MK} \cdot \mathbb{E}_0\Big[n^{K+1}(1,j) \mid N^{K+1}(1)\Big]}. \qquad (62)
\end{aligned}
$$

where the first inequality is based on Lemma 24 and the fact that, conditional on $N^{K+1}(1)$, $n^{K+1}(1,j)$ has distribution supported on $[0, N^{K+1}(1)]$; the second inequality follows from Equation (61) and the triangle inequality; the third inequality uses Pinsker's inequality and Lemma 30 (the divergence decomposition lemma); the fourth inequality uses Lemma 32 and the fact that $\Delta \leq \frac{1}{4}$; and the last inequality follows from Jensen's inequality.

Since $N^{K+1}(1)$ has the same distribution under both $\mathbb{P}_0$ and any $\mathbb{P}_j$ (which is $\mathrm{Bin}(K, \frac{1}{S_1})$), taking expectation with respect to $N^{K+1}(1)$, we have that, for any $j \in [b+1]$,

$$
\mathbb{E}_j\Big[n^{K+1}(1,j)\Big] - \mathbb{E}_0\Big[n^{K+1}(1,j)\Big] \leq \mathbb{E}_0\left[ \frac{1}{8} N^{K+1}(1) \sqrt{\frac{l+1}{MK} \cdot \mathbb{E}_0\Big[n^{K+1}(1,j) \mid N^{K+1}(1)\Big]} \right].
$$

In subsequent derivations, we can now avoid bounding the conditional expectation. Specifically, we have

$$\frac{1}{b+1} \sum_{j \in [b+1]} \mathbb{E}_j \left[ n^{K+1}(1,j) \right]$$

$$\leq \frac{1}{b+1} \sum_{j \in [b+1]} \mathbb{E}_0 \left[ n^{K+1}(1,j) \right] + \frac{1}{b+1} \sum_{j \in [b+1]} \mathbb{E}_0 \left[ \frac{1}{8} N^{K+1}(1) \sqrt{\frac{l+1}{MK} \cdot \mathbb{E}_0 \left[ n^{K+1}(1,j) \mid N^{K+1}(1) \right]} \right]$$

$$\leq \frac{1}{b+1} \mathbb{E}_0 \left[ \sum_{j \in [b+1]} n^{K+1}(1,j) \right] + \mathbb{E}_0 \left[ \frac{1}{8} N^{K+1}(1) \sqrt{\frac{l+1}{MK} \cdot \frac{1}{b+1} \sum_{j \in [b+1]} \mathbb{E}_0 \left[ n^{K+1}(1,j) \mid N^{K+1}(1) \right]} \right]$$

$$\leq \frac{1}{b+1} \mathbb{E}_0 \left[ N^{K+1}(1) \right] + \mathbb{E}_0 \left[ \frac{1}{8} \sqrt{\frac{l+1}{MK} \cdot \frac{1}{b+1}} \left( N^{K+1}(1) \right)^{\frac{3}{2}} \right]$$

$$\leq \frac{1}{b+1} \mathbb{E}_0 \left[ N^{K+1}(1) \right] + \frac{1}{8} \sqrt{\frac{S_1}{MK}} \cdot \mathbb{E}_0 \left[ \left( N^{K+1}(1) \right)^{\frac{3}{2}} \right], \tag{63}$$

where the first inequality follows from Eq. (62) and algebra; the second inequality uses linearity of expectation and Jensen's inequality; the third inequality uses the facts that $\sum_{j \in [b+1]} n^{K+1}(1,j) \leq N^{K+1}(1)$ and, for every $z \in [0] \cup [b+1]$,

$$\sum_{j \in [b+1]} \mathbb{E}_z \left[ n^{K+1}(1,j) \mid N^{K+1}(1) \right] \leq \sum_{j \in \mathcal{A}} \mathbb{E}_z \left[ n^{K+1}(1,j) \mid N^{K+1}(1) \right] = N^{K+1}(1);$$

and the last inequality uses the linearity of expectation and the construction that $b = \lceil \frac{l}{S_1} \rceil$, which implies that $l \leq bS_1$ and therefore $l + 1 \leq bS_1 + 1 \leq bS_1 + S_1 = (b+1)S_1$.

It follows from Equation (63) that

$$\frac{1}{b+1} \sum_{j \in [b+1]} \mathbb{E}_j \left[ n^{K+1}(1,j) \right] \leq \frac{1}{b+1} \cdot \frac{MK}{S_1} + \frac{1}{8} \sqrt{\frac{S_1}{MK}} \cdot \mathbb{E}_0 \left[ \left( N^{K+1}(1) \right)^{\frac{3}{2}} \right]$$

$$\leq \frac{MK}{2S_1} + \frac{1}{4} \sqrt{\frac{S_1}{MK} \left( \frac{MK}{S_1} \right)^3}$$

$$\leq \frac{3MK}{4S_1},$$

where the second inequality uses the fact that $\frac{1}{b+1} \leq \frac{1}{2}$ and Lemma 33 under the assumption that $K \geq S_1$.

It then follows that

$$\frac{1}{b+1} \sum_{j \in [b+1]} \mathbb{E}_j \left[ N^{K+1}(1) - n^{K+1}(1,j) \right] \geq \frac{1}{b+1} \sum_{j \in [b+1]} \mathbb{E}_j \left[ N^{K+1}(1) \right] - \frac{3MK}{4S_1} = \frac{MK}{4S_1},$$

and we have

$$\mathbb{E}_{\mathbf{a} \sim \text{Unif}\left( [b+1]^{S_1} \right)} \mathbb{E}_{\mathfrak{M}(\mathbf{a})} \left[ N^{K+1}(1) - n^{K+1}(1,a_1) \right] \geq \frac{MK}{4S_1}.$$

**Case 2: $Ml^C \geq l$.** Again, let $S_1 = S - 2(H-1)$. Let $u = \lceil \frac{l}{S_1} \rceil$ and $v = A - u = A - \lceil \frac{l}{S_1} \rceil$. Furthermore, let $\Delta = \sqrt{\frac{vS_1}{384K}}$, and $\epsilon = 2H\Delta$. We note that under the assumption that $K \geq SA$ and the fact that $vS_1 \leq SA$, we have $\Delta \leq \frac{1}{4}$. We will define $v^{S_1 \times M}$ $\epsilon$-MPERL problem instances, each indexed by an element in $[v]^{S_1 \times M}$. It suffices to show that, on at least one of the instances, $\mathbb{E} \left[ \text{Reg}_{\text{Alg}}(K) \right] \geq \Omega \left( M \sqrt{H^2 l^C K} \right)$.

**Facts about $v$.** There are two helpful facts about $v$ that can be easily verified:

- $vS_1 \geq \frac{1}{2}l^C$. This is true because, by definition, $vS_1 \geq S_1A - l - S_1 = S_1A - (SA - l^C) - S_1 = l^C - (SA - S_1A) - S_1 = l^C - (2(H-1)A + S_1)$; since, by assumption, $l \leq SA - 4(S + HA)$, we have $l^C \geq 4(HA + S) \geq 2(2(H-1)A + S_1)$; it then follows that $vS_1 \geq l^C - (2(H-1)A + S_1) \geq \frac{1}{2}l^C$.

- $v \geq 2$. This is true because, as shown above, $vS_1 \geq \frac{1}{2}l^C$ and $l^C \geq 4(HA + S)$, which imply that $v \geq \frac{2(HA+S)}{S_1} \geq \frac{2S_1}{S_1} = 2$.

**Construction.** For $\mathbf{a} = (a_{1,1}, \ldots, a_{1,M}, a_{2,1}, \ldots, a_{S_1,M}) \in [v]^{S_1 \times M}$, we define the following $\epsilon$-MPERL problem instance, $\mathfrak{M}(\mathbf{a}) = \{\mathcal{M}_p\}_{p=1}^{M}$, with $S$ states, $A$ actions, and an episode length of $H$, such that for each $p \in [M]$, $\mathcal{M}_p$ is constructed in the same way as it is for case 1, except for the transition probabilities of $(s, a) \in \mathcal{S}_1 \times \mathcal{A}$:

- For each state $s \in [S_1]$,

$$
\mathbb{P}_p(S_1 + 1 \mid s, a) = \begin{cases} \frac{1}{2} + \Delta, & \text{if } a = a_{s,p}; \\ \frac{1}{2}, & \text{if } a \in [v] \setminus a_{s,p}; \\ 0, & \text{if } a \notin [v]; \end{cases}
$$

and for each $a \in \mathcal{A}$, $\mathbb{P}_p(S_1 + 2 \mid s, a) = 1 - \mathbb{P}_p(S_1 + 1 \mid s, a)$, and $R_p(s, a) = 0$.

We now verify that $\mathfrak{M}(\mathbf{a})$ is an $\epsilon$-MPMAB problem instance. It can be easily observed that the reward distributions are the same for all players, i.e., for every $p, q \in [M]$ and every $(s, a) \in \mathcal{S} \times \mathcal{A}$,

$$
|R_p(s, a) - R_q(s, a)| = 0 \leq \epsilon.
$$

Regarding the transition probabilities, for every $(s, a) \in ((\mathcal{S}_1 \times (\mathcal{A} \setminus [v]))) \cup ((\mathcal{S} \setminus \mathcal{S}_1) \times \mathcal{A})$, we observe that the transition probabilities are the same for all players. Furthermore, for every $p, q \in [M]$ and every $(s, a) \in \mathcal{S}_1 \times [v]$,

$$
\left\| \mathbb{P}_p(\cdot \mid s, a) - \mathbb{P}_q(\cdot \mid s, a) \right\|_1 \leq 2\Delta = \frac{\epsilon}{H}.
$$

Therefore, $\mathfrak{M}(\mathbf{a})$ is an $\epsilon$-MPMAB problem instance.

**Suboptimality gaps.** Similar to the arguments in Case 1, it can be shown that for every $p \in [M]$, and every $(s, a) \in (\mathcal{S} \setminus \mathcal{S}_1) \times \mathcal{A}$, $\text{gap}_p(s, a) = 0$. And, for every $p \in [M]$, and every $s \in \mathcal{S}_1$,

$$
\text{gap}_p(s, a) = \begin{cases} 0, & \text{if } a = a_{s,p}; \\ (H-1)\Delta, & \text{if } a \in [v] \setminus a_{s,p}; \\ (H-1)\left(\frac{1}{2} + \Delta\right), & \text{if } a \notin [v]. \end{cases}
$$

**Subpar state-action pairs.** Based on the above construction, for every $(s, a) \in \mathcal{S}_1 \times (\mathcal{A} \setminus [v])$ and every $p \in [M]$, $\text{gap}_p(s, a) = (H-1)\left(\frac{1}{2} + \Delta\right) \geq 3(H-1)\Delta = \frac{3(H-1)}{2H}\epsilon \geq \frac{3}{4}\epsilon \geq 96H\left(\frac{\epsilon}{192H}\right)$, where the first inequality uses the fact that $\Delta \leq \frac{1}{4}$. Therefore, there are at least $(A - v)S_1 = uS_1 \geq l$ state-action pairs in $\mathcal{I}_{\frac{\epsilon}{192H}}$, i.e., $\left|\mathcal{I}_{\frac{\epsilon}{192H}}\right| \geq l$.

Now, it suffices to prove that

$$
\mathbb{E}_{\mathbf{a} \sim \text{Unif}([v]^{S_1 \times M})} \mathbb{E}_{\mathfrak{M}(\mathbf{a})} \left[ \text{Reg}_{\text{Alg}}(K) \right] \geq \frac{1}{240} M\sqrt{H^2 l^C K},
$$

where we recall that $\mathbf{a} = (a_{1,1}, \ldots, a_{1,M}, a_{2,1}, \ldots, a_{S_1,M})$. It suffices to show, for any $s' \in [S_1]$ and any $p' \in [M]$,

$$
\mathbb{E}_{\mathbf{a} \sim \text{Unif}([v]^{S_1 \times M})} \mathbb{E}_{\mathfrak{M}(\mathbf{a})} \left[ N_{p'}^{K+1}(s') - n_{p'}^{K+1}(s', a_{s'}) \right] \geq \frac{K}{4S_1}, \tag{64}
$$

where $N_{p'}^{K+1}(s') = \sum_{a \in \mathcal{A}} n_{p'}^{K+1}(s', a)$. To see this, by Lemma 29, we have

$$\mathbb{E}_{\mathbf{a} \sim \mathrm{Unif}([v]^{S_1 \times M})} \mathbb{E}_{\mathfrak{M}(\mathbf{a})} \left[ \mathrm{Reg}_{\mathrm{Alg}}(K) \right] \geq \sum_{p=1}^{M} \sum_{s' \in \mathcal{S}_1} (H-1)\Delta \cdot \mathbb{E}_{\mathbf{a} \sim \mathrm{Unif}([v]^{S_1 \times M})} \mathbb{E}_{\mathfrak{M}(\mathbf{a})} \left[ N_p^{K+1}(s') - n_p^{K+1}(s', a_{s'}) \right]$$

$$\geq \frac{H-1}{4} M K \sqrt{\frac{vS_1}{384K}}$$

$$\geq \frac{1}{160} M \sqrt{H^2(vS_1)K}$$

$$\geq \frac{1}{240} M \sqrt{H^2 l^C K},$$

where the last inequality uses the fact that $vS_1 \geq \frac{1}{2} l^C$.

Without loss of generality, let us choose $s' = 1$ and $p' = 1$. Similar to case 1, we define a set of helper problem instances: for any $(a_{1,2}, \ldots, a_{S_1,M}) \in [v]^{S_1 \times M - 1}$, we define a problem instance $\mathfrak{M}(0, a_{1,2}, \ldots, a_{S_1,M})$ such that it agrees with $\mathfrak{M}(a_{1,1}, a_{1,2}, \ldots, a_{S_1,M})$ on everything but $\mathbb{P}_1(\cdot \mid 1, a_1)$, namely, in $\mathfrak{M}(0, a_{1,2}, \ldots, a_{S_1,M})$, $\mathbb{P}_1(S_1 + 1 \mid 1, a_1) = \frac{1}{2}$.

For each $(j, a_{1,2}, \ldots, a_{S_1,M}) \in ([0] \cup [v]) \times [v]^{S_1 \times M - 1}$, let $\mathbb{P}_{j, a_{1,2}, \ldots, a_{S_1,M}}$ denote the probability measure on the outcomes of running Alg on the problem instance $\mathfrak{M}(j, a_{1,2}, \ldots, a_{S_1,M})$. Further, for each $j \in \{0\} \cup [v]$, we define

$$\mathbb{P}_j = \frac{1}{v^{S_1 \times M - 1}} \sum_{a_{1,2}, \ldots, a_{S_1,M} \in [v]^{S_1 \times M - 1}} \mathbb{P}_{j, a_{1,2}, \ldots, a_{S_1,M}};$$

and we use $\mathbb{E}_j$ to denote the expectation with respect to $\mathbb{P}_j$. In subsequent calculations, for any $m \in ([0] \cup [v]) \times [v]^{S_1 \times M - 1}$, we also denote by $\mathbb{P}_m \left( \cdot \mid N_1^{K+1}(1) \right)$ and $\mathbb{E}_m \left[ \cdot \mid N_1^{K+1}(1) \right]$ the probability and expectation conditional on a realization of $N_1^{K+1}(1)$ under $\mathbb{P}_m$. Similar to case 1, it can be shown that, for any $j \in \{0\} \cup [v]$,

$$\mathbb{P}_j(\cdot \mid N^{K+1}(1)) = \frac{1}{v^{S_1 \times M - 1}} \sum_{a_{1,2}, \ldots, a_{S_1,M} \in [v]^{S_1 \times M - 1}} \mathbb{P}_{j, a_{1,2}, \ldots, a_{S_1,M}} \left( \cdot \mid N^{K+1}(1) \right). \quad (65)$$

Now, for each $j \in [v]$, we have

$$\mathbb{E}_j \left[ n_1^{K+1}(1, j) \mid N_1^{K+1}(1) \right] - \mathbb{E}_0 \left[ n_1^{K+1}(1, j) \mid N_1^{K+1}(1) \right]$$

$$\leq N_1^{K+1}(1) \left\| \mathbb{P}_j \left( \cdot \mid N_1^{K+1}(1) \right) - \mathbb{P}_0 \left( \cdot \mid N_1^{K+1}(1) \right) \right\|_1$$

$$\leq N_1^{K+1}(1) \cdot \frac{1}{v^{S_1 \times M - 1}} \sum_{a_{1,2}, \ldots, a_{S_1,M} \in [v]^{S_1 \times M - 1}} \left\| \mathbb{P}_{j, a_{1,2}, \ldots, a_{S_1,M}} \left( \cdot \mid N_1^{K+1}(1) \right) - \mathbb{P}_{0, a_{1,2}, \ldots, a_{S_1,M}} \left( \cdot \mid N_1^{K+1}(1) \right) \right\|_1$$

$$\leq N_1^{K+1}(1) \cdot \frac{1}{v^{S_1 \times M - 1}} \sum_{a_{1,2}, \ldots, a_{S_1,M} \in [v]^{S_1 \times M - 1}} \sqrt{2 \mathrm{KL} \left( \mathrm{Ber}(\frac{1}{2} + \Delta), \mathrm{Ber}(\frac{1}{2}) \right) \mathbb{E}_{0, a_2, \ldots, a_{S_1}} \left[ n_1^{K+1}(1, j) \mid N_1^{K+1}(1) \right]}$$

$$\leq N_1^{K+1}(1) \cdot \frac{1}{v^{S_1 \times M - 1}} \sum_{a_{1,2}, \ldots, a_{S_1,M} \in [v]^{S_1 \times M - 1}} \sqrt{6\Delta^2 \mathbb{E}_{0, a_2, \ldots, a_{S_1}} \left[ n_1^{K+1}(1, j) \mid N_1^{K+1}(1) \right]}$$

$$\leq N_1^{K+1}(1) \cdot \sqrt{\frac{6vS_1}{384MK} \cdot \mathbb{E}_0 \left[ n_1^{K+1}(1, j) \mid N_1^{K+1}(1) \right]}$$

$$= \frac{1}{8} N_1^{K+1}(1) \sqrt{\frac{vS_1}{MK} \cdot \mathbb{E}_0 \left[ n_1^{K+1}(1, j) \mid N_1^{K+1}(1) \right]}. \quad (66)$$

where the first inequality is based on Lemma 24 and the fact that, conditional on $N_1^{K+1}(1)$, $n_1^{K+1}(1, j)$ has distribution supported on $[0, N_1^{K+1}(1)]$; the second inequality follows from Equation (65) and the triangle inequality; the third inequality uses Pinsker's inequality and Lemma 30 (the

divergence decomposition lemma); the fourth inequality uses Lemma 32 and the fact that $\Delta \leq \frac{1}{4}$; and the last inequality follows from Jensen's inequality.

Using arguments similar to the ones shown for case 1, we have that

$$\frac{1}{v} \sum_{j \in [v]} \mathbb{E}_j \left[ n_1^{K+1}(1, j) \right]$$

$$\leq \frac{1}{v} \mathbb{E}_0 \left[ n_1^{K+1}(1, j) \right] + \mathbb{E}_0 \left[ \frac{1}{8} N_1^{K+1}(1) \sqrt{\frac{vS_1}{K} \cdot \frac{1}{v} \sum_{j \in [v]} \mathbb{E}_0 \left[ n_1^{K+1}(1, j) \mid N_1^{K+1}(1) \right]} \right]$$

$$\leq \frac{1}{v} \mathbb{E}_0 \left[ N^{K+1}(1) \right] + \frac{1}{8} \sqrt{\frac{S_1}{K}} \cdot \mathbb{E}_0 \left[ \left( N_1^{K+1}(1) \right)^{\frac{3}{2}} \right]$$

$$\leq \frac{1}{v} \cdot \frac{K}{S_1} + \frac{1}{4} \sqrt{\frac{S_1}{K} \left( \frac{K}{S_1} \right)^3}$$

$$\leq \frac{3K}{4S_1},$$

where the second to last inequality is from Lemma 33 under the assumption that $K \geq S_1$, and the last inequality uses the fact that $v \geq 2$.

It then follows that

$$\frac{1}{v} \sum_{j \in [v]} \mathbb{E}_j \left[ N_1^{K+1}(1) - n_1^{K+1}(1, j) \right] \geq \frac{1}{v} \sum_{j \in [v]} \mathbb{E}_j \left[ N_1^{K+1}(1) \right] - \frac{K}{4S_1} = \frac{K}{4S_1},$$

and we thereby have shown that

$$\mathbb{E}_{\mathbf{a} \sim \mathrm{Unif}\left( [v]^{S_1 \times M} \right)} \mathbb{E}_{\mathfrak{M}(\mathbf{a})} \left[ N_1^{K+1}(1) - n_1^{K+1}(1, a_1) \right] \geq \frac{K}{4S_1}. \qquad \square$$

### D.3  Gap dependent lower bound

**Theorem 35** (Restatement of Theorem 8). *Fix $\epsilon \geq 0$. For any $S \in \mathbb{N}$, $A \geq 2$, $H \geq 2$, $M \in \mathbb{N}$, such that $S \geq 2(H-1)$, let $S_1 = S - 2(H-1)$; and let $\left\{ \Delta_{s,a,p} \right\}_{(s,a,p) \in [S_1] \times [A] \times [M]}$ be any set of values such that*

- *for every $(s, a, p) \in [S_1] \times [A] \times [M]$, $\Delta_{s,a,p} \in [0, H/48]$;*

- *for every $(s, p) \in [S_1] \times [M]$, there exists at least one action $a \in [A]$ such that $\Delta_{s,a,p} = 0$;*

- *and, for every $(s, a) \in [S_1] \times [A]$ and $p, q \in [M]$, $\left| \Delta_{s,a,p} - \Delta_{s,a,q} \right| \leq \epsilon/4$.*

*There exists an $\epsilon$-MPERL problem instance with $S$ states, $A$ actions, $M$ players and an episode length of $H$, such that $\mathcal{S}_1 = [S_1]$, $|\mathcal{S}_h| = 2$ for all $h \geq 2$, and*

$$\mathrm{gap}_p(s, a) = \Delta_{s,a,p}, \quad \forall (s, a, p) \in [S_1] \times [A] \times [M].$$

*For this problem instance, any sublinear regret algorithm $\mathrm{Alg}$ for the $\epsilon$-MPERL problem must have regret at least*

$$\mathbb{E} \left[ \mathrm{Reg}_{\mathrm{Alg}}(K) \right] \geq \Omega \left( \ln K \left( \sum_{\substack{p \in [M]}} \sum_{\substack{(s,a) \in \mathcal{I}^C_{(\epsilon/192H)}: \\ \mathrm{gap}_p(s,a) > 0}} \frac{H^2}{\mathrm{gap}_p(s, a)} + \sum_{(s,a) \in \mathcal{I}_{(\epsilon/192H)}} \frac{H^2}{\min_p \mathrm{gap}_p(s, a)} \right) \right).$$

*Proof.* The construction and techniques in this proof are inspired by [36] and [43].

**Proof outline.** We will construct an $\epsilon$-MPERL problem instance, $\mathfrak{M}$, and show that, for any sublinear regret algorithm and sufficiently large $K$, the following two claims are true:

1. for any $(s, a) \in \mathcal{S} \times \mathcal{A}$ such that for all $p$, $\mathrm{gap}_p(s, a) > 0$,

$$\mathbb{E}_{\mathfrak{M}}\left[n^K(s, a)\right] \geq \Omega\left(\frac{H^2}{\left(\min_p \mathrm{gap}_p(s, a)\right)^2} \ln K\right); \tag{67}$$

2. for any $(s, a) \in \mathcal{I}^C_{\frac{\epsilon}{192H}}$ and any $p \in [M]$ such that $\mathrm{gap}_p(s, a) > 0$,

$$\mathbb{E}_{\mathfrak{M}}\left[n_p^K(s, a)\right] \geq \Omega\left(\frac{H^2}{\left(\mathrm{gap}_p(s, a)\right)^2} \ln K\right). \tag{68}$$

The rest then follows from Lemma 29 (the regret decomposition lemma).

**Construction of $\mathfrak{M}$.** Given any set of values $\left\{\Delta_{s,a,p}\right\}_{(s,a,p)\in[S_1]\times[A]\times[M]}$ that satisfies the assumptions in the theorem statement, we can construct a collection of MDPs $\left\{\mathcal{M}_p\right\}_{p=1}^M$, such that for each $p \in [M]$, $\mathcal{M}_p$ is as follows, and $\mathfrak{M} = \left\{\mathcal{M}_p\right\}_{p=1}^M$ is an $\epsilon$-MPERL problem instance:

- $\mathcal{S}_1 = [S_1]$, and $p_0$ is a uniform distribution over the states in $\mathcal{S}_1$.

- For $h \in [2, H]$, $\mathcal{S}_h = \{S_1 + 2h - 3, S_1 + 2h - 2\}$.

- $\mathcal{A} = [A]$.

- For all $(s, a) \in \mathcal{S} \times \mathcal{A}$, the reward distribution $r_p(s, a)$ is a Bernoulli distribution, $\mathrm{Ber}(R_p(s, a))$, and we specify $R_p(s, a)$ subsequently.

- For every $(s, a) \in \mathcal{S}_1 \times [A]$, set $\bar{\Delta}^p_{s,a} = \frac{\Delta_{s,a,p}}{H-1}$. Then, let

$$\mathbb{P}_p\left(S_1 + 1 \mid s, a\right) = \frac{1}{2} - \bar{\Delta}^p_{s,a}, \quad \mathbb{P}_p\left(S_1 + 2 \mid s, a\right) = \frac{1}{2} + \bar{\Delta}^p_{s,a},$$

and $R_p(s, a) = 0$. Since $\Delta_{s,a,p} \in [0, H/48]$, $\bar{\Delta}^p_{s,a} \leq \frac{H}{48(H-1)} \leq \frac{1}{24}$, where the last inequality follows from the assumption that $H \geq 2$. Therefore, $\mathbb{P}_p\left(S_1 + 1 \mid s, a\right) \in [0, 1]$, and $\mathbb{P}_p\left(S_1 + 2 \mid s, a\right) \in [0, 1]$.

- For $h \in [2, H]$, and $a \in [A]$, let

  - $\mathbb{P}_p\left(S_1 + 2h - 1 \mid S_1 + 2h - 3, a\right) = 1$, $\mathbb{P}_p\left(S_1 + 2h \mid S_1 + 2h - 3, a\right) = 0$, and $R_p(S_1 + 2h - 3, a) = 1$.
  - $\mathbb{P}_p\left(S_1 + 2h \mid S_1 + 2h - 2, a\right) = 0$, $\mathbb{P}_p\left(S_1 + 2h - 1 \mid S_1 + 2h - 2, a\right) = 1$, and $R_p(S_1 + 2h - 2, a) = 0$.

By the assumption that for every $(s, p) \in [S_1] \times [M]$, there exists at least one action $a \in [A]$ such that $\Delta_{s,a,p} = 0$, we have that there is at least one action $a$ such that $\bar{\Delta}^p_{s,a} = 0$. We verify that for every $(s, a, p) \in [S_1] \times [A] \times [M]$,

$$\begin{aligned}
\mathrm{gap}_p(s, a) &= V_p^\star(s) - Q_p^\star(s, a) \\
&= \max_{a'} Q_p^\star(s, a') - Q_p^\star(s, a) \\
&= (H - 1)\bar{\Delta}^p_{s,a} \\
&= \Delta_{s,a,p}.
\end{aligned}$$

We now verify that the above MPERL problem instance $\mathfrak{M} = \{\mathcal{M}_p\}_{p=1}^M$ is an $\epsilon$-MPERL problem instance:

1. The reward distributions are the same for all players, namely, for all $p, q$,

$$\left| R_p(s,a) - R_q(s,a) \right| = 0 \le \epsilon, \forall (s,a) \in \mathcal{S} \times \mathcal{A}.$$

2. Further, by the assumption that for every $(s,a) \in [S_1] \times [A]$ and $p, q \in [M]$, $\left| \Delta_{s,a,p} - \Delta_{s,a,q} \right| \le \epsilon/4$, we have that

$$\left| \bar{\Delta}_{s,a}^p - \bar{\Delta}_{s,a}^q \right| = \frac{\left| \Delta_{s,a,p} - \Delta_{s,a,q} \right|}{H-1} \le \frac{\epsilon}{4(H-1)} \le \frac{\epsilon}{2H}.$$

It then follows that

$$\| \mathbb{P}_p \left( \cdot \mid s, a \right) - \mathbb{P}_q \left( \cdot \mid s, a \right) \|_1 = 2 \left| \bar{\Delta}_{s,a}^p - \bar{\Delta}_{s,a}^q \right| \le \frac{\epsilon}{H}.$$

Meanwhile, for every $(s,a) \in \left( \mathcal{S} \setminus \mathcal{S}_1 \right) \times \mathcal{A}$

$$\| \mathbb{P}_p \left( \cdot \mid s, a \right) - \mathbb{P}_q \left( \cdot \mid s, a \right) \|_1 = 0 \le \frac{\epsilon}{H}.$$

In summary, for every $(s,a) \in \mathcal{S} \times \mathcal{A}$,

$$\| \mathbb{P}_p \left( \cdot \mid s, a \right) - \mathbb{P}_q \left( \cdot \mid s, a \right) \|_1 \le \frac{\epsilon}{H}.$$

We are now ready to prove the two claims:

1. **Proving claim 1 (Equation (67)):**

   Fix any $(s_0, a_0) \in [S_1] \times [A]$ such that $\bar{\Delta}_{s_0,a_0}^{\min} = \min_p \bar{\Delta}_{s_0,a_0}^p > 0$. It can be easily observed that $\operatorname{gap}_p(s_0, a_0) > 0$ for all $p$. Define $p_0 = \operatorname{argmin}_p \bar{\Delta}_{s_0,a_0}^p$. We can construct a new problem instance, $\mathfrak{M}'$, which agrees with $\mathfrak{M}$, except that

$$\forall p \in [M], \mathbb{P}_p \left( S_1 + 1 \mid s_0, a_0 \right) = \frac{1}{2} - \bar{\Delta}_{s_0,a_0}^p + 2\bar{\Delta}_{s_0,a_0}^{\min}, \mathbb{P}_p \left( S_1 + 2 \mid s_0, a_0 \right) = \frac{1}{2} + \bar{\Delta}_{s_0,a_0}^p - 2\bar{\Delta}_{s_0,a_0}^{\min}.$$

   $\mathfrak{M}'$ is an $\epsilon$-MPERL problem instance. To see this, we note that the only change is in $\mathbb{P}_p \left( \cdot \mid s_0, a_0 \right)$ for all $p \in [M]$. In this new instance, it is still true that for every $p, q \in [M]$,

$$\| \mathbb{P}_p \left( \cdot \mid s_0, a_0 \right) - \mathbb{P}_q \left( \cdot \mid s_0, a_0 \right) \|_1 = 2 \left| \bar{\Delta}_{s_0,a_0}^p - \bar{\Delta}_{s_0,a_0}^q \right| \le \frac{\epsilon}{H}.$$

   Fix any sublinear regret algorithm Alg for the $\epsilon$-MPERL problem. By Lemma 30 (the divergence decomposition lemma), we have

$$\mathrm{KL}(\mathbb{P}_{\mathfrak{M}}, \mathbb{P}_{\mathfrak{M}'}) = \sum_{p=1}^M \mathbb{E}_{\mathfrak{M}} \left[ n_p^K(s_0, a_0) \right] \mathrm{KL} \left( \mathbb{P}_p^{\mathfrak{M}}(\cdot \mid s_0, a_0), \mathbb{P}_p^{\mathfrak{M}'}(\cdot \mid s_0, a_0) \right),$$

   where $\mathbb{P}_{\mathfrak{M}}$ and $\mathbb{P}_{\mathfrak{M}'}$ are the probability measures on the outcomes of running Alg on $\mathfrak{M}$ and $\mathfrak{M}'$, respectively; $\mathbb{P}_p^{\mathfrak{M}}(\cdot \mid s_0, a_0)$, $\mathbb{P}_p^{\mathfrak{M}'}(\cdot \mid s_0, a_0)$ are the transition probabilities for $(s_0, a_0)$ and player $p$ in $\mathfrak{M}$ and $\mathfrak{M}'$, respectively.

   We observe that, for any $p \in [M]$,

$$\mathrm{KL} \left( \mathbb{P}_p^{\mathfrak{M}}(\cdot \mid s_0, a_0), \mathbb{P}_p^{\mathfrak{M}'}(\cdot \mid s_0, a_0) \right)$$

$$= \mathrm{KL} \left( \mathrm{Ber} \left( \frac{1}{2} - \bar{\Delta}_{s_0,a_0}^p \right), \mathrm{Ber} \left( \frac{1}{2} - \bar{\Delta}_{s_0,a_0}^p + 2\bar{\Delta}_{s_0,a_0}^{\min} \right) \right)$$

$$\le 12 (\bar{\Delta}_{s_0,a_0}^{\min})^2,$$

where the last inequality follows from Lemma 32 and the assumption that $\Delta_{s,a,p} \leq \frac{H}{48}$.

In addition, $\sum_{p=1}^{M} \mathbb{E}_{\mathfrak{M}} \left[ n_p^K(s_0, a_0) \right] = \mathbb{E}_{\mathfrak{M}} \left[ n^K(s_0, a_0) \right]$. It then follows that

$$\mathrm{KL}(\mathbb{P}_{\mathfrak{M}}, \mathbb{P}_{\mathfrak{M}'}) \leq 12 \mathbb{E}_{\mathfrak{M}} \left[ n^K(s_0, a_0) \right] (\bar{\Delta}_{s_0,a_0}^{\min})^2. \tag{69}$$

Now, in the original $\epsilon$-MPERL problem instance, $\mathfrak{M}$, by Equation (57) and Markov's Inequality, we have

$$\mathbb{E}_{\mathfrak{M}} \left[ \mathrm{Reg}_{\mathrm{Alg}}(K) \right] \geq \frac{K}{4S_1} \left( (H-1) \bar{\Delta}_{s_0,a_0}^{\min} \right) \mathbb{P}_{\mathfrak{M}} \left( n_{p_0}^K(s_0, a_0) \geq \frac{K}{4S_1} \right);$$

where we note that $\bar{\Delta}_{s_0,a_0}^{p_0} = \bar{\Delta}_{s_0,a_0}^{\min}$. In $\mathfrak{M}'$, the new $\epsilon$-MPERL problem instance, we have

$$\begin{aligned}
\mathbb{E}_{\mathfrak{M}'} \left[ \mathrm{Reg}_{\mathrm{Alg}}(K) \right] &\geq \left( (H-1) \bar{\Delta}_{s_0,a_0}^{\min} \right) \mathbb{E}_{\mathfrak{M}'} \left[ \sum_{a \neq a_0} n_{p_0}(s_0, a) \right] \\
&= \left( (H-1) \bar{\Delta}_{s_0,a_0}^{\min} \right) \mathbb{E}_{\mathfrak{M}'} \left[ N_{p_0}^K(s_0) - n_{p_0}(s_0, a_0) \right] \\
&\geq \frac{K}{4S_1} \left( (H-1) \bar{\Delta}_{s_0,a_0}^{\min} \right) \mathbb{P}_{\mathfrak{M}'} \left( N_{p_0}^K(s_0) - n_{p_0}(s_0, a_0) \geq \frac{K}{4S_1} \right) \\
&\geq \frac{K}{4S_1} \left( (H-1) \bar{\Delta}_{s_0,a_0}^{\min} \right) \mathbb{P}_{\mathfrak{M}'} \left( N_{p_0}^K(s_0) \geq \frac{K}{2S_1}, n_{p_0}(s_0, a_0) \leq \frac{K}{4S_1} \right) \\
&\geq \frac{K}{4S_1} \left( (H-1) \bar{\Delta}_{s_0,a_0}^{\min} \right) \left( \mathbb{P}_{\mathfrak{M}'} \left( n_{p_0}(s_0, a_0) \leq \frac{K}{4S_1} \right) - \exp(-\frac{K}{8S_1}) \right),
\end{aligned}$$

where the first inequality is by Equation (57); the second inequality is by Markov's Inequality; the third inequality is by simple algebra; and the last inequality is by Chernoff bound that $\mathbb{P}_{\mathfrak{M}'} \left( N_{p_0}^K(s_0) < \frac{K}{2S_1} \right) \leq \exp(-\frac{K}{8S_1})$, and $\mathbb{P}(A \cap B) \geq \mathbb{P}(B) - \mathbb{P}(A^C)$ for events $A, B$.

It then follows that

$$\begin{aligned}
&\mathbb{E}_{\mathfrak{M}} \left[ \mathrm{Reg}_{\mathrm{Alg}}(K) \right] + \mathbb{E}_{\mathfrak{M}'} \left[ \mathrm{Reg}_{\mathrm{Alg}}(K) \right] \\
&= \frac{K}{2} \left( (H-1) \bar{\Delta}_{s_0,a_0}^{\min} \right) \left( \mathbb{P}_{\mathfrak{M}} \left( n_{p_0}^K(s_0, a_0) \geq \frac{K}{2} \right) + \mathbb{P}_{\mathfrak{M}'} \left( n_{p_0}^K(s_0, a_0) < \frac{K}{2} \right) - \exp(-\frac{K}{8S_1}) \right) \\
&\geq \frac{K}{2} \left( (H-1) \bar{\Delta}_{s_0,a_0}^{\min} \right) \left( \frac{1}{2} \exp \left( -\mathrm{KL}(\mathbb{P}_{\mathfrak{M}}, \mathbb{P}_{\mathfrak{M}'}) \right) - \exp(-\frac{K}{8S_1}) \right) \\
&\geq \frac{K}{2} \left( (H-1) \bar{\Delta}_{s_0,a_0}^{\min} \right) \left( \frac{1}{2} \exp \left( -12 \mathbb{E}_{\mathfrak{M}} \left[ n^K(s_0, a_0) \right] (\bar{\Delta}_{s_0,a_0}^{\min})^2 \right) - \exp(-\frac{K}{8S_1}) \right),
\end{aligned}$$

where the first inequality follows from Lemma 31 (the Bretagnolle-Huber inequality), and the second inequality follows from Eq. (69). Observe that $\mathbb{E}_{\mathfrak{M}} \left[ n^K(s_0, a_0) \right] \leq \frac{K}{S_1}$; in addition, by our assumption that $\Delta_{s,a,p} \leq \frac{H}{48}$ for every $(s, a, p)$, we have $\bar{\Delta}_{s_0,a_0}^{\min} \leq \frac{1}{24}$. These together implies that $\frac{1}{4} \exp \left( -12 \mathbb{E}_{\mathfrak{M}} \left[ n^K(s_0, a_0) \right] (\bar{\Delta}_{s_0,a_0}^{\min})^2 \right) \geq \exp(-\frac{K}{8S_1})$. Therefore, we have

$$\mathbb{E}_{\mathfrak{M}} \left[ \mathrm{Reg}_{\mathrm{Alg}}(K) \right] + \mathbb{E}_{\mathfrak{M}'} \left[ \mathrm{Reg}_{\mathrm{Alg}}(K) \right] \geq \frac{K}{2} \left( (H-1) \bar{\Delta}_{s_0,a_0}^{\min} \right) \cdot \frac{1}{4} \exp \left( -12 \mathbb{E}_{\mathfrak{M}} \left[ n^K(s_0, a_0) \right] (\bar{\Delta}_{s_0,a_0}^{\min})^2 \right).$$

Now, under the assumption that $\mathrm{Alg}$ is a sublinear regret algorithm, we have

$$\frac{K}{8} \left( (H-1) \bar{\Delta}_{s_0,a_0}^{\min} \right) \exp \left( -12 \mathbb{E}_{\mathfrak{M}} \left[ n^K(s_0, a_0) \right] (\bar{\Delta}_{s_0,a_0}^{\min})^2 \right) \leq 2CK^{\alpha}.$$

It follows that

$$\mathbb{E}_{\mathfrak{M}}\left[n^K(s_0, a_0)\right] \geq \frac{1}{12\left(\bar{\Delta}^{\min}_{s_0, a_0}\right)^2} \ln\left(\frac{(H-1)\bar{\Delta}^{\min}_{s_0, a_0} K^{1-\alpha}}{16C}\right)$$

$$= \frac{(H-1)^2}{12\left(\min_p \text{gap}_p(s_0, a_0)\right)^2} \ln\left(\frac{\min_p \text{gap}_p(s_0, a_0) K^{1-\alpha}}{16C}\right)$$

$$\geq \frac{H^2}{24\left(\min_p \text{gap}_p(s_0, a_0)\right)^2} \ln\left(\frac{\min_p \text{gap}_p(s_0, a_0) K^{1-\alpha}}{16C}\right).$$

We then have

$$\mathbb{E}_{\mathfrak{M}}\left[n^K(s_0, a_0)\right] \geq \Omega\left(\frac{H^2}{\left(\min_p \text{gap}_p(s_0, a_0)\right)^2} \ln K\right).$$

2. **Proving Claim 2 (Equation (68)):**

Fix any $(s_0, a_0) \in \mathcal{I}^C_{\frac{\epsilon}{192H}}$ and $p_0 \in [M]$ such that $\bar{\Delta}^{p_0}_{(s_0, a_0)} > 0$, which means that $\text{gap}_{p_0}(s_0, a_0) > 0$. We have that for all $p \in [M]$,

$$\bar{\Delta}^p_{s_0, a_0} = \frac{\Delta^p_{s_0, a_0}}{H-1} = \frac{\text{gap}_p(s_0, a_0)}{H-1} \leq \frac{24H(\epsilon/(192H))}{(H-1)} \leq \frac{\epsilon}{8(H-1)} \leq \frac{\epsilon}{4H}. \tag{70}$$

We can construct a new problem instance, $\mathfrak{M}'$, which agrees with $\mathfrak{M}$ except that

$$\mathbb{P}_{p_0}\left(S_1 + 1 \mid s_0, a_0\right) = \frac{1}{2} - \bar{\Delta}^{p_0}_{s_0, a_0} + 2\bar{\Delta}^{p_0}_{s_0, a_0} = \frac{1}{2} + \bar{\Delta}^{p_0}_{s_0, a_0},$$

$$\mathbb{P}_{p_0}\left(S_1 + 2 \mid s_0, a_0\right) = \frac{1}{2} + \bar{\Delta}^{p_0}_{s_0, a_0} - 2\bar{\Delta}^{p_0}_{s_0, a_0} = \frac{1}{2} - \bar{\Delta}^{p_0}_{s_0, a_0}.$$

$\mathfrak{M}'$ is an $\epsilon$-MPERL problem instance. To see this, we note that the only change is in $\mathbb{P}_{p_0}\left(\cdot \mid s_0, a_0\right)$. In this new instance, it is still true that for any $q \neq p_0$,

$$\|\mathbb{P}_{p_0}(\cdot \mid s_0, a_0) - \mathbb{P}_q(\cdot \mid s_0, a_0)\|_1 \leq 2\left|\bar{\Delta}^{p_0}_{s_0, a_0} + \bar{\Delta}^q_{s_0, a_0}\right| \leq \frac{\epsilon}{H}.$$

where the last inequality uses Equation (70) that $\bar{\Delta}^p_{s_0, a_0} \leq \frac{\epsilon}{4H}$ for every $p \in [M]$.

Fix any sublinear regret algorithm Alg. By Lemma 30 (the divergence decomposition lemma), we have

$$\text{KL}(\mathbb{P}_{\mathfrak{M}}, \mathbb{P}_{\mathfrak{M}'}) = \mathbb{E}_{\mathfrak{M}}\left[n^K_{p_0}(s_0, a_0)\right] \text{KL}\left(\mathbb{P}^{\mathfrak{M}}_{p_0}(\cdot \mid s_0, a_0), \mathbb{P}^{\mathfrak{M}'}_{p_0}(\cdot \mid s_0, a_0)\right).$$

Using a similar reasoning as before, we can show that

$$\text{KL}(\mathbb{P}_{\mathfrak{M}}, \mathbb{P}_{\mathfrak{M}'}) \leq 12\mathbb{E}_{\mathfrak{M}}\left[n^K_{p_0}(s_0, a_0)\right](\bar{\Delta}^{p_0}_{s_0, a_0})^2. \tag{71}$$

Similar to case 1, we have the following argument. In the original $\epsilon$-MPERL problem instance, $\mathfrak{M}$, we have $\mathbb{E}_{\mathfrak{M}}\left[\text{Reg}_{\text{Alg}}(K)\right] \geq \frac{K}{4S_1}\left((H-1)\bar{\Delta}^{p_0}_{s_0, a_0}\right)\mathbb{P}_{\mathfrak{M}}\left(n^K_{p_0}(s_0, a_0) \geq \frac{K}{4S_1}\right)$; and in $\mathfrak{M}'$, the new $\epsilon$-MPERL problem instance, we have $\mathbb{E}_{\mathfrak{M}'}\left[\text{Reg}_{\text{Alg}}(K)\right] \geq \frac{K}{4S_1}\left((H-1)\bar{\Delta}^{p_0}_{s_0, a_0}\right)\left(\mathbb{P}_{\mathfrak{M}'}\left(n^K_{p_0}(s_0, a_0) < \frac{K}{4S_1}\right) - \exp(-\frac{K}{8S_1})\right).$

It then follows that

$$\mathbb{E}_{\mathfrak{M}}\left[\mathrm{Reg}_{\mathrm{Alg}}(K)\right] + \mathbb{E}_{\mathfrak{M}'}\left[\mathrm{Reg}_{\mathrm{Alg}}(K)\right]$$

$$\geq \frac{K}{2}\left((H-1)\bar{\Delta}_{s_0,a_0}^{p_0}\right)\left(\frac{1}{2}\exp\left(-\mathrm{KL}(\mathbb{P}_{\mathfrak{M}},\mathbb{P}_{\mathfrak{M}'})\right) - \exp(-\frac{K}{8S_1})\right)$$

$$\geq \frac{K}{8}\left((H-1)\bar{\Delta}_{s_0,a_0}^{p_0}\right)\exp\left(-12\mathbb{E}_{\mathfrak{M}}\left[n^K(s_0,a_0)\right](\bar{\Delta}_{s_0,a_0}^{p_0})^2\right).$$

Now, under the assumption that $\mathrm{Alg}$ is a sublinear regret algorithm, we have

$$\frac{K}{8}\left((H-1)\bar{\Delta}_{s_0,a_0}^{p_0}\right)\exp\left(-12\mathbb{E}_{\mathfrak{M}}\left[n_{p_0}^K(s_0,a_0)\right](\bar{\Delta}_{s_0,a_0}^{p_0})^2\right) \leq 2CK^\alpha.$$

It follows that

$$\mathbb{E}_{\mathfrak{M}}\left[n_{p_0}^K(s_0,a_0)\right] \geq \frac{1}{12\left(\bar{\Delta}_{s_0,a_0}^{p_0}\right)^2}\ln\left(\frac{(H-1)\bar{\Delta}_{s_0,a_0}^{p_0}K^{1-\alpha}}{16C}\right)$$

$$\geq \frac{H^2}{24\left(\mathrm{gap}_{p_0}(s_0,a_0)\right)^2}\ln\left(\frac{\mathrm{gap}_{p_0}(s_0,a_0)K^{1-\alpha}}{16C}\right).$$

We then have that

$$\mathbb{E}_{\mathfrak{M}}\left[n_{p_0}^K(s_0,a_0)\right] \geq \Omega\left(\frac{H^2}{\left(\mathrm{gap}_{p_0}(s_0,a_0)\right)^2}\ln K\right).$$

**Combing the two claims:** We note that in $\mathfrak{M}$, for any $(s,a,p) \in \left(\mathcal{S}\setminus\mathcal{S}_1\right)\times\mathcal{A}\times[M]$, $\mathrm{gap}_p(s,a) = 0$. It then follows from Lemma 29 (the regret decomposition lemma) and the fact that for any $(s,a,p) \in \mathcal{I}_{\epsilon/192H}\times[M], \mathrm{gap}_p(s,a) > 0$, that

$$\mathbb{E}\left[\mathrm{Reg}_{\mathrm{Alg}}(K)\right] \geq \sum_{p=1}^{M}\sum_{(s,a)\in\mathcal{S}_1\times\mathcal{A}}\mathbb{E}\left[n_p^K(s,a)\right]\mathrm{gap}_p(s,a)$$

$$\geq \Omega\left(\ln K\left(\sum_{\substack{p\in[M]\\(s,a)\in\mathcal{I}_{\epsilon/192H}^C:\\\mathrm{gap}_p(s,a)>0}}\frac{H^2}{\mathrm{gap}_p(s,a)} + \sum_{(s,a)\in\mathcal{I}_{\epsilon/192H}}\frac{H^2}{\min_p\mathrm{gap}_p(s,a)}\right)\right).$$

$\square$