# OpenReview forum: "Provably efficient multi-task reinforcement learning with model transfer"
_NeurIPS.cc/2021/Conference — NeurIPS 2021 Poster_

### Official Review · Reviewer_GxNs · 2021-07-15

**Rating:** 7
**Confidence:** 4

**Summary:**

This paper studies multi-task reinforcement learning (RL) in tabular episodic MDPs. It studies an online setting, by showing that when the agents are facing similar but different MDPs, their collective performance, in terms of both gap-dependent and -independent regrets, can be improved through information sharing. Almost matching lower bounds are also provided.

**Limitations And Societal Impact:**

Yes. Both limitations and potential societal impact have been discussed appropriately.

**Main Review:**

Originality: The multi-task RL definition in this paper, in terms of Definition 1, is new to my knowledge of the literature. The work combines multiple important techniques and ideas from the literature of multi-task bandits and single-task gap-dependent regret analysis. It is very clear how the work differs from the literature, and reading the related work is enjoyable.


Quality: The submission is technically sound, and the claims are well-supported by detailed proofs (though with several errors corrected in the errata in the appendix). This is definitely a complete piece of work. The authors have also done a good job discussing the weaknesses/limitations of the work. I have tried to do some sanity-check of the proofs, which turned out to be correct. However, as the appendix is very long, I may fail to catch some errors, if there are.


Clarity: The paper is well-written, well-organized, and enjoyable to read. It has also adequately informed the reader, by providing enough information to reproduce the results. As a theory paper, the claims are clear to read, and each result is followed by explanations to help the reader understand.


Significance: The results are important, since a good formulation and a sharp theoretical understanding of multi-task RL is relatively open. The results in this paper would open up several interesting future directions. The setting is defined clearly, and requires significant effort to address (compared to the counterpart in the bandit setting). The main contribution, which concerns how to use the data from other tasks by constructing \bar{agg-Q_p} and \underline{agg-Q_p}, is advancing the SOTA theory of the area, and is unique in the area of multi-task RL, to my knowledge.


Minor comments:
1. In pseudocode of Algorithm 1, it seems that line 2 and line 16 should have the same indent (both are for all p)? It would be clearer to specify the (initial and updated) values of ind-b_p and agg-b_p, and how hat{r} and hat{P} are updated (at least point to some equations in the paper).
2. Overflow of equations in the end of page 5.
3. The "e.g." is misplaced (should be outside "[]") in line 267. Same as in line 289.


Overall, this is a well-written paper, addressing an important theoretical question in RL, though by combining several ideas in the literature. The theoretical results are sound, as both (almost-tight) upper and lower bounds are provided.

**Time Spent Reviewing:**

3

---

> ### Author Response · Authors · 2021-08-10
> **Response**
>
> Thank you for your review and comments. We also hope that this work can open up future directions in multi-task RL.
>
> > In pseudocode of Algorithm 1, it seems that line 2 and line 16 should have the same indent (both are for all p)?
>
> Yes, thanks for catching this. We will make sure line 16 is correctly indented in the final version.
>
> > It would be clearer to specify the (initial and updated) values of ind-b_p and agg-b_p, and how hat{r} and hat{P} are updated (at least point to some equations in the paper).
>
> Thanks - we will refer to the respective equations in the algorithm.
>
> > Overflow of equations in the end of page 5.\
> > The "e.g." is misplaced (should be outside "[]") in line 267. Same as in line 289.
>
> Thanks for the suggestions - we will address these in the final version.

---

### Official Review · Reviewer_tFPt · 2021-07-15

**Rating:** 5
**Confidence:** 2

**Summary:**

This paper formulates a multi-task concurrent RL problem, even the tasks are not identical. it also proposes a prototype algorithm for tabular MDPs and provides theory bounds.

**Main Review:**

-Pros

-- This paper provides a clear problem formulation and theoretical derivation.

-Cons:

-- No experiment results to show the superiority of the proposed algorithm, even in a toy environment.


Frankly speaking, This paper does not fall in the main scope of my review range, I think the main idea is soundness but without verification of the deduction. It may provide some insight for multi-task RL, however, it's currently far away from the practical condition.

**Time Spent Reviewing:**

3

---

> ### Author Response · Authors · 2021-08-10
> **Response**
>
> Thank you for your review and comments. The main contribution of this work is to provide a theoretical study of multi-task RL and our goal is to lay the theoretical groundwork for characterizing the amenability of model transfer across different MDPs by providing regret upper and lower bounds. We leave the empirical evaluation of our algorithm as future work.

---

### Official Review · Reviewer_p2Ef · 2021-07-16

**Rating:** 6
**Confidence:** 2

**Summary:**

This paper studies a multi-task RL problem, called epsilon-Multi-Player Episodic Reinforcement Learning. This study got inspired by a recent study on multi-task multi-armed bandits. The assumption is that all tasks are defined with the same state and action spaces, and $epsilon$-dissimilar in terms of transition and reward functions. As the first main contribution, the authors define the notion of subpar state-action pairs which captures the amenability of information sharing among tasks. Then, the author propose a model-based algorithm MULTI-TASK-EULER algorithm, which exploits the given dissimilarity parameter as an advantageous information sharing among tasks to achieve better regret upper bounds in comparing to being without knowing $epsilon$. The authors also present the gap-dependent and gap-independent regret lower bounds.


**Ethics Review Area:**

["I don’t know"]

**Limitations And Societal Impact:**

Yes

**Main Review:**

In overall, the paper pursues an interesting research problem. The proposed method makes sense. It's interesting to see how each agent can exploit the knowledge transfer and the disssimilarity level to select actions. Both uncertainty are combined in a principled way to compute LB and UB.

Given the corrections in appendix, would the upper bound in Theorem 6 becomes larger, then how is it compared to the collective regret bound of the individual STRONG-EULER? And how should the discussion about the gap between UB and LB in the paragraph below Theorem 8 change?

The bounds can be arbitrarily large in a factor of H. In RL setting, this would not be interesting, where H can be very large or infinite. It's understandable that the proofs are simply inspired by those in the the multi-task multi-armed bandits paper [37]. However it will become more interesting if the bounds can be independent from H, e.g. with a different objective: infinite-horizon or discounted return.

An important point would be the proposed theory should be backed up by some experiment results, e.g. both on the multi-task multi-armed bandits [37] and one multi-task RL problem.




**Time Spent Reviewing:**

4

---

> ### Author Response · Authors · 2021-08-10
> **Response**
>
> Thank you for your review and comments. For your questions:
>
>
>
> > Given the corrections in appendix, would the upper bound in Theorem 6 becomes larger, then how is it compared to the collective regret bound of the individual STRONG-EULER?
>
> First, we apologize that the last logarithmic term under line 231 (the corresponding term in the regret of individual Strong-Euler) should also be revised to $\log(\frac{HSA}{\mathrm{gap}_{\min}})$. This follows from (Simchowitz and Jamieson, Corollary 2.1), where the $M$ symbol therein denotes $(HSA)^2$.
>
> You are still right that the last term of Theorem 6, $\log(\frac{MHSA}{\mathrm{gap}_{\min}})$, has an extra factor $M$ inside log compared to the last term in the equation under line 231; however, they are of the same order under mild conditions: so long as the number of tasks $M \leq \mathrm{poly}(HSA)$, the orders of these two terms will be the same.
>
> > And how should the discussion about the gap between UB and LB in the paragraph below Theorem 8 change?
>
> Theorems 6 and 8 provide *gap-dependent* regret guarantees of multi-task RL algorithms (including Multi-task-Euler) and focus on the dependence on the suboptimality gaps. We therefore view the trailing term $\ln(MSAHK) \cdot M S^2 A H^3 \cdot \ln(MHSA/\mathrm{gap}_\min)$ in Theorem 6 as a lower order term. The "nearly-matching"  discussions below Theorem 8 concerns the matching of the second and the third terms in Theorem 6 and the lower bound in Theorem 8, so it does not need to be changed.
>
>
> > The bounds can be arbitrarily large in a factor of H. In RL setting, this would not be interesting, where H can be very large or infinite. It's understandable that the proofs are simply inspired by those in the the multi-task multi-armed bandits paper [37]. However it will become more interesting if the bounds can be independent from H, e.g. with a different objective: infinite-horizon or discounted return.
>
> Thank you for the suggestion. Indeed, many recent works (see the first paragraph of Section 5) study the fixed-horizon episodic MDP setting, with regret guarantees scaling as $\tilde{O}( poly(H) \sqrt{K})$ or $\tilde{O}( poly(H) log(K) )$. The dependences on H are in general unavoidable (see, e.g., our lower bounds Theorem 7 and 8). Studying horizon-free regret bounds in other settings (e.g., the ones you suggested, or the setting of [WDYK20, JA18]) would be an interesting direction.
>
> [WDYK20] Wang, Du, Yang and Kakade, Is Long Horizon Reinforcement Learning More Difficult Than Short Horizon Reinforcement Learning? NeurIPS 2020. \
> [JA18] Jiang and Agarwal, Open Problem: The Dependence of Sample Complexity Lower Bounds on Planning Horizon, COLT 2018.

---

> > ### Comment · Reviewer_p2Ef · 2021-08-26
> > **Thanks**
> >
> > Thanks authors for the response.

---

### Official Review · Reviewer_VyG2 · 2021-08-03

**Rating:** 7
**Confidence:** 2

**Summary:**

The submission investigates the problem of multi-task reinforcement learning, which is where a collection of agents simultaneous learn similar tasks, ideally while exploiting the similarities in the task structure. The submission (i) proposes a formalism for this setting and (ii) designs and analyzes a model-based algorithm (Multi-task Euler) for the aforementioned formalism. The main engine Multi-task Euler is the exploitation of "subpar state-action pairs" --- i.e., state-action pairs which are suboptimal across all tasks. When there are many subpar state-action pairs, Multi-task Euler strictly outperforms Euler without information sharing. In any case, Multi-task Euler is never outperformed by Euler without information sharing.

**Limitations And Societal Impact:**

It is confusing that the first sentence of the societal impact paragraph is about the work's limitations, rather than its societal impact.

This sentence is worded awkwardly: To avoid unintended consequences when deployed in the real world, another major open problem is to ensure safety of our algorithm under model misspecification.

It also doesn't really clarify the submission's position of the societal impact of the work. If the submission does not believe the work will have any broader societal impact, then it should either simply say so, or remove the (one sentence) societal impact section altogether.

**Main Review:**

I found the paper well-organized and largely digestible, despite the technical nature of the information content.

Multi-task learning is an important problem. The formalism suggested by the paper seems reasonable. It seems undesirable that the allowed distance between transition probabilities scales inversely with the horizon length, but I can see how this constraint would be necessary to avoid compounding divergence.

The discussion on where the improvement comes from in Theorem 5 is nicely stated.

I was added as a last minute reviewer and so did not have time to go through the proofs. Additionally, I did not have time to go through the related work, with which I am unfamiliar. Thus, I am unfortunately not able to give an assessment with high confidence. That being said, the work appears both rigorous and topical, and I have no major complaints.

# Small Comments
Footnote 1: Isn’t this assumption necessarily true for MDPs with fixed horizon H?
Typo line 11 of algorithm? (ind-Q floor should be agg-Q ceiling?)

**Time Spent Reviewing:**

3

---

> ### Author Response · Authors · 2021-08-10
> **Response**
>
> Thank you for your review and comments. For your comments:
>
> - *Footnote 1*:
> While many recent works study a non-layered stationary episodic MDP setting (e.g., Simchowitz and Jamieson, 2019, abbrev. SJ19), we chose to focus on a layered episodic MDP setting mainly to ease the presentation of the suboptimality gap definition (in that it does not need to take minimum over all h’s as in SJ19), which will also simplify the presentation for our gap-dependent and gap-independent regret bounds.
>
> - *Typo line 11 of algorithm*:
> Yes, thank you for pointing out the typo - we will correct it.
>
> - *Societal impacts*:
> Thank you for your comments and suggestions on the societal impacts of our work. We are aware that reinforcement learning in general has certain societal impact because of the nature of an agent’s interaction with an environment; our work belongs to the theoretical studies of tabular MDPs, and we do not foresee any direct societal consequences. We will clarify this in the final version.

---

### Official Review · Reviewer_Bk4C · 2021-08-04

**Rating:** 6
**Confidence:** 4

**Summary:**

This paper is about multi-task learning in episodic tabular MDPs. M learner's interact with M tabular MDPs
with a shared state/action space but (potentially) different transition and reward functions. The authors
assume that the mean rewards are epsilon close over all state/action pairs and that the transition kernels are epsilon/H close
in L1 over all state/action pairs where H is the episode length. The learner's have unlimited communication at the end
of every episode.

The main contribution is a new algorithm and analysis showing that the information sharing can help when epsilon is small and
the number of state/action pairs with a large V^*(s) - Q*(s,a) gap is large. The basic idea is to maintain empirical estimates
of transitions and rewards for each learner individually and also an aggregate of all estimates. Of course the latter is somehow
an estimate of some average reward/transitions, so the algorithm adds to the corresponding confidence bounds a term that depends
on epsilon. Besides the modified bonus terms, the learner uses essentially an instance of the Euler algorithm for each MDP.
That's quite a natural idea, and the resulting bounds make sense.

The authors also provide lower bounds. These are of a minimax-ish flavour and show that there /exist/ MDPs where the upper bounds
are tight. I find this to be only quite weak evidence that the upper bounds are expressed using the "right" quantities, if such
things even exist.

**Limitations And Societal Impact:**

I have no concerns about this aspect.

**Main Review:**

Clarity:

The paper is generally very easy to read. I did collect quite a long list of minors below that the authors might find useful for
improving the manuscript.


Correctness:

The results pass basic sanity checks, but I did not check at all the proofs.


Impact:

Tabular MDPs won't find applications anytime soon (especially in complicated settings like this), so the main question is
what will be the theoretical impact. Actually the authors could do a lot better to explain the impact of the technical contribution. See
the next comment.


Novelty:

The specific setting is new to my knowledge, though I have not followed in great detail this line of work. The approach (at least in hindsight) feels
like about the first thing you would try.

What is missing from the authors is a clear explanation of the technical novelty of their analysis. If I were to dive into the 40 page appendix,
what parts will I find most exciting and why? How can I use the ideas in my papers? This is especially important because
the analysis is borrowing (as it must) a lot of material from the Euler analysis.


Overall:

The topic of study is interesting and the paper is well written. I would be much more excited if the authors can argue convincingly that they have developed
some new and useful techniques. Nevertheless I lean towards acceptance.


Minors:

* What if epsilon is unknown? Can you prove a lower bound in this case?

* L51. Given the minimax nature of the lower bounds, I am not sure the upper and lower bounds truly characterise the complexity of the problem.
Maybe the 'real' bound depends on some other complexity measures? Or there is a non-trivial Pareto frontier?

* L55. What is the complement of a set without a universe?

* L58. Using a \vee b = min(a, b) and a \wedge b = max(a, b) is a bit of an outrageous choice. Surely you want the other way around.

* L64. Some people might like a definition of Delta(S).

* L68. I am being petty, but the assumptions here say that P_H(. | s, a) must be supported on S_{H+1} = {bot} which is explicitly stated as not being
in S. And yet P_H(. | s, a) in Delta(S).

* L70. Some readers might like to know a little about the independence structure here.

* L77. More pettyness, but there is some dangerous language in these definitions. "expected return of player p conditioned on ..." which measure exactly
are you conditioning on and what event. Does this event really have positive probability (what if the policy will never reach some state). It seems to me
that the right way to define these value functions is via the Bellman equations, which are written just below anyways.

* L82. Maybe I missed it, but is R_p(s, a) defined? It should be the mean of r_p(s, a).

* L82. Since you have a layered MDP, I am wondering if you really need the h indices in the value function notation?

* L108. The O(sqrt(H^2 SAK)) claim is a bit hard to believe. First, there must be logarithms. Second, are there not lower order terms
that can be dominant in some regimes. I suggest to specify precisely what is meant by Big-O here. From a mathematicians perspective this is quite
non-standard.

* L111. Note that "large enough K" here is not a universal lower bound, but a problem-dependent one. So the use of the lesssim symbol may not be quite justified.

* L140. There is a little danger in calling a players own data unbiased. Even in the single-task setting there needs to be a great deal of effort needed
to overcome the fact that the data collected by a learning algorithm may be biased.

* L218. I find it a bit hard to believe there are no log factors in the dominant terms of Theorem 5 and similar results.

* L244. I am uncomfortable with the Big-O notation used in the statement of this theorem.

* L249. This definition of a sublinear regret algorithm would be clearer with a few quantifiers. I guess you want to fix a state/action space, episode length and
so forth and then say there exists a constant depending on these things such that for all transition kernels and reward distributions the regret is upper bounded
by such and such. Also, T should be K in this definition, right?

**Time Spent Reviewing:**

2

---

> ### Author Response · Authors · 2021-08-10
> **Response**
>
> Thank you for your detailed comments. We have also noticed some of the minor issues after the main paper submission deadline, and addressed them in the errata in the supplementary material; we will make sure to make a pass over the paper and incorporate your comments that haven’t been addressed yet. Here, we would also like to answer some of your questions:
>
> * The main technical novelty of this paper lies in:
>
>     * Algorithmically, although our Multi-task-Euler algorithm is based on the well-known "optimism in the face of uncertainty" principle, the construction of the optimistic value function estimates and optimistic policies are novel.\
>  \
>  Specifically, for a player $p \in [M]$, it is unclear apriori how to use trajectories collected by other players $q \neq p$ in MDP $\mathcal{M}_q$ to achieve more-refined optimistic value iteration for player $p$ in a sample-efficient and statistically consistent manner. To deal with this challenge, our algorithm maintains aggregate model estimates, and carefully adds corresponding bonus terms that take into account the \epsilon-dissimilarity among the MDPs. The design of our bonus terms may appear simple (as it basically adds $\epsilon$ factors to existing bonus terms and replaces the individual counts with aggregate counts), but showing that it possesses the optimism property is not trivial; see below.
>
>     * In the analysis, the key difference between our Multi-task-Euler and the strong-Euler algorithm is that, we prove the necessary concentration inequalities for establishing the strong optimism property with the usage of *heterogeneous, non-iid* data, specifically Lemma 13, Equations (25), (27), and (29) (note that the aggregate estimates $\hat{R}^k$, $\hat{\mathbb{P}}^k$ are calculated from data collected from $M$ heterogeneous MDPs).  \
>  \
>  To this end, we prove a generalization of empirical Bernstein inequality for heterogeneous data, namely Lemma 16, item 2, which we believe are new and non-trivial: specifically, although $\hat{\mathbb{P}}^k$ is a convex combination of $\hat{\mathbb{P}}^k_p$, the empirical variance $\mathrm{var}_{\hat{\mathbb{P}}^k(\cdot \mid s,a)}[V_p^*]$ is not linear in $\hat{\mathbb{P}}^k_p$ - this requires extra care when establishing its concentration. \
>  \
>  Finally, Multi-task-Euler’s refined surplus bound (Lemma 19) is novel: if we consider the individual strong-Euler baseline, the leading term in the surplus bound, $B_p^{k, \mathrm{lead}}$ term, formally defined under line 791, would only have its first and third terms present, which does not take into account inter-task similarity; in contrast, Multi-task-Euler’s $B_p^{k, \mathrm{lead}}$ term has an additional second term which involves the task similarity parameter $\epsilon$ and the aggregate count $n^k(s,a)$, which is the key to achieving a reduced regret compared to the baseline. See also the discussions in lines 568-573 in Appendix C.
>
>
> * Unknown $\epsilon$: this has been investigated in the multi-armed bandit setting ($H=1$ and $S=1$) in reference [37], where they have shown that: (1) achieving significantly-improved gap-dependent regret bounds than running UCB individually is impossible, unless we give up on sublinear regret guarantees in some environments; (2) achieving improved gap-independent regret bounds than running UCB individually is possible using bandit model selection algorithms. We conjecture that these results may generalize to the RL setting, and will investigate this in future work.
>
> * L58: Sorry about the typo - we intended to define $a \vee b = \max(a, b)$ and $a \wedge b = \min(a, b)$; we will correct it in the final version.
>
> * L108: You are right that we are missing two terms in the regret bound here: first, there is a $\log(HSAK)$ multiplicative factor missing; second, there is an additive term $M H^4 S^2 A \log^2(HSAK)$ missing; we will correct this in our final version. We will also define the big-$O$ and $\tilde{O}$ notation respectively in the final version, specifically $f(\cdot) = \tilde{O}(g(\cdot))$ (resp. $f(\cdot) = O(g(\cdot))$) iff there exists a numerical constant such that $f \leq c g \cdot \mathrm{polylog}(g)$ (resp. $f \leq c g$).

---

### Decision · Program_Chairs · 2021-09-27

**Decision:**

Accept (Poster)

**Comment:**

The paper presents a method to transform a multi-task MDP into a multi-player game where agents act in related but slightly environments, and share information. Several reviewers commented on the quality and clarity of writing, and while there were many specific technical questions, they were mostly clarified in the responses. One point of criticism from several reviewers is the lack of technical novelty. However, in the discussion, the reviewers also agreed on the importance of multi-task RL and that the theoretical results are state-of-the-art. As stated by two of the reviewers, experiments in common domains could be useful to demonstrate the practical applicability. I encourage the authors to consider this point in future work.